# Pathway level subtyping identifies a slow-cycling biological phenotype associated with poor clinical outcomes in colorectal cancer

Molecular stratification using gene-level transcriptional data has identified subtypes with distinctive genotypic and phenotypic traits, as exemplified by the consensus molecular subtypes (CMS) in colorectal cancer (CRC). Here, rather than gene-level data, we make use of gene ontology and biological activation state information for initial molecular class discovery. In doing so, we defined three pathway-derived subtypes (PDS) in CRC: PDS1 tumors, which are canonical/LGR5+ stem-rich, highly proliferative and display good prognosis; PDS2 tumors, which are regenerative/ANXA1+ stem-rich, with elevated stromal and immune tumor microenvironmental lineages; and PDS3 tumors, which represent a previously overlooked slow-cycling subset of tumors within CMS2 with reduced stem populations and increased differentiated lineages, particularly enterocytes and enteroendocrine cells, yet display the worst prognosis in locally advanced disease. These PDS3 phenotypic traits are evident across numerous bulk and single-cell datasets, and demark a series of subtle biological states that are currently under-represented in pre-clinical models and are not identified using existing subtyping classifiers.

Molecular subtyping in cancer has identified biomarkers that stratify tumors according to biological and clinical phenotypes, providing an improved understanding of the signaling underpinning tumor development and treatment response. Numerous studies have leveraged gene-level expression values to identify tumor subtypes[1–3], followed by downstream characterization using collections of pathway-level gene signatures that represent biologically important phenotypes[4]. The value of this approach is exemplified in CRC by the CMS[5], colorectal intrinsic subtypes (CRIS[3]) and the single-cell RNA sequencing (scRNA-seq)-based intrinsic subtypes (iCMS[6]). These gene-level subtyping approaches are dominated by associations with genetic alterations, particularly those underpinning the Vogelstein paradigm[7,8]. Although iCMS sub-stratified stromal CMS4 tumors, it did not identify any heterogeneity within the largest subtype of epithelial-rich tumors, CMS2.

Although subtype discovery using gene-level data represents the most commonly deployed approach, pathway-level data can provide a closer link with molecular mechanisms and clinical phenotypes[9]. Therefore, we reasoned that by using pathway-level data, initially in KRAS mutant (KRASmut) CRC tumors, we can identify a more comprehensive view of biological signaling related to disease.

Using transcriptional data from a series of colorectal tumors, we identify, validate and characterize a set of pathway-derived subtypes (PDS), providing unique insights into tumor biology regardless of mutational status. Although this was developed in bulk CRC, PDS classification reveals a previously unseen continuum of epithelial cell states within CMS2, associated with cell cycle, transcriptional activity and lineage differentiation states. These intrinsic biological traits are distinct from the iCMS classification system when applied to the same single-cell cohorts.

Overall, these data support the use of the PDS system in conjunction with existing subtyping approaches to ensure that tumor studies are informed by multiple tiers of cancer-relevant information that cannot be fully revealed by individual subtyping methods.

✉e-mail: p.dunne@qub.ac.uk

## Results

### Pathway-led CRC subtype discovery and classification

We generated a matrix of pathway-level single sample gene set enrichment analysis (ssGSEA) scores ($n$ = 1,783 per tumor) as an initial framework for mapping the tumor activation status in a subset of the FOCUS[10,11] clinical trial (ISRCTN79877428, $n$ = 360 tumors; S:CORT cohort) across $n$ = 640,000+ combinations of biological processes, including BIOCARTA, PID, KEGG and REACTOME collections from the Molecular Signature Database (MSigDB[4]). Following pathway-level conversion, *KRAS*mut samples ($n$ = 165) were used for unsupervised class discovery that, through $k$-means clustering, revealed three distinct PDS: PDS1 (~27%), PDS2 (~38%) and PDS3 (~35%) (Fig. 1a, Extended Data Fig. 1a–c and Supplementary Table 1). Assessment of mutations according to PDS revealed no distinct mutational type for *KRAS* itself (Fig. 1b) or numerous key mutations (Fig. 1c). Visualization of the $n$ = 626 gene sets most significantly associated with PDS revealed how transcriptionally distinct these genetically indistinguishable groups were (Fig. 1d and Supplementary Table 2). These analyses also indicated that CMS2 tumors, and to a lesser extent CMS3 tumors, were distributed across PDS1 and PDS3, whereas the CMS1 and CMS4 inflammatory and stromal tumors were merged within PDS2 (Fig. 1e,f and Extended Data Fig. 1d).

Consistent with CMS1–CMS4 associations, PDS2 tumors were enriched for inflammatory and immune signaling pathways, such as interferon-α and interferon-γ response as well as stromal-related epithelial-to-mesenchymal and transforming growth factor β (TGF-β) activation using the Hallmark collection[4] (Fig. 1g). PDS1 tumors displayed elevated cell-cycle-related pathways, including MYC and E2F targets and G2M checkpoint, whereas there was near universal repression of cancer-associated hallmark signaling in PDS3 (Fig. 1g). Overall, these three distinct transcriptional classes confirm the extent of signaling heterogeneity within *KRAS*mut CRC and suggest that PDS classification provides a novel basis for forward and reverse translation studies.

To ensure that PDS classification could be performed on additional datasets, we explored different classification algorithms, in which the PDS-specific $n$ = 626 gene set signatures (Fig. 1d) were used as features for training and developing the classification model (Fig. 1h). Given the best overall accuracy displayed by the support vector machine via radial basis function (svmRBF) algorithm (Extended Data Fig. 1e,f), the PDS classifier was based on this model. The PDS classification model also includes a prediction probability scoring read-out so users can clearly enumerate intratumoral subtype heterogeneity in individual samples. We defined 0.6 as the default threshold for PDS classification, with tumors that do not reach this threshold termed 'mixed' (Fig. 1i and Extended Data Fig. 1g,h). To ensure that this classifier can be easily used by the wider research community, an R-based classification package was developed: *PDSclassifier* (see Code availability), which accurately classified tumors into the same robust clusters identified during class discovery (Fig. 1h and Extended Data Fig. 1i).

### Genomic and mutational landscapes of PDS1–PDS3 tumors

Although PDS was developed initially within *KRAS*mut tumors, when tested on the entire FOCUS cohort ($n$ = 360) it classified 87% of all samples independent of *KRAS* mutational status, PDS1 = 26%, PDS2 = 31%, PDS3 = 30%, mixed = 13% (Fig. 2a). The same PDS–CMS associations were observed in *KRAS* wild-type (WT) and *KRAS*mut tumors (Fig. 2b); PDS2 were predominantly CMS4 or CMS1, and CMS2 and CMS3 were distributed across PDS1 and PDS3 (Fig. 2c,d). No clinical or pathological differences were identified between PDS subtypes (Extended Data Fig. 2a). In line with CMS associations, there was enrichment of *BRAF* mutations and fewer *APC* mutations in PDS2 than in PDS1 or PDS3 (Fig. 2e and Extended Data Fig. 2b). Despite such distinct biological signaling between PDS1 and PDS3, these tumors had identical mutational profiles across key genes within the WNT, MAPK, PIK3CA, cell cycle or TGF-β pathways (Fig. 2e and Extended Data Fig. 2b). This was

further observed when copy number estimates were assessed (Fig. 2f and Extended Data Fig. 2c,d).

Overall, despite being developed within *KRAS*mut tumors, these PDS identify distinct biology across all CRC tumors, regardless of mutational status.

### Transcriptional characteristics of PDS1–PDS3 tumors

Using the Hallmark and DoRothEA[12] algorithms, we characterized PDS-specific traits in the entire FOCUS cohort (GSE156915; $n$ = 360), alongside CRC samples from two additional independent non-randomized cohorts, including the cohort used in the development of CMS (GSE39582 (ref. 2); $n$ = 566) and another independent cohort from the S:CORT[13,14] program (SPINAL cohort, GSE248381; $n$ = 258). Again, there was a clear transcriptional distinction between each subtype, with the most prominent biological signaling cascades representing cell-cycle-related activity in PDS1 and stromal or inflammatory signaling in PDS2; the only consistent signal elevated in PDS3 was the repression signature 'KRAS signaling down' (Fig. 2g,h and Extended Data Fig. 3a,b). These transcriptional landscapes were underpinned by distinct transcription factor (TF) activity, with cell-cycle-related TFs, such as E2F2, being significantly active in PDS1 (Fig. 2i). PDS2 tumors were significantly activated for $n$ = 32 stromal and inflammatory TFs, including SMAD3, STAT3, IRF1 and ERG. Although repressed for biological pathways within the Hallmark collection in general, PDS3 exhibited activation of TFs relating to a diverse set of hormonal and developmental processes, including FOXA2 and NR2F2 (Fig. 2i).

To test the clinical relevance of PDS classification, we assessed relapse-free survival (RFS) rates, which revealed that PDS2 and PDS3 displayed poor prognosis compared to PDS1 in the GSE39582 cohort (Fig. 2j) regardless of *KRAS* mutational status (Extended Data Fig. 2e). These findings were independently validated in the randomized PETACC-3 trial cohort[15], further reinforcing the poor prognosis of PDS3 compared to PDS1 (Fig. 2j and Extended Data Fig. 2e). When tested in microsatellite stable tumors only, the prognostic relevance of PDS3 remains (Extended Data Fig. 2e). Given that the majority of PDS1 and PDS3 tumors were also classified as CMS2 (Extended Data Fig. 2f), we sub-stratified this otherwise uniform subtype, revealing that CMS2–PDS1 tumors have a significantly better outcome than CMS2–PDS3 tumors (Extended Data Fig. 2g), with PDS3 tumors having as poor an outcome as the stroma-rich CMS4 tumor subtype (Fig. 2j and Extended Data Fig. 2h).

### Intratumoral heterogeneity across PDS

To identify the cellular source of the signaling underpinning PDS, we assessed gene expression scores in micro-dissected tumors (GSE31279), which highlighted that the PDS1 and PDS2 gene sets are primarily derived from epithelial and stromal regions, respectively (Fig. 3a,b), whereas the PDS3 gene sets do not show any significant association between epithelium or stroma. Using hematoxylin and eosin (H&E) slides from FOCUS and SPINAL samples, we set out to identify distinguishing histological features between PDS classes using a methodology similar to the image-based CMS (imCMS) classifier[16,17] (Fig. 3c). PDS1 and PDS2 tumors contain visual features that our models used to discriminate between the two classes (PDS1 vs PDS2 and PDS3: area under the receiver operating characteristic curve (AUROC), 0.740 ± 0.019; F1-score, 0.574; PDS2 vs PDS1–PDS3: AUROC, 0.810 ± 0.033; F1-score, 0.618; Fig. 3d,e). However, our trained models were unable to identify PDS3-specific morphological patterns, as the learned features that were discriminative for PDS1 and PDS2 were also prominent in PDS3-labeled images, resulting in poor cross-validation performances for PDS3 (PDS3 vs PDS1 and PDS2: AUROC, 0.557 ± 0.026; F1-score, 0.338; Fig. 3d,f). The mixed morphology of PDS3 was also supported by the intermediate description of these tumors using ESTIMATE[18] (Fig. 3g and Extended Data Fig. 3b,c), making our models erroneously classify PDS3 images

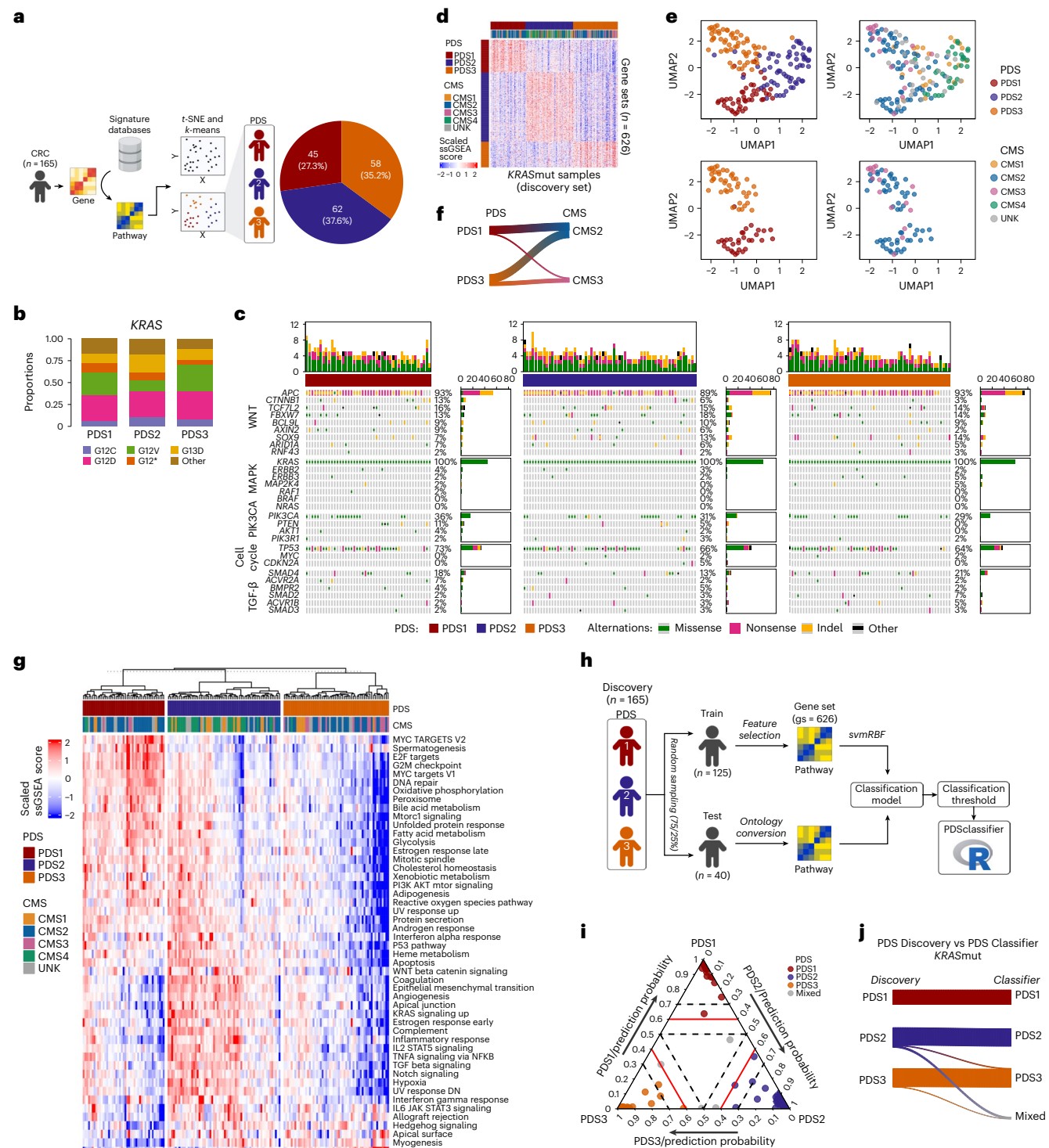

**Fig. 1 | PDS of CRC highlights two subsets of CMS2. a,** Schematic of class discovery: gene expression to pathway matrix using gene signature databases (see Methods) on *KRAS*mut CRC samples (*n* = 165) from the FOCUS cohort (used as the discovery set) defines three PDS following series of dimensionality reduction (*t*-SNE) and unsupervised *k*-means clustering. **b,** Bar chart highlighting the proportion of *KRAS*mut variants across PDS. **c,** Oncoprint with key cancer driver genes across PDS in *KRAS*mut CRC samples. **d,** Heatmap depicting the PDS-specific ssGSEA scores across the discovery set (*n* = 165) with PDS and CMS annotation. **e,** UMAP from the PDS-specific ssGSEA scores on the discovery set with PDS (left) and CMS (right) annotations, using all samples from the discovery set (top) and PDS1 and PDS3, and CMS2 and CMS3 samples only (bottom). **f,** Sankey plot focusing on CMS2 and CMS3 primarily subdivided into PDS1 and

PDS2. **g,** Heatmap visualization of the 'Hallmark' gene sets ssGSEA scores across the discovery set, with annotated PDS discovery calls and CMS. **h,** Schematic of classifier development: the FOCUS discovery set was divided into training (*n* = 125) and test (*n* = 40) sets, and subsequently, the svmRBF classification algorithm was trained on the training set and tested on the test set to finalize performance of the classification model. The classification prediction threshold was determined and the *PDSclassifier* R package was developed. **i,** Ternary plot displaying PDS prediction probabilities using *PDSclassifier* on the FOCUS test subset (*n* = 40). The red line denotes the default PDS prediction threshold at 0.6 and the dashed black line represents the PDS prediction threshold at 0.5 and 0.7. **j,** Sankey plot highlighting the overall concordance of PDS calls between discovery and classifier calls, suggesting robustness of the classifier. UNK, unknown.

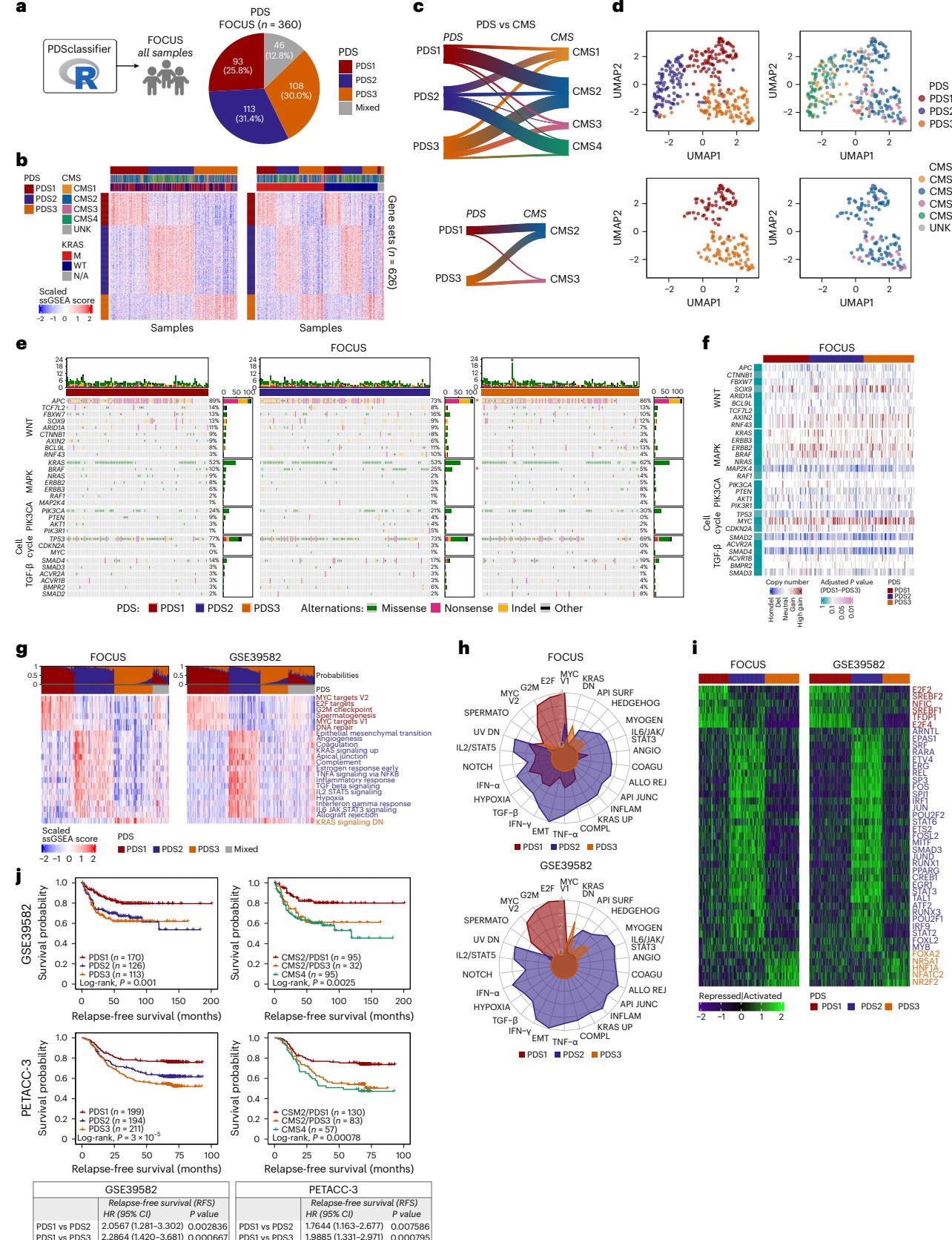

as either PDS1 or PDS2. Even a more stringent 0.8 PDS classification threshold, which identifies transcriptionally more homogeneous PDS tumors and maintains the same PDS prognostic value (Fig. 3h–j), did not improve the image-based classifier.

Overall, these data indicate that although PDS1 and PDS2 tumors are dominated by epithelial and stromal influences, respectively, PDS3 tumors are morphologically indistinguishable from these two classes. However, despite this histological heterogeneity, PDS can be used to

**Fig. 2 | PDS tumors are genomically indifferent but transcriptionally distinct with prognostic value. a**, PDS classification on all FOCUS samples ($n$ = 360) yields ~12.8% of samples with 'mixed' PDS biology that did not reach the threshold of 0.6. **b**, Heatmap depicting the PDS-specific ssGSEA across the FOCUS cohort, recapturing the same pattern regardless of *KRAS* mutational status. The top annotation bar indicates PDS, CMS and *KRAS* mutation. **c**, Sankey plot highlighting the FOCUS cohorts showing CMS–PDS alignment (top) and how CMS2 and CMS3 are subdivided into PDS1 and PDS3 (bottom). **d**, UMAP from the PDS-specific ssGSEA scores on the FOCUS cohort with PDS (left) and CMS (right) annotations using all samples from FOCUS cohort (top) and CMS2 and CMS3 and PDS1 and PDS3 samples only (bottom). **e**, Cancer driver mutations displayed in oncoprint with (red *) *BRAF* enrichment and low *APC* mutation in PDS2. **f**, Copy

number alterations in heatmap indicating no significant difference across PDS. Statistics: Fisher's exact test between PDS1 and PDS3. **g**, Significant 'Hallmark' ssGSEA heatmaps in the FOCUS and GSE39582 cohorts, with top annotation of PDS prediction probabilities and PDS calls. **h**, Radar plots highlight highly upregulated Hallmark gene sets prominent for each PDS in the FOCUS (top) and GSE39582 (bottom) cohorts. **i**, Transcriptional factor activity shown in the heatmaps displays transcriptional factors activated or repressed across PDS. **j**, Kaplan–Meier RFS plots in GSE39582 and PETACC-3 colon cancer cohorts (top). Univariate Cox proportional analysis outcome displayed in a tabular format (bottom), comparing PDS1 with PDS2 and PDS3 with a hazard ratio (HR), 95% confidence intervals (CI) and *P* value.

identify tumors that are transcriptionally homogeneous, particularly when the classification threshold is increased, ensuring that sample classification is based on robust and clearly distinct signaling (Fig. 3l).

### PDS3 tumors are devoid of canonical LGR5[+] and regenerative ANXA1[+] stem traits

The lack of robust morphological patterns in PDS3 tumors prompted us to next assess their stem cell and precursor associations. Using the intestinal stem cell (ISC) index[19], we observed an association between PDS1 tumors and the canonical LGR5[+] crypt-base columnar cell (CBC) signature, PDS2 tumors aligned strongly with LGR5[−]/ANXA1[+] regenerative stem cell (RSC) signature and PDS3 tumors scored low for both stem signatures (Fig. 4a and Extended Data Fig. 4a–d). A subset of cases ($n$ = 20) from the SPINAL cohort was selected for fluorescence in situ hybridization, which further confirmed increased epithelial *LGR5* expression in PDS1 tumors, epithelial *ANXA1* expression in PDS2 tumors and an almost complete absence of staining for both markers in the PDS3 tumors tested (Fig. 4b). Furthermore, PDS1 and PDS2 tumors were associated with tubular and serrated gene signatures, respectively[20], but PDS3 tumors had no clear association with these precursor features. This was confirmed using a cohort of transcriptionally profiled polyp samples (S:CORT polyp cohort[19]) pathologically defined as tubulovillous adenomas, sessile serrated lesions and traditional serrated adenomas, in which tubulovillous adenomas were predominantly PDS1 and sessile serrated lesions associated with PDS2, but PDS3 tumors lacked any precursor association (Fig. 4c and Extended Data Fig. 4b–d). Additionally, Ki67[+] immunohistochemistry and transcriptional signatures of proliferation[21] and replication stress[22] further highlighted these features in PDS1, with significantly lower signaling in PDS3 tumors (Fig. 4d–f).

Overall, these results indicate the presence of numerous well-characterized cancer-related features associated with both PDS1 and PDS2 tumors; however, these approaches provided limited insights into the biology underpinning the ~30% of CRC tumors classified as PDS3.

To complement the non-exhaustive PDS characterizations presented here and to support the FAIR principles[23], we developed the 'SubtypeExploreR' platform. This enables any user to interrogate

transcriptional genes and/or signatures, including existing signatures from numerous databases or an unlimited combination of de novo unpublished classifiers, according to PDS and other CRC subtypes in our bulk cohorts (Fig. 4g; https://subtypeexplorer.qub.ac.uk).

This resource will ensure that these data are not just accessible, but (re)usable to a much wider audience.

### Intrinsic stem-to-differentiated axis aligns with PDS1 and PDS3 tumors

To uncover biological interactions that underpin PDS3, genes within the PDS3-specific gene sets ($n$ = 961) were assessed using Enrichr[24] and STRING[25] (Extended Data Fig. 5a,b), revealing an association with the polycomb repressive complex (PRC), which has an established role in repressing PRC targets that regulate cellular differentiation. Assessment of the gene expression of key PRC genes, such as *EZH2*, revealed that they were significantly lower in PDS3 tumors (Extended Data Fig. 5c), resulting in enrichment of PRC targets genes[26] in PDS3 tumors compared to PDS1 and PDS2 tumors (Fig. 5a and Extended Data Fig. 5e). On the contrary, MYC targets, known for their role in maintaining a stem-like or pluripotent state[26], were enriched in PDS1 and PDS2 compared to in PDS3 tumors (Fig. 5b and Extended Data Fig. 5f). We found a highly significant negative correlation between individual tumor scores in these bulk tumor tissue along a MYC–PRC axis, with PDS1 tumors displaying high-MYC–low-PRC target expression and PDS3 tumor displaying low-MYC–high-PRC target expression, and PDS2 tumors being intermediate (Fig. 5c and Extended Data Fig. 5g). In combination with our data indicating lower levels of cell cycle and transcriptional activity, and an absence of canonical and RSCs (Figs. 3 and 4), these data suggest that the cellular epithelial states that dominate PDS3 tumors may have shifted towards a greater proportion of differentiated lineages than in PDS1 and PDS2 tumors.

To test this premise, we analyzed transcriptional data from an independent cohort of mouse colonic epithelial cells across six differentiation states (GSE143915 (ref. 27); Fig. 5d), which revealed a significant association between Myc-PRC target gene expression levels and cellular or biological (as opposed to tumor grade or histological) differentiation status. Stem and progenitor cells displayed

**Fig. 3 | PDS intratumoral heterogeneity exits with PDS3 displaying no defined histological features. a**, Heatmap visualization of the ssGSEA score using PDS-specific gene in laser capture micro-dissected CRC dataset with annotated epithelium or stroma region per sample. **b**, Comparison of PDS-specific gene set ssGSEA score between epithelium ($n$ = 8) and stroma ($n$ = 10), using two-sided Wilcoxon rank-sum test. Boxplots depict the median and interquartile range, with whiskers extending to the minimum and maximum values (excluding outliers as dots). **c**, Workflow schematic on development of the image-based PDS classifier (*imPDSclassifier*). **d**, Confusion matrix displaying the PDS prediction based on the *imPDSclassifier* to the transcriptomic-based PDS calls. **e,f**, imPDS predictions on the digital whole slide H&E images with tile-level confidence probability on PDS1, PDS2 (**e**) and PDS3 (**f**) samples. Scale bars, 2 mm. **g**, Stromal and immune ESTIMATE score across PDS in the FOCUS

(PDS1, $n$ = 93; PDS2, $n$ = 113; PDS3, $n$ = 108) and SPINAL (PDS1, $n$ = 80; PDS2, $n$ = 54; PDS3, $n$ = 82) cohorts. Boxplots inside the violin plots depict the median and interquartile range, with whiskers extending to the minimum and maximum values (excluding outliers as dots). *P* values: two-sided Wilcoxon rank-sum test. **h**, Ternary plots represent the utility of PDS prediction probability thresholds, whereby increasing the threshold from 0.6 (left) to stringent 0.8 (right) can lead to more homogeneous PDS tumor samples while **i,j**, still retaining its prognostic value (PDS1, $n$ = 109; PDS2, $n$ = 83; PDS3, $n$ = 80). Error bars, 95% confidence intervals. **k**, PDS prediction probability stacked bars and **l**, density plots highlighting the level of PDS heterogeneity found in each sample, with dashed lines representing the default threshold of 0.6 (top); increasing threshold, for example, to 0.8 (bottom), makes it more transcriptionally homogeneous.

highly proliferative and PDS1-like (high-Myc–low-PRC target) gene expression, whereas more differentiated enterocyte and enteroendocrine cell populations displayed low proliferation and PDS3-like (high-PRC–low-Myc target) gene expression (Fig. 5e). These data again

highlight the strong association between cellular stem–differentiation and the MYC–PRC axis, with the phenotypic traits associated with the PDS classification system in bulk tumor data (Fig. 5f and Extended Data Fig. 5d,h).

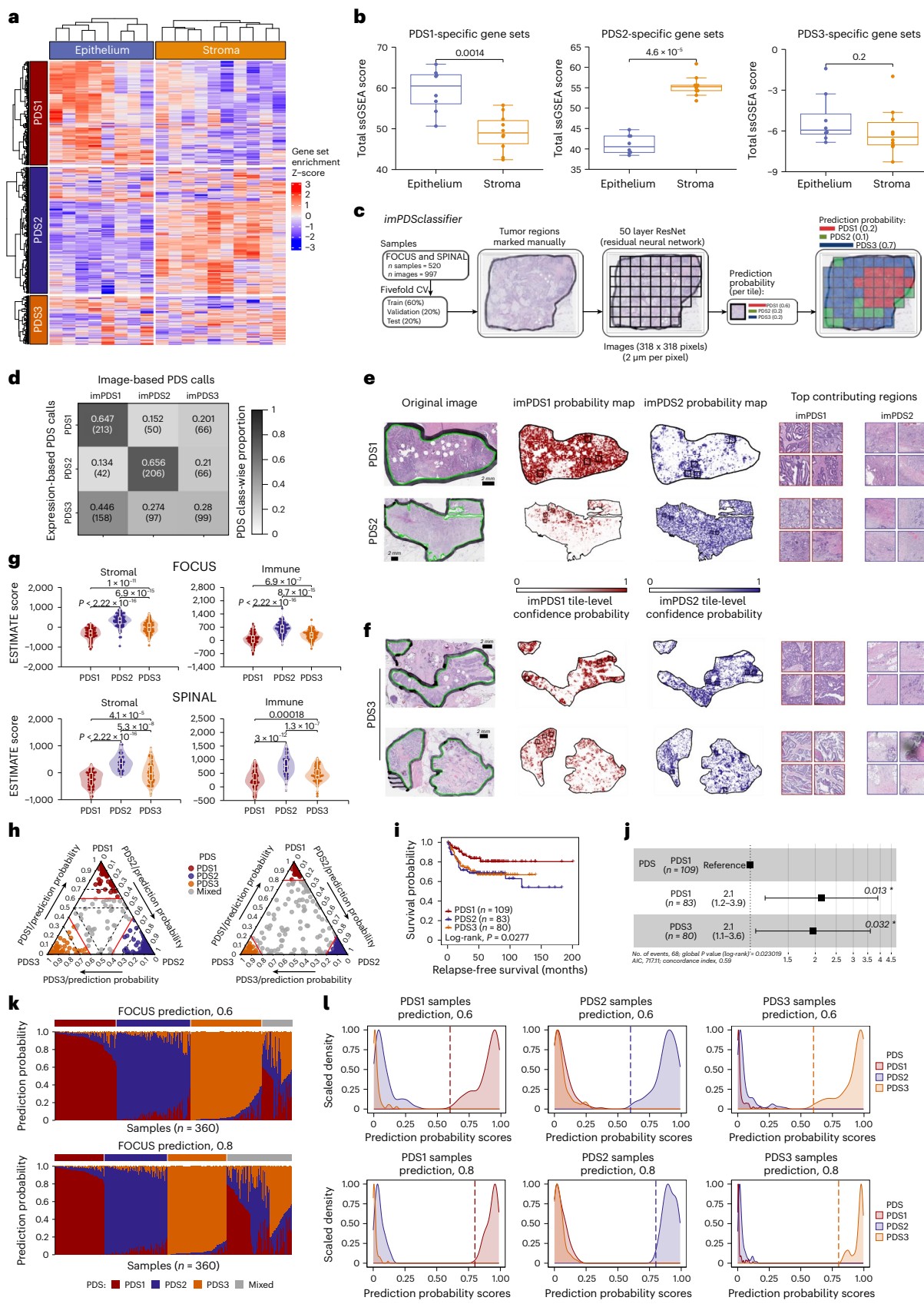

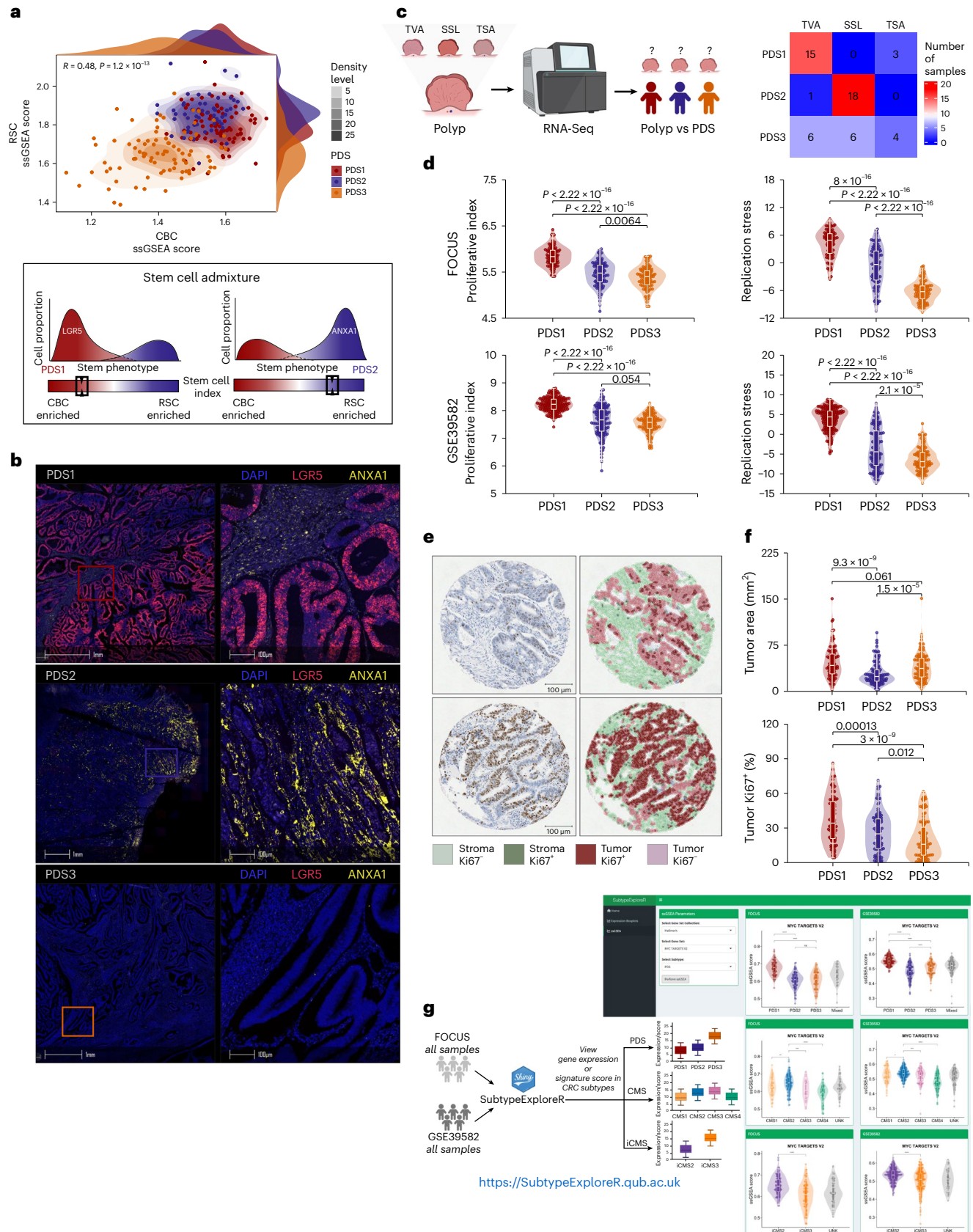

To provide more granularity, we used scRNA-seq data (*n* = 29,452 epithelial cells) generated from murine CRC organoids mono-cultured or co-cultured with fibroblasts and/or macrophages with existing lineage annotations[28] (Fig. 5g). We defined MYC targets-high and PRC targets-high cell populations based on the MYC and PRC targets ssGSEA score, representing PDS1-like and PDS3-like cells, respectively. Notably, we observed that MYC targets-high cells strongly aligned with the annotated canonical stem cell populations[28] (colonic stem cells

**Fig. 4 | PDS3 tumors indicate no stem-like and precursor lesion associations.** **a**, Scatter gradient-density plot representing LGR5+ CBC and LGR5−/ANXA1+ RSC ssGSEA scores with PDS sample annotations in the SPINAL cohort (top). Stem cell plasticity landscape schematic highlighting the PDS1 association with CBC and the PDS2 association with RSC gene signatures along the stem cell index proposed in a previous study[19] (bottom). **b**, In situ hybridization multiplex staining with LGR5, ANXA1 and DAPI across PDS samples in the SPINAL (PDS1, n = 7; PDS2, n = 7; PDS3, n = 6) cohort. Scale bars, 1 mm and 100 µm. **c**, Schematic representing PDS classification of the colorectal polyp dataset with categorized polyp types (left); confusion matrix depicting the association between PDS calls and polyps (right). **d**, Proliferation index and replication stress measure across PDS in the FOCUS (PDS1, n = 93; PDS2, n = 113; PDS3, n = 108) and

GSE39582 (PDS1, n = 186; PDS2, n = 140; PDS3, n = 122) cohorts. **e**, Representative immunohistochemistry images of Ki67+ staining. Scale bars, 50 µm. **f**, Violin plot showing the tumor area (mm²) determined from H&E digital histological scores using HALO (PDS1, n = 91; PDS2, n = 111; PDS3, n = 107) and the per cent of tumor Ki67+ per sample (represented in **e**) across PDS in the FOCUS (PDS1, n = 91; PDS2, n = 108; PDS3, n = 105) cohort. Boxplots inside the violin plots (in **d** and **f**) depict the interquartile range, median, minimum and maximum values (excluding outliers as dots). P values: two-sided Wilcoxon rank-sum test. **g**, ShinyApp platform (SubtypeExploreR), integrated with the FOCUS and GSE39582 cohorts allows interrogation into gene expressions and signatures across three different CRC subtypes, including PDS, CMS and iCMS. TVA, tubulovillous adenomas; SSL, sessile serrated lesions; TSA, traditional serrated adenomas.

and hyper-proliferative colonic stem cells) and, to a lesser extent, the regenerative or revival stem cells, in a gradient of gene expression along the cell fate trajectory towards PRC targets-high cells, which were closely associated with differentiated cell populations, including transit-amplifying, goblet and late enterocyte cells (Fig. 5h,i and Extended Data Fig. 6c). Given the absence of stem populations in PDS3 and an association of MYC–PRC with stem–differentiation, we proposed a PDS-related stem–differentiation single sample scoring system for use in bulk and single-cell data, termed the 'stem maturation index' (SMI; Fig. 5j). Evaluation of the SMI in single-cell and bulk tissue datasets confirmed a clear correlation with PRC targets scores, differentiated colonic cells and PDS3 tumors (Fig. 5k,l and Extended Data Fig. 5i,j). Additionally, expression levels of several differentiation marker genes were also elevated in PDS3 and positively correlated with the SMI, further confirming PDS association along the phenotypic landscape of stem-to-differentiation, with PDS3 tumors having reduced stem populations and an increased abundance of epithelial cells that are further along the differentiated trajectory (Fig. 5m, Extended Data Figs. 5k,l and 6a,b).

Taken together, although the use of CMS and stem cell signatures can enumerate the canonical versus regenerative stem-like state of a tumor, our data supports that the use of these existing approaches in conjunction with PDS and SMI classification provides an otherwise unseen view of the overarching stem–differentiation cellular landscape and information about the overall differentiation state (distinct from the tumor grade) of a bulk tumor or a single cell (Fig. 5n).

### Pre-clinical models fail to recapitulate PDS3 tumor biology

Using equivalent mouse pathways for PDS classification and bulk tumor transcriptional data from n = 51 genetically engineered mouse models (GEMMs) across n = 6 genotypes[29]—Apc^fl/+ (A); Apc^fl/+ Kras^G12/+ (AK); Braf^V600E/+ Trp53^fl/fl (BP) and Braf^V600E/+ Trp53^fl/fl Notch1^Tg/+ (BPN), Kras^G12D/+ Trp53^fl/fl Notch1^Tg/+ (KPN); Kras^G12D/+ Trp53^fl/fl (KP)—we assessed how well mouse models align to human PDS (Fig. 6a). The A and AK models were exclusively PDS1, whereas the KP, KPN and BPN models were divided into PDS2 and PDS3 (n = 1 BPN as PDS1), with BP models aligning with PDS3 (Fig. 6a and Extended Data Fig. 6d). There was a clear alignment of the biological hallmarks, TF activation states and CMS classifications associated with PDS1 and PDS2 between human and mouse tumors (Fig. 6b,c). However, in contrast to human PDS3 biology, GEMM tumors classified as PDS3 were not associated with CMS2 or the transcriptional repression for hallmarks signaling observed in human tumors and displayed signaling similar to GEMMs classified as a 'mixed' subtype (Fig. 6b,c), indicating that the mouse models used did not accurately represent human PDS3 biology (Fig. 2). Assessment of proliferation index and replication stress according to both PDS classification and genotype demonstrated elevation of both phenotypes in PDS1, but no further suppression was observed in PDS3 compared to PDS2 (Fig. 6d and Extended Data Fig. 6e). Finally, although elevation for the Myc–PRC targets signaling continuum is maintained in PDS1 GEMMs, there was again little distinction between PDS2 and PDS3 (Fig. 6e).

In line with the murine tumor tissue above, AK organoids remain strongly associated with a PDS1-like stem-enrichment (Fig. 6f) and also display a high entropy and differentiation potential when viewed using a Waddington-like landscape (Fig. 6g)[30]. By contrast, in the absence of mutations and culture media supplements (WNT3A, EGF, Noggin and R-spondin-1; WENR), WT murine organoids display a normal-like homeostatic distribution of stem-to-differentiated cell populations (Fig. 6f) and reduced differentiation potential (Fig. 6g); a pattern that most closely reflects PDS3-like lineage distributions. Importantly, the lack of PDS3 pre-clinical models may be a result of pre-clinical in vitro and/or ex vivo culturing conditions, as organoid media containing these growth factors drives a strong selective pressure towards an AK-like stem-enriched and highly proliferative cellular hierarchy, even in WT models (Fig. 6h–k and Extended Data Fig. 6f–h). Although WT organoids are most closely aligned to PDS3 stem-differentiation patterns, it must be clearly stated that such models do not capture the full extent of overarching PDS3 tumor biology.

These findings indicate a critical limitation for pre-clinical modeling of the worst prognostic group of CRC tumors, restricting opportunities for improving our mechanistic understanding and testing of PDS3-specific therapeutic options.

### Assessment of PDS-specific phenotypes from bulk to single-cell

We next leveraged the human scRNA-seq data used to develop iCMS, n = 49,155 epithelial cells from n = 63 CRC tumors[6]. In line with bulk tumors and epithelial-specific intestinal lineages, these analyses revealed that neoplastic cells states can be classified along an MYC–PRC target axis, in individual patient clusters (Fig. 7a–c and Extended Data Fig. 7a,b) or when clustered according to the pathway-level gene sets used for PDS development (Fig. 7d). Cell cycle annotation[31] revealed that PDS1-like neoplastic cells displaying higher MYC target signaling were more likely to be in S or G2M phase, whereas PDS3-like cells elevated for PRC targets expression were more likely to be in G1 phase (Fig. 7e). These results are more pronounced when assessed in cells at either end of the MYC–PRC continuum, which are most representative of PDS1 and PDS3 biology (Fig. 7c and Extended Data Fig. 7c,d). Additional characterization further confirms faithful alignment between bulk and scRNA-seq, whereby phenotypes of elevated proliferation and replication stress are observed in MYC targets-high PDS1-like cells alongside elevation of the KRAS signal repression signature (KRAS signal DN) in PRC targets-high PDS3-like cells (Fig. 7f,g and Extended Data Fig. 8e). Moreover, CBC stem cells are aligned with MYC targets-high, whereas RSC coincides more closely between the boundary of MYC targets-high and PRC targets-high cells, highlighting the features of PDS2-like intrinsic traits (Fig. 7h,i and Extended Data Fig. 7f). These findings suggest that biology identified by PDS1–PDS3 tumors in bulk tissue strongly correlates with the intrinsic biology defined by MYC–PRC target gene signatures at a single-cell level.

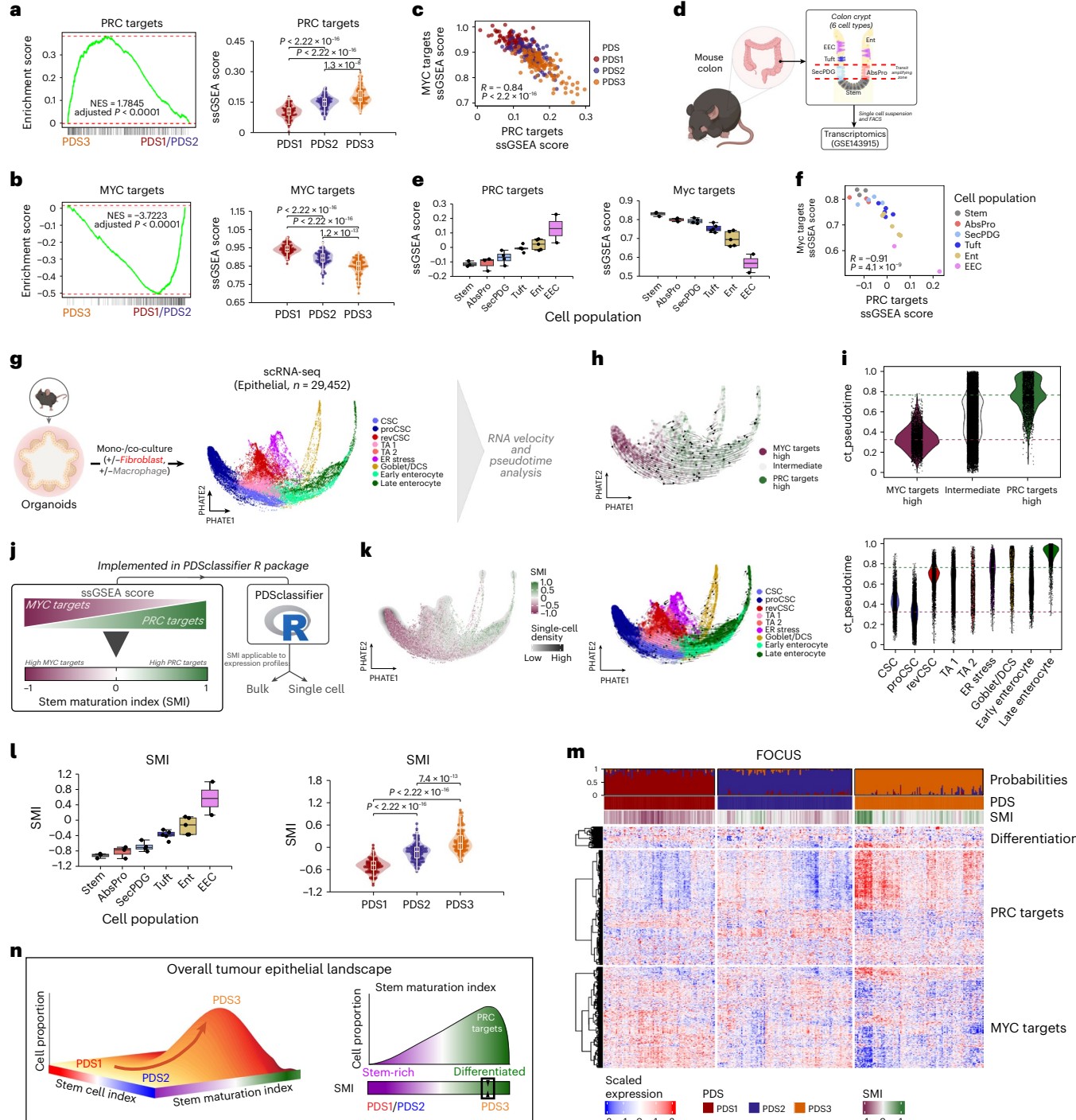

**Fig. 5 | MYC–PRC targets biological axis associated with stem–differentiation cellular dynamics. a**, GSEA enrichment and violin plot displaying ssGSEA scores compared across PDS in the FOCUS cohort, using PRC targets gene signature. **b**, as in **a**, but with MYC targets gene signature (PDS1, $n = 93$; PDS2, $n = 113$; PDS3, $n = 108$). Boxplots inside the violin plots depict the interquartile range, median, minimum and maximum value (excluding outliers as dots). $P$ values (violin plots): two-sided Wilcoxon rank-sum test. GSEA: Benjamini–Hochberg adjusted $P$ value. NES, normalized enrichment score. **c**, Scatterplot depicting the inverse correlation between MYC targets and PRC targets ssGSEA scores, with annotated PDS samples. $P$ value: two-sided Pearson correlation coefficient. **d**, Schematic of mouse intestinal crypt epithelium transcriptional dataset (GSE143915 (ref. 27)). **e**, PRC targets and Myc targets ssGSEA scores across colon epithelial cells. **f**, as in **c**, but with colon epithelial cell types. **g**, Schematic of murine organoid-derived epithelial single-cell dataset[28] used for RNA velocity and pseudotime trajectory analysis. **h**, PHATE visualizations with cells annotated as MYC targets-high and PRC

targets-high (top) and epithelial cell types (bottom). Arrows: RNA velocity based on CellRank[39] method. **i**, Pseudotime score comparisons across groups (top) and cell types (bottom). The red and green dash lines represent the median of MYC and PRC targets ssGSEA scores, respectively. **j**, Schematic of SMI, made available in the *PDSclassifier* R package for both bulk and single-cell data. **k**, PHATE visualization with SMI. **l**, Comparison of SMI across epithelial cell types in GSE143915 and PDS in the FOCUS cohort ($n$ as in **b**). $P$ values: two-sided Wilcoxon rank-sum test. **m**, Heatmap visualization of the differentiation, PRC targets and MYC targets gene expressions in the FOCUS cohort, with annotated PDS prediction probabilities, PDS and SMI. **n**, In addition to the stem phenotypic landscape described in a previous work[19], an overall tumor epithelial landscape describes the stem–differentiation cellular dynamics along the SMI, in which PDS3 tumors are highly enriched for differentiated-like traits that correspond to high PRC target expression. AbsPro, absorptive progenitor; SecPDG, secretory progenitor/deep crypt secretory cells/goblet, Ent, enterocytes; EEC, enteroendocrine cell.

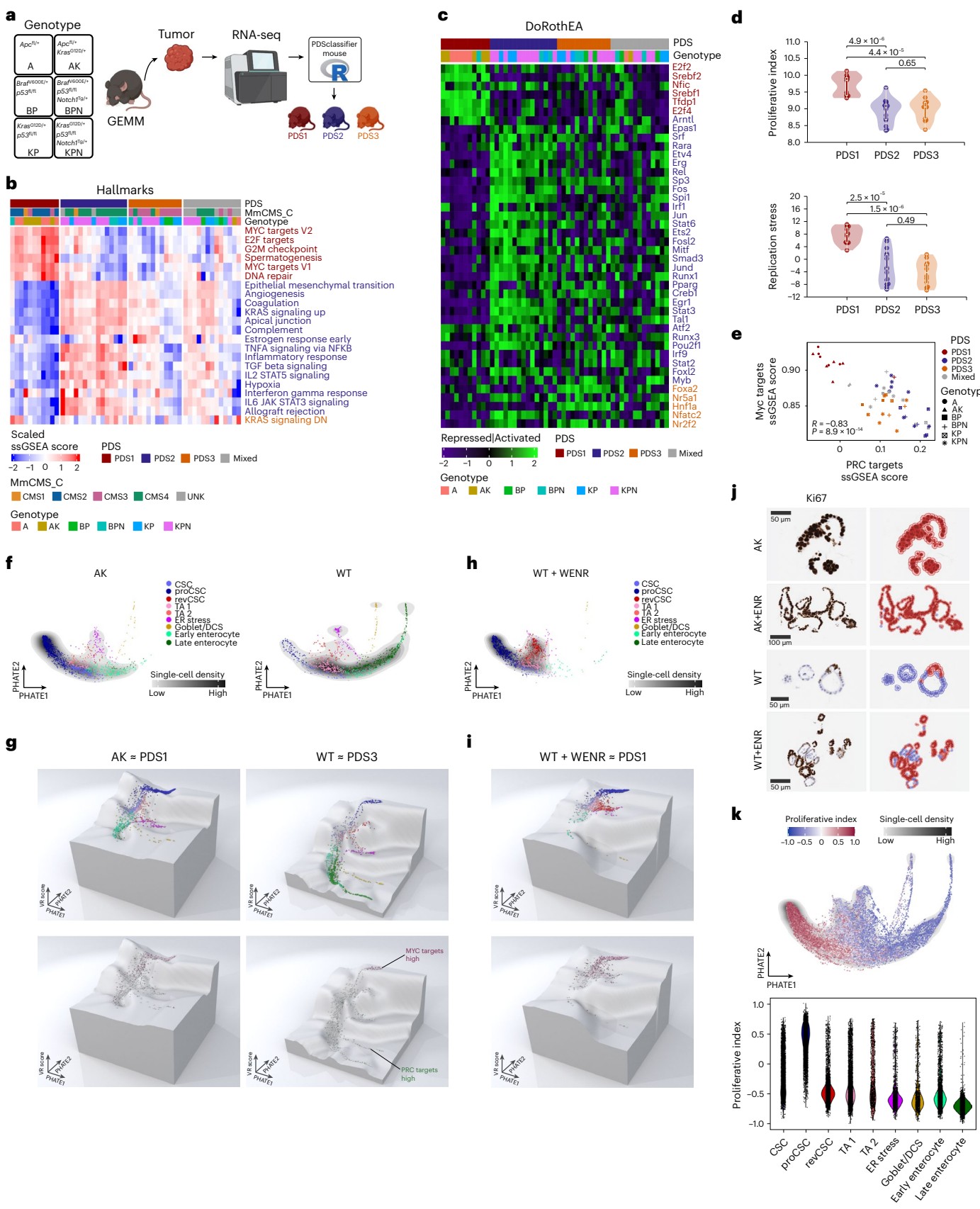

## Cellular differentiation trajectory status in colorectal tumors

We next assessed the RNA velocity and pseudotime trajectories (Fig. 8a). Pseudotime analysis revealed that the cell fate trajectories in each patient cluster were highly correlated with the PDS1–PDS3 axis of MYC targets–PRC targets or SMI (Fig. 8b and Extended Data Fig. 8a,b). Moreover, RNA velocity further validated that the directionality of

**Fig. 6 | PDS3 closely resembles WT normal differentiation patterns and lacks pre-clinical models. a**, Six genotypic GEMM primary tumors were sequenced and PDS were called. **b**, Heatmap displaying the ssGSEA scores for the Hallmark gene sets in GEMMs across PDS, including 'mixed' samples. Top annotation bar indicates PDS calls, mouse CMS (*MmCMS*) and genotypes. **c**, Heatmap displays TF activity from DoRothEA[12]. **d**, Violin plots showing proliferative index (top) and replication stress (bottom) across PDS in GEMMs (PDS1, *n* = 11; PDS2, *n* = 15; PDS3, *n* = 12). Boxes inside the violin plots depict the interquartile range, median, minimum and maximum values. *P* values: two-sided Wilcoxon rank-sum test. **e**, Scatterplot highlighting the inverse correlation between Myc targets and PRC targets ssGSEA scores recapitulated in GEMMs, with PDS annotation. *P* value: two-sided Pearson correlation coefficient. **f**, PHATE visualization on murine organoid scRNA-seq data with cell density and cell type annotations for AK (left) and WT (right). **g**, Waddington landscape representing AK and WT models annotated with cell types (top) or MYC–PRC target-high groups (bottom), closely resembling PDS1 stem-like vs PDS3 differentiated-like traits, respectively. **h**, PHATE visualization for WT cultured in WENR (W, WNT3A; E, EGF; N, Noggin; R, R-spondin-1) in which cellular states are skewed towards stem cell enrichment with limited differentiation. **i**, Waddington landscape of WT + WENR annotated with cell types (top) and MYC–PRC target-high groups (bottom), closely resembling PDS1 stem enrichment. **j**, Representative Ki67+ ISH images of murine organoids (AK, AK + ENR, WT, WT + ENR; *n* = 2 replicates each per group). Scale bars: AK, 50 μm; AK + ENR, 100 μm; WT, 50 μm; WT + ENR, 50 μm. **k**, PHATE visualization for proliferation index scores across all populations.

eventual cell fate transcriptional changes—from MYC targets-high to PRC targets-high cell populations—was present in each patient (Fig. 8c–e). As the original iCMS classifier provides patient-level classification, it provides limited information on the strength and/or extent of iCMS-related biological signaling in individual cells. Therefore, it is unsurprising that iCMS is not strongly aligned with the fundamental cell fate trajectory associated with the stem–differentiation dynamics that are found in each patient (Fig. 8e–h and Extended Data Fig. 8c,d). To provide more clarity into iCMS at the single-cell level, we applied single sample iCMS2–iCMS3 classifier scores from the top tertile of each group, which clearly revealed that the cells least aligned with either iCMS2 or iCMS3 are those that are most aligned with PDS3, SMI and PRC targets-high cells (Fig. 8i and Extended Data Fig. 8e).

Collectively, our findings uncovered a phenotypically subtle subset of colorectal tumors characterized by a lack of numerous cancer-related features, many of which are viewed as essential hallmarks for tumor growth. Overall, the use of PDS, in conjunction with existing CRC classifications, provides a more comprehensive overview of both the dominant and nuanced biology underpinning CRC. It is only through the assessment of all these phenotypic traits—not alone but in combination—that the field can advance in its goals to understand and therapeutically target the mechanisms underpinning CRC (Fig. 8j).

## Discussion

Numerous studies have used individual gene-level data for molecular subtyping class discovery[3,5]. However, in this study, we leverage pathway-level data for class discovery, using signatures that mechanistically underpin important cancer-relevant functions to provide a more direct link with cancer phenotypes in a way that transcends current transcriptional subtypes. This approach identified three PDS, PDS1–PDS3, which merged the inflammatory/stromal CMS1 and CMS4 tumors as PDS2 and sub-stratified epithelial-rich CMS2 and CMS3 tumors into two distinct PDS: PDS1 and PDS3. Biological and molecular characterization reveal that PDS3 lesions display lower cell cycle and stress response, reduced proliferation and stem cell populations, alongside increased cellular differentiation compared to PDS1 and PDS2 tumors. Given the repression for many cancer-related biological phenotypic features that dominate PDS3, these lesions display the worst RFS rates in multiple adjuvant cohorts, including the randomized PETACC-3 clinical trial[15] (summarized in Fig. 8j). Our data also clearly demonstrate how substantial phenotypic and clinical heterogeneity persist within genetically identical tumors. The PDS classifier can be used to stratify tumor samples in a genotype-agnostic manner across numerous independent cohorts, providing an additional informative link between gene expression data and clinical or biological phenotypes.

A recent scRNA-seq study with tumor epithelial cells used gene-level data for class discovery, resulting in sub-stratification based on intrinsic biology (iCMS2 and iCMS3). However, iCMS did not identify different subtypes within epithelial-rich CMS2 tumors[6]. In the data presented here, we clearly demonstrate that by using pathway-level data for class discovery in bulk data rather than a gene-level method in single-cell neoplastic lineages, CMS2 tumors can be robustly and reproducibly sub-stratified into clearly distinct prognostic groups driven by subtle biological signaling that is not evident in iCMS or CMS alone. Importantly, these traits are evident even in the bulk and scRNA-seq transcriptomic datasets used for the development of the CMS and iCMS classifiers. As such, the biological pathway activation approach that underpins PDS classification provides a novel basis for forward and reverse translation studies in combination with these existing subtyping approaches.

The poor outcomes associated with PDS3 tumors in the adjuvant setting require further investigation, as it is possible that the repressed and differentiated patterns that dominate PDS3 bulk tumor profiles are concealing small subsets of aggressive and/or chemotherapy-resistant subclones, similar to mesenchymal subpopulations recently described in melanoma[32], or stem populations not characterized by our CBC and RSC investigations[33]. The poor outcomes associated with PDS3 align with a recently reported role for PRC components in determining how well tumors and tumor cells survive during sustained environmental stress; whereby cells that are deficient in epigenetic regulators, particularly EZH2, have a superior fitness advantage over EZH2-proficient tumors[34]. These stress-resistant subpopulations were characterized by low cycling 'transcriptional numbness', similar to the characteristics of PDS3 tumors presented here. Finally, as our study focused on molecular information, there may be a series of unaccounted-for epidemiological, microbial or viral factors that underpin both the biological traits of the PDS classes and their associated clinical outcomes.

The histological features and molecular profiles in PDS3 lesions were indistinguishable from the other classes, yet these tumors have clearly distinct biological and transcriptional states. The apparent depletion of canonical or RSCs in PDS3 tumors, alongside their slow-cycling nature and elevated proportion of mature and apparently differentiated lineages sits in contrast with the definitive pathological adenocarcinoma classification and CMS2-dominant nature of these tumors. This demonstrates a clear disconnect between biological or lineage-related differentiation and pathological or grade-related differentiation and suggests the presence of additional stem-like populations that underpin tumor viability and maintenance[33,35,36]. Given the distinct activation states across a series of cancer-relevant signaling cascades and phenotypic features, future work is now needed to detangle the histological and biological mechanisms underpinning these lesions.

Data presented here demonstrate that PDS1 and PDS2 tumors align well with a range of available GEMMs and organoids currently used in cancer research, whereas there appears to be no GEMMs or organoids that faithfully align to the biological signaling that characterize PDS3 in human tumors; a point that currently limits more comprehensive mechanistic studies. The lack of accurate PDS3 models is intriguing but perhaps not surprising based on how current systems are developed. Given the relatively short lifespan of mice, there is a bias toward the development of fast-growing, stem-rich lesions that thrive following an

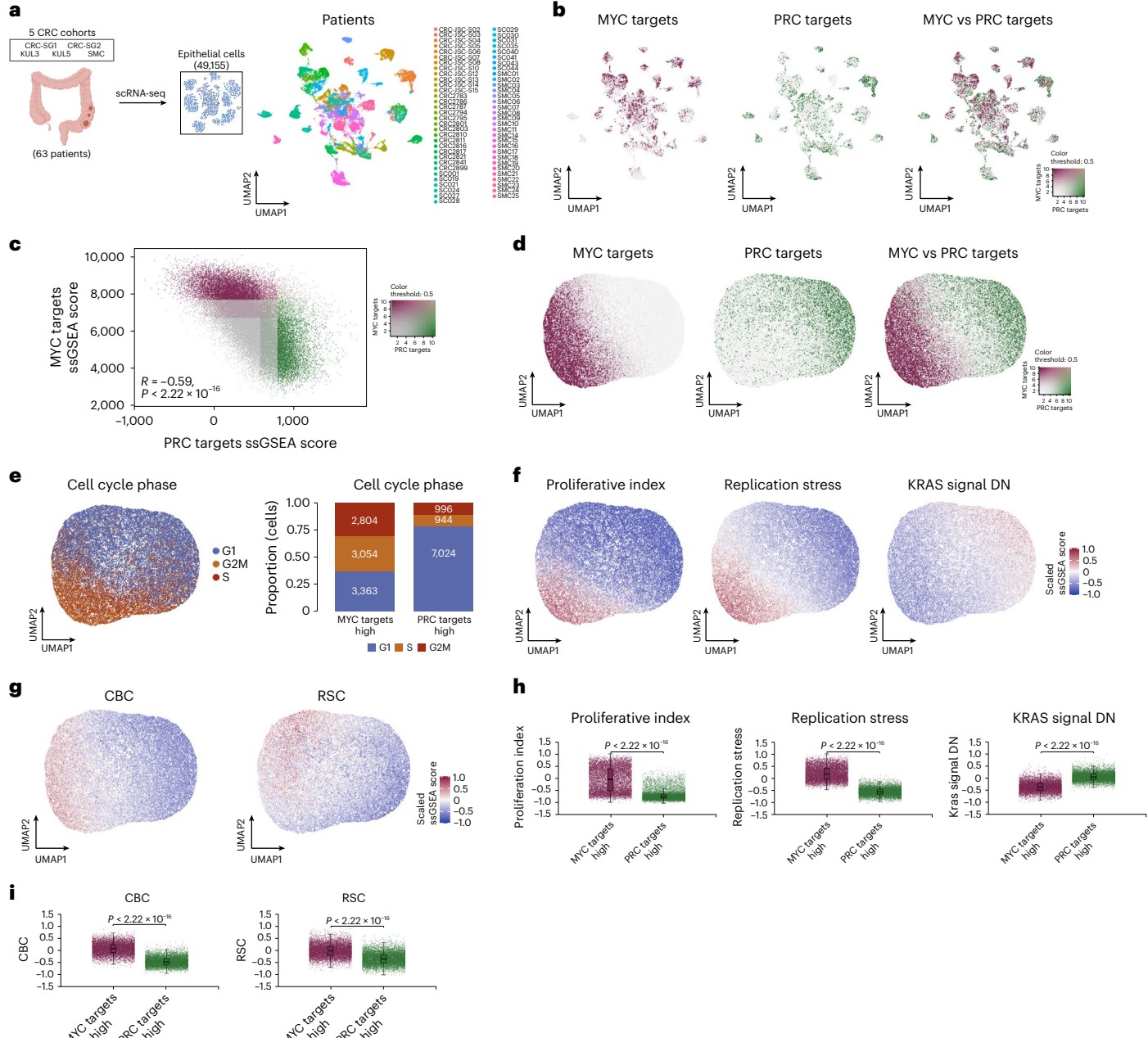

**Fig. 7 | Single-cell analysis of CRC confirms MYC–PRC targets axis. a,**
Epithelial scRNA-seq dataset from 63 patients with CRC over five cohorts; UMAP
displaying cell clustering based on patients. **b,** UMAP visualization indicating
cells annotated with MYC targets ssGSEA score (left), PRC targets ssGSEA score
(center) and a blend of the two (right). **c,** Scatterplot highlighting an inverse
correlation between MYC targets and PRC targets at the single-cell level (color
key as in **b**). **d,** UMAP visualization using PDS gene set scores with cells annotated
with MYC targets (left), PRC targets (center) and a blend of the two (right). **e,**
UMAP visualization annotated with cell cycle phases per cell (left) and stacked

bar chart display of the proportion and number of cells at different G1, S or
G2M cell cycle phases. **f,** UMAP visualization annotated with cells indicating
proliferative index, replicative stress and KRAS signal DN. **g,** Boxplots displaying
a comparison between MYC targets-high ($n$ = 9,221) and PRC targets-high
($n$ = 8,964) cells for proliferative index, replicative stress and KRAS signal DN.
Boxplots depict the interquartile range, median, minimum and maximum values
(excluding outliers as dots). $P$ values: two-sided Wilcoxon rank-sum test. **h,** UMAP
visualization annotated with cells indicating CBC and RSC stem gene signatures.
**i,** Boxplots as in **g** for CBC and RSC.

'all at once' induction of driver genes[37]. This scenario does not replicate
the sequential accumulation of genetic alterations and environmental
changes seen in human tumor development, which can occur over a
15+ year period[38], nor the highly differentiated nature of PDS3 tumors.
Furthermore, even when slow-cycling WT organoids are grown in
enriched media of growth factor ligands, these additives rapidly drive
models towards the PDS1 and PDS2 high-entropy biological traits rather
than retaining the more differentiated PDS3 tumor biology. As such, if
PDS3 biology is to be modeled in a way that more faithfully aligns with

human tumors, methodologies underpinning model development
require urgent attention.

In summary, our study presents an approach to tumor classifica-
tion that relies on patterns observable within broad biological signaling
rather than individual gene clustering. Although it would be easy to
view independent subtyping approaches as competing, such as CMS,
iCMS, CRIS and PDS, it will only be through the classification of samples
using these complementary approaches in parallel that we can reveal
the biological granularity and a greater mechanistic understanding

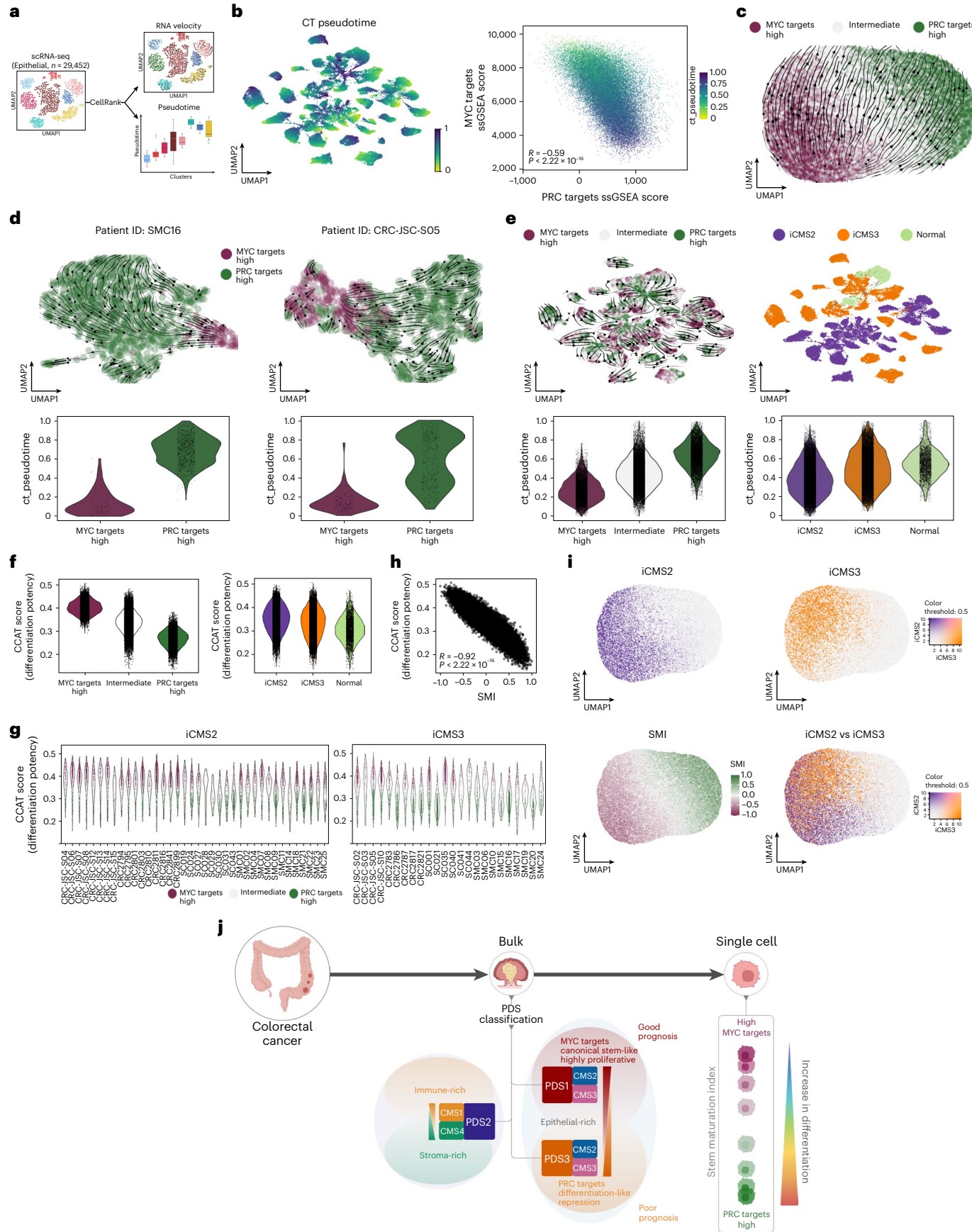

that would otherwise be missed by using them in isolation. Maximizing the amount of phenotypic information that we derived from tumor data, using classification tools that provide synergistic insights into

different yet equally important transcriptional signaling, will provide the most comprehensive landscape for the stratification of tumors into discrete biological groups.

**Fig. 8 | PDS3 differentiation-like trait relates to tumor biology unexplained by single-cell-derived iCMS. a**, RNA velocity and pseudotime analysis using the CellRank[39] method, applied to the CRC epithelial scRNA-seq data. **b**, UMAP visualization displays patient-based clusters (left) and scatterplot between MYC targets and PRC targets ssGSEA scores (right) with cells annotated with pseudotime. *P* value: two-sided Pearson correlation coefficient. **c**, UMAP visualization of PDS gene set scores with MYC and PRC targets-high cell annotations and arrows indicating RNA velocity using the CellRank method. **d**, UMAP visualizations (top) of data from two patients with CRC, with MYC and PRC targets-high cell annotations and RNA velocity displayed with arrows. Violin plots (bottom) display pseudotime analysis for the respective CRC patients. **e**, UMAP visualizations (top) of data from all patients, annotated with MYC and PRC targets-high (left) or iCMS calls (right), and violin plots (bottom) displaying

pseudotime analysis for the MYC–PRC groups and iCMS. **f**, Violin plots display differentiation potency using correlation of connectome and transcriptome (CCAT) comparing MYC–PRC targets-high (left) and iCMS (right). **g**, Violin plots for CCAT measure per iCMS2 (left) or iCMS3 (right) patients. **h**, Scatterplot displaying the association between CCAT and SMI. **i**, UMAP visualization on PDS gene set scores, cell annotation with iCMS2 (top-left) and image-based CMS3 (top-right) ssGSEA scores, a blend of the two (bottom-right) and SMI (bottom-left). **j**, Summarized schematic describing the CRC bulk tumor classified into PDS, outlining the biological and clinical features of PDS tumors, in which MYC targets in association with PDS1 and PRC targets in association with PDS3 led to the identification of an SMI that enumerates stem–differentiation shifts in cellular state.

## Online content

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

Sudhir B. Malla [1], Ryan M. Byrne [1,14], Maxime W. Lafarge [2,14], Shania M. Corry[1], Natalie C. Fisher [1], Petros K. Tsantoulis [3], Megan L. Mills[4], Rachel A. Ridgway [4], Tamsin R. M. Lannagan[4], Arafath K. Najumudeen[4], Kathryn L. Gilroy [4], Raheleh Amirkhah [1], Sarah L. Maguire[1], Eoghan J. Mulholland [5], Hayley L. Belnoue-Davis [5], Elena Grassi [6,7], Marco Viviani [6,7], Emily Rogan[1], Keara L. Redmond[1], Svetlana Sakhnevych[1], Aoife J. McCooey[1], Courtney Bull[1], Emily Hoey[1], Nicoleta Sinevici[1], Holly Hall [4], Baharak Ahmaderaghi [8], Enric Domingo [9], Andrew Blake[9], Susan D. Richman [10], Claudio Isella[6,7], Crispin Miller[4,11], Andrea Bertotti[6,7], Livio Trusolino [6,7], Maurice B. Loughrey[1,12], Emma M. Kerr [1], Sabine Tejpar [13], S:CORT consortium*, Timothy S. Maughan[9], Mark Lawler[1], Andrew D. Campbell [4], Simon J. Leedham [5], Viktor H. Koelzer [2,9], Owen J. Sansom [4,11] & Philip D. Dunne [1,4] ✉

[1]The Patrick G Johnston Centre for Cancer Research, Queen's University Belfast, Belfast, UK. [2]Department of Pathology and Molecular Pathology, University Hospital Zurich, University of Zurich, Zurich, Switzerland. [3]Faculty of Medicine, Université de Genève, Geneva, Switzerland. [4]Cancer Research UK Scotland Institute, Glasgow, UK. [5]Centre for Human Genetics, University of Oxford, Oxford, UK. [6]Candiolo Cancer Institute, FPO IRCCS, Candiolo, Torino, Italy. [7]Department of Oncology, University of Torino, Candiolo, Torino, Italy. [8]School of Electronics, Electrical Engineering and Computer Science, Queen's University Belfast, Belfast, UK. [9]Department of Oncology, University of Oxford, Oxford, Oxfordshire, UK. [10]Leeds Institute of Medical Research, University of Leeds, Leeds, UK. [11]School of Cancer Sciences, University of Glasgow, Glasgow, UK. [12]Department of Cellular Pathology, Royal Victoria Hospital, Belfast Health and Social Care Trust, Belfast, UK. [13]Department of Oncology, Katholieke Universiteit Leuven, Leuven, Belgium. [14]These authors contributed equally: Ryan M Byrne, Maxime W Lafarge. *Lists of authors and their affiliations appear at the end of the paper. ✉e-mail: p.dunne@qub.ac.uk

**S:CORT consortium**

Andrew Blake[9], Enric Domingo[9], Viktor H. Koelzer[2,9], Simon J. Leedham[5], Timothy S. Maughan[9], Susan D. Richman[10], Philip D. Dunne[1,4], Mark Lawler[1] & Keara L. Redmond[1]

A full list of members appears in the Supplementary Information.

## Methods

The FOCUS data (GSE156915 ref. [11]) used for class discovery and classifier development in this study were accessed through the UK Medical Research Council S:CORT program[14]. All subjects provided written informed consent on their samples at the time of entry to the clinical trials for further research. The clinical trial (FOCUS; reference 79877428) and the study (S:CORT; reference 15/EE/0241) were approved by the National Research Ethics Service in the United Kingdom[11].

### Bulk datasets

The FOCUS (GSE156915; ref. [11]) and SPINAL (GSE248381) datasets were generated within the S:CORT program[14], in which microarray gene expression profiles, mutation, clinical, immunohistochemistry, tissue blocks and tumor microarrays were available. FOCUS is a Medical Research Council-funded randomized trial cohort consisting of 360 formalin-fixed paraffin-embedded primary tumor samples for metastatic CRC. SPINAL consists of 258 formalin-fixed paraffin-embedded samples from patients with CRC, at mixed stages. Other publicly available datasets were accessed from Gene Expression Omnibus with accession numbers GSE39582 (ref. [2]), GSE31279 (ref. [40]), GSE143915 (ref. [27]) and GSE218776 (ref. [29]), and from ArrayExpress E-MTAB-6363 (ref. [41]). The validation of clinical association was carried out with the PETACC-3 cohort[15]. The Cancer Genome Atlas (TCGA) dataset for colon and rectal adenocarcinoma (COREAD[42]) was accessed and extracted from the Genomic Data Commons (GDC) through TCGAbiolinks[43]. In all microarray datasets, the probes-to-genes were collapsed using the *collapseRows* function in the *WGCNA* R package (v.1.70-3), in which the probe with the highest average value per gene was selected[44]. For TCGA COREAD RNA-seq data, the *TCGAbiolinks* R package (v.2.16.1) was used to download HT-seq counts. Using the *varFilter* in the *genefilter* R package (v.1.70.0), low-variance genes (var.cutoff = 0.25) were excluded followed by quantile normalization and $\log_2$ transformation to the count matrix.

### Single-cell datasets

Two epithelial scRNA-seq datasets were also used for the study—a CRC tissue-derived scRNA-seq merged dataset from five different cohorts[6] and a scRNA-seq dataset derived from murine organoids mono-cultured or co-cultured with fibroblasts and/or macrophages[28]. For the scRNA-seq human CRC dataset[6], the processed count expression matrix for $n = 49,155$ epithelial cells and the corresponding epithelial metadata were downloaded through the Synapse under accession code syn26844071. The murine organoid scRNA-seq dataset[28] consists of $n = 29,452$ epithelial cells from WT mouse colonic organoids and at least five different genotypic CRC organoids, including *shApc* (A), *Kras*^G12D/+ (K), *shApc* and *Kras*^G12D/+ (AK), *Kras*^G12D/+ and *Trp53*^R172H/- (KP) and *shApc*, *Kras*^G12D/+ and *Trp53*^R172H/- (AKP), and all the corresponding metadata were also downloaded from a previous publication[28].

### GEMMs

All animal experiments were performed according to a UK Home Office license (Project License 70/8646) and were reviewed by the animal welfare and ethical board of the University of Glasgow. Both male and female 6–12-week-old mice were systematically induced with a single injection of 2 mg tamoxifen (Sigma-Aldrich, T5648) using intraperitoneal injection and sampled at clinical endpoint, which was defined as weight loss and/or hunching and/or cachexia. All experiments were performed on mice with a C57BL/6 background. RNA was extracted using either an RNeasy mini kit (Qiagen) or TRIzol reagent (ThermoFisher Scientific) and its concentrations were assessed using a NanoDrop 200c spectrophotometer (ThermoScientific). RNA quality was evaluated using an Agilent 220 TapeStation ScreenTape and sequenced using an Illumina TruSeq RNA sample prep kit, then run on an Illumina NextSeq using the High Output 75 cycles kit (2 × 36 cycles, paired-end reads, single index). The quality of the raw sequence was assessed using the FastQC algorithm (v.0.11.8). Sequences were trimmed to

remove adaptors and low-quality base calls, defined as those with a Phred score of <20, using the Trim Galore tool (v.0.6.4). Thereafter, the trimmed sequences were aligned to the mouse genome build GRCm38.98 using HISAT2 (v.2.1.0), then FeatureCounts (v.1.6.4) was used to determine raw counts per gene. Two mouse cohorts that were used in our previous study[29], namely small cohort ($n = 18$; E-MTAB-6363 (ref. [41])) and large cohort ($n = 39$; GSE218776 (ref. [29])), were amalgamated to create a larger cohort of GEMMs. APN and AP models were excluded, as the batch they were sequenced in was deeply confounded by genotype, resulting in a collection of $n = 51$ tumor samples, including six genotypes: *Apc*^fl/+ (A); *Apc*^fl/+ *Kras*^G12/+ (AK); *Braf*^V600E/+ *Trp53*^fl/fl (BP); *Braf*^V600E/+ *Trp53*^fl/fl Notch1^Tg/+ (BPN); *Kras*^G12D/+ *Trp53*^fl/fl Notch1^Tg/+ (KPN); and *Kras*^G12D/+ *Trp53*^fl/fl (KP). After batch correction using the *ComBat*_seq function in the *sva* R package (v.3.40.0), the *vst* function in the *DESeq2* R package (v.1.32.0) was applied to normalize the read counts.

**Intestinal organoid culture.** Organoid lines were generated from the small intestine of uninduced WT and tamoxifen-induced *Villin*^CreER *Apc*^fl/fl *Kras*^G12D/+ (AK) mice. Two biological replicates per genotype were used. Tissue segments were collected from the small intestine (5–15 cm from the pyloric sphincter), and samples were immediately washed with PBS and kept on ice. For isolation[45], the samples were dissociated in 2 mM EDTA at 4 °C for 30 min and subsequently shaken and separated into four fractions. Fractions two to four, containing intestinal crypt material, were filtered through a 70 µm cell strainer and centrifuged at 600 rpm for 3 min. The cell pellet was resuspended in Matrigel (BD Bioscience, 356231) and seeded in accordance with the pellet volume obtained.

For culturing[46], organoid base media was made up of Advanced DMEM/F12 (ThermoFisher Scientific, 12634010) supplemented with 1 mM L-glutamine (ThermoFisher Scientific, 25030024), 10 µM HEPES Buffer (ThermoFisher Scientific, 15630056), penicillin–streptomycin (100 U ml⁻¹) (ThermoFisher Scientific, 15140122), N2 (ThermoFisher Scientific, 17502001) and B27 (ThermoFisher Scientific, 12587001) and is hereafter referred to as complete organoid base media. Intestinal crypts were cultured in complete organoid base media further supplemented with 50 ng ml⁻¹ Recombinant Human EGF (Peprotech, AF10015), 100 ng ml⁻¹ Recombinant Murine Noggin (Peprotech, 25038) and 500 ng ml⁻¹ Recombinant Murine R-spondin1 (R&D Systems, 3474-RS), hereafter referred to as ENR media.

To assess the effect of ENR withdrawal on organoid transformation, WT and AK organoid lines were split at a ratio of 1:6 following the formation of large budding structures and then cultured for 48 h with either complete organoid media supplemented with ENR or complete organoid base media only. Organoids were removed from the culture plate using ice-cold PBS and then transferred to a 15 ml falcon and subjected to 3× wash and centrifugation steps at 600 rpm for 3 min. The resulting cell pellet was fixed in 10% paraformaldehyde at 4 °C for 10 min. After aspirating the supernatant, the cell pellet was resuspended in 2% agarose (Melford, A20090-500) and embedded in paraffin blocks. The following antibodies were used according to standard histological processing techniques: BrdU (BD Biosciences, 347580, TRS High, 1:250), Ki67 (Cell Signaling, 12202, ER2 20 min, 1:1000), Chromogranin-A (AbCam, ab108388, TRS High, 1:600) and Synaptophysin (Cell Signaling, 36406, TRS High, 1:150). In situ hybridization was performed on the Leica Bond Rx Autostainer using the following RNAscope probes according to the manufacturer's instructions: Lgr5 (312178), Anxa1 (509298), Clu (427898) and Olfm4 (311838). Positive (mmPpib, 313918) and negative (mm-DapB, 312038) control probes were implemented to ensure staining integrity and accuracy. Images were acquired using an Olympus BX51 and Zen2 Lite Blue imaging software at ×20 magnification.

### Gene signatures

The majority of the gene set signatures, including the Hallmark, BIOCARTA, KEGG, PID and REACTOME gene set collections, were

accessed from the MSigDB using the *msigdbr* R package (v.7.4.1) for both human and mouse species[4]. Other gene set signatures included: (1) stem-related CBC and RSC[19]; (2) precursor polyp-related tubular and serrated signatures obtained from differential gene expression analysis between tubular and serrated adenomas from GSE45270 (ref. 20) using the *limma* R package[47] (v.3.46.0); (3) MYC and PRC target modules[26]; and (4) a list differentiation-specific markers[28,48,49] (Supplementary Table 3) used for the pathway-based analyses. These gene signatures were also applied to mouse data where they were directly applicable using MSigDB, or the gene signatures were converted to the mouse orthologues with *ensembl* using the *biomaRt* R package[50] (v.2.50.3).

## Unsupervised discovery of PDS

Four well-curated publicly available gene set collections, BIOCARTA (gene set, $n = 289$), KEGG ($n = 186$), PID ($n = 196$) and REACTOME ($n = 1499$), were accessed from the MSigDB with the *msigdbr* R package (v.7.0.1) and were used to generate a matrix of ssGSEA scores from gene expressions of the FOCUS cohort using the *GSVA* R package (v.1.26.0). The parameters *min. sz = 10*, *method = 'ssgsea'* and *ssgsea.norm = T* were set, resulting in the scores for 1,783 gene sets. A subset of 165 *KRAS*mut primary tumor CRC samples were selected from the ssGSEA score matrix (excluding *KRAS* WT, *BRAF* and *NRAS* mutants) for the unsupervised class discovery phase. The score matrix was subjected to dimensionality reduction (*t*-SNE) analysis, and the two continuous variables (Dim1 and Dim2) were obtained using the *Rtsne* R package (v.0.15). The variables were scaled before applying unsupervised *k*-means clustering. The silhouette width and elbow methods determined $k = 3$ as an optimal number of clusters (*cluster* R package v.2.1.2; *factorextra* R package v.1.0.7), and the bootstrap resampling method from the *fpc* R package (v.2.2.3) identified $k = 3$ as the highly stable number for clustering. Following unsupervised *k*-means clustering, three groups of clusters were named: PDS1, PDS2 and PDS3.

## Development and application of the PDS classification system

Using the discovery subset ($n = 165$) in which the PDS classes were defined previously, the discovery set was randomly divided into a training set ($n = 125$) and a test set ($n = 40$) based on the bootstrap resampling method using the *caret* R package (v.6.0-90). Three different classification algorithms were tested, including the nearest shrunken centroid (or prediction analysis of microarrays), lasso and elastic-net regularized generalized linear model (glmnet) and svmRBF, implemented in the *caret* R package (v.6.0-90). As a feature selection step to reduce the number of gene sets and draw out only subgroup-specific gene sets, ssGSEA scores that were above average specific to each subgroup were selected and further highly correlated gene sets (>0.9) were excluded, resulting in 626 gene sets in total (Supplementary Table 2). To make it feasible for users, the test run was performed on the gene expression matrix, and the ssGSEA score conversion steps were implemented within the classification model. The gene expression matrix of the test samples was first converted to the ssGSEA scores with the *gsva* function along with the parameters *ssgsea.norm = F*, which generated unscaled scores. The unscaled ssGSEA scores were scaled using the min–max scaling method in the *gsva* function, in which the minimum and maximum values determined during the class discovery phase were used. Once the score matrix of the test samples was generated, it was batch-corrected against the training set as the reference batch using the *ComBat* function from the *sva* R package (v.3.42.0).

Before running the classification algorithms, leave-one-out cross-validation was used in the training set to minimize overfitting of classification. The classification algorithms were trained on the training set and the corresponding hyperparameters were adjusted to finalize the models. Out of the three classification models, the svmRBF algorithm displayed a high classification performance on the test data; therefore, it was selected to develop the PDS classification system. The *PDSclassifier* R package has been developed for the PDS classification model that is available to share (see Code availability).

## Pathway analysis

To define subtype-specific biological associations, GSEA was used with the *fgsea* R package[51] (v.1.21.0) with *eps = 0* and *nPerSimple = 10,000*. For *fgsea*, a ranked gene list was first obtained for each comparison in each dataset with the *limma* R package (v.3.50.3). The comparison between subtypes was made in a grouped pairwise manner. Statistical significance was measured with a Benjamini–Hochberg false discovery rate of <0.05, and normalized enrichment scores indicate upregulation (positive value) or down-regulation (negative value).

The ssGSEA scores in bulk datasets were generated with the *GSVA* R package (v.1.42.0) with the 'ssgsea' method[52] and *ssgsea.norm = T*. For scRNA-seq datasets, ssGSEA scores per cell from the signatures were calculated using the *enrichIt* function in the *escape* R package[53] (v.1.6.0) with *method = 'ssgsea'* and *min.size = 1*.

## Proliferative index, replication stress and ISC index

Replication stress was calculated from a collection of transcriptional signatures ($n = 20$; Supplementary Table 4) associated with cell cycle and DNA repair extracted from the MSigDB for both species, and GSVA scores were generated using *'gsva'* method from *GSVA* R package[52] (v1.42.0), followed by the total sum of the GSVA scores per sample across gene sets[22]. The list of gene sets was used to determine the enrichment scores per cell using *escape* followed by the sum of enrichment scores per cell to obtain replication stress scores, which were subsequently scaled between −1 and 1 using the *rescale* function in the *scales* R package (v.1.2.0) in the scRNA-seq dataset. The transcriptomic measure of proliferation was calculated with the *ProliferativeIndex* R package[21] (v.1.0.1), which calculates a proliferative index from a list of proliferative cell nuclear antigen-associated genes. For the mouse model, the proliferative cell nuclear antigen-associated gene signature was converted to mouse orthologues as mentioned previously before applying the signature to calculate the proliferative index score. The gene signature was extracted directly from the R package and used to calculate enrichment scores per cell using *escape* in the scRNA-seq cohort. Likewise, the *CellCycleScoring* function in the *Seurat* R package[31,54] was also used to predict the cell cycle phase per cell into G1, S and G2M. The ISC index provides a continuum score from the gene expression dataset that represents the stem cell phenotype, with the extreme ends of the scoring scale as either a strongly conventional CBC or an RSC phenotype. The *ISCindex* R package[19] (v.0.0.0.9) was downloaded and used for the bulk dataset. Additionally, these CBC and RSC gene signatures were also used to obtain ssGSEA scores for the bulk and scRNA-seq datasets.

## TF activity analysis

For the quantification of TF activity from the gene expression profiles, the collection of well-curated TFs and their targets in the DoRothEA[12] database was accessed; the TF-target interaction with high confidence A and B were selected for the analyses of both human and mouse using the *dorothea* R package (v.1.6.0). Statistically significant differences ($P < 0.05$) in TF activity between subtypes were determined using the '*rowTtest*' function from the *viper* R package[55] (v.1.28.0). A list of uniquely activated TFs per subtype across cohorts was identified and visually presented in a heatmap.

## Cell lineage analysis

ESTIMATE[18] was applied to produce immune and stromal fractions as well as tumor purity scores using the *estimate* R package (v.1.0.13) and visualized using the ggplot2 R package.

## CRC molecular subtyping

Tumor samples in each cohort were classified into CMS using the random forest method from *CMSclassifier* (v.1.0.0) at a default threshold, with the exception of the FOCUS and SPINAL cohorts, for which the CMS posterior probability threshold levels were reduced to 0.4

(ref. 5). CMS for mouse data were called using the 'Option C' classification in the *MmCMS* R package[29] (v.0.1.0). CRIS classification was made using *CRISclassifier* (v.1.0.0) at default settings[3]. Unclassified samples were determined using the recommended Benjamini–Hochberg false discovery rate of >0.2. For iCMS bulk classification[6], the iCMS gene signatures were extracted to create the iCMS template, which was subsequently used with the nearest template prediction method embedded in *CMScaller* (v.2.0.1)[56]. Samples above a false discovery rate of 0.05 were classed as 'unknown'. In the scRNA-seq dataset, iCMS labels were used as previously defined. Furthermore, iCMS2-high and iCMS3-high labels were created for each cell using the ssGSEA score from the iCMS gene signatures and selecting top tertiles.

## Immunohistochemistry and digital histology scoring

Tissue microarrays were scanned at ×20 magnification and imported into QuPath[57] (v.0.2.3). The suitability for inclusion of individual Ki67[+] immunohistochemistry-stained cores was determined by manual visual assessment of the scanned images, after application of the QuPath tissue microarray dearraying tool. Color deconvolution was applied to separate stains, followed by tissue detection (pixel threshold, resolution, low (7.96 µm per pixel); channel, average channels; Gaussian prefilter; smoothing sigma, 2.0; tissue threshold, 235). Cells were detected within the annotated tissue (requested pixel size, 0.5; nucleus background radius, 15.0; median filter radius, 0.0; sigma, 2.5; minimum area, 10.0; maximum area, 300.0; threshold, 0.1, maximum background intensity, 1.0; exclude DAB, false; cell parameters, default; general parameters, default) and smoothed (radius, 25 µm). An object classifier (random trees; default settings) was trained by examples of annotated tumor epithelium and stroma. Set cell intensity classification (nucleus: DAB OD mean, 0.18) to differentiate positive and negative tumor epithelium and stroma. Additionally, an H-score was also generated by setting a three-tier cell intensity classification (nucleus: DAB OD mean, 0.10; 0.25; 0.42). This resulted in a total of 354 patients with matched PDS call and Ki67[+] assessment. The total tumor area was also enumerated and assessed based on the digital histology scoring using the HALO platform (Indica Labs, Albuquerque, NM, USA) on the H&E whole slide images (WSIs) from the FOCUS cohort (*n* = 356)[58].

## Digital image-based PDS classifier

We developed a set of deep-learning classifiers and analyzed their performances using WSIs from the FOCUS and SPINAL cohorts. After rejecting images of poor quality and images from patients with undefined PDS calls after pathologist review, we used a dataset of 997 WSIs of H&E-stained resection specimens from 520 patients (PDS1, *n*_slides = 329; PDS2, *n*_slides = 314; PDS3, *n*_slides = 354). We conducted experiments under a fivefold cross-validation protocol: for each fold, we split the data according to a 60–20–20 training–validation–test distribution such that classes and cohorts were stratified. We made sure that the validation and test splits did not overlap across the five folds and that images from the same patient were always in the same split. Tumor regions were manually annotated in all WSIs; we then restricted our experiments to the use of image data from these regions to prevent potential classification bias from non-tumor regions. The tumor regions of these WSIs were tiled into sets of image patches of size 318 × 318 pixels, extracted at ×5 magnification (resolution ~2 µm per pixel) with 50% overlap. We used a customized 50-layer ResNet as a deep-learning architecture to process the input image patches and output a probability density over the PDS classes. Each image patch was labeled using the PDS class of their WSI of origin, and our models were trained to maximize the output probability for the target class (minimization of the cross-entropy loss) using mini-batches of size 16. At inference time, we applied the trained models on all the generated tiles from a given WSI and then averaged their predicted probability densities to produce slide-level probability estimates. We then selected the class with the highest relative probability score as the image-based

PDS call for this WSI. This approach was based on the weakly supervised learning protocol proposed within the image-based CMS study[16]. The classification performance of the trained models was systematically assessed using the test patrician for each fold.

## Mutation and copy number profiles

Mutational and copy number associations between PDS were determined, when available, across at least four different CRC cohorts: FOCUS[11], GSE39582) ref. 2), SPINAL (S:CORT cohort) and TCGA COREAD[42]. The proportion of driver mutations and their variants (*KRAS*, *BRAF* and *TP53*) were examined in detail across these cohorts. The mutational data for TCGA COREAD were retrieved from the GDC with the *TCGAbiolinks* R package (v.2.25.2) and analyzed using the *maftools* R package (v.2.12.0)[59]. Oncoprint was also used to interrogate and visualize the genetic alterations in key driver genes along signaling pathways, including WNT, MAPK, PIK3CA, cell cycle and TGF-β pathways, using the *ComplexHeatmap* (v.2.12.0) and *circlize* (v.0.4.15) R packages[60]. For the FOCUS and SPINAL cohorts, copy number chromosomal arm calls and copy number estimations per gene were available via the S:CORT consortium. For TCGA COREAD, the GISTIC data were accessed through the GDC data portal[42].

## Survival analyses

Two different cohorts (GSE39582 and PETACC-3) were explored for clinical and prognostic evaluation. Only PDS samples were considered for the analysis, excluding 'mixed' samples. Moreover, patients with missing information on RFS status, relapse-free months or chemotherapy treatment status, and patients with records dating back less than one month were excluded from all survival analyses in GSE39582. The Cox proportional hazards method was also performed to calculate the hazard ratio and confidence intervals for statistical group comparisons. The analysis for the PETACC-3 clinical trial (NCT00026273) was performed by P.K.T. All survival analyses and visualizations were carried out with the *survival* (v.3.2-13), *survminer* (v.0.4.9) and *ggplot*2 (v.3.3.6) R packages.

## Single-cell human data analyses

The count expression matrix for tumor epithelial cells was normalized using *SCTransfrom* in the *Seurat* R package[54] (v.4.1.1) with *method* = 'glmGamPoi' implemented from the *glmGamPoi* R package (v.1.8.0). Using Seurat functions, the data were subsequently clustered and visualized using uniform manifold approximation and projection (UMAP). Additionally, the *n* = 626 PDS-specific gene sets were used to generate a single-cell matrix of ssGSEA scores using the *enrichit* function from the *escape* R package. The matrix was used as an assay in the Seurat object, in which it was clustered and visualized in a UMAP plot. Cells were defined as 'high', 'mid' or 'low' based on tertile using MYC targets and PRC targets ssGSEA scores, whereby MYC-high–PRC-low were considered as MYC targets-high and PRC-high–MYC-low were considered as PRC targets-high. Similarly, in addition to the previous classification of iCMS on the data, iCMS gene signatures were used to generate iCMS2 and iCMS3 ssGSEA scores and, with the same approach, iCMS2-high and iCMS3-high were also defined. In addition to UMAP visualization from Seurat, the data were also visualized with the high-dimensionality reduction PHATE method[61] using the *phateR* (v.1.0.7) R package.

## Trajectory inference analysis and differentiation potency

The single-cell fate mapping and trajectory inference were examined using the Python-based *CellRank* (v.1.5.2) method[39] in both single-cell datasets. To maintain consistency in both datasets, *CytoTRACEkernel* was used to compute pseudotime, and the information was projected onto the UMAP with arrows displaying RNA velocity-like directionality towards increasing differentiation status.

Further assessment of differentiation status was performed using the *SCENT* (v.1.0.3) R package, which computed the correlation of connectome and transcriptome scores using a human or murine version of the net17Jan16 protein–protein interaction network for both scRNA-seq

data, which highlights the differentiation potency of an epithelial cell[62]. Waddington-like landscapes were visualized in 3D using SideFX Houdini 19.5 as previously described[28].

## Calculation of the SMI

The SMI provides a method of transcriptomic measure along the stem–differentiation continuum in association with stem-like or differentiation-like properties. SMI can be calculated from the gene expression profile of both bulk tumor tissue and single-cell data and is simply the difference between the PRC targets and MYC targets ssGSEA score scaled between a value of −1 and 1. The method has also been implemented in the *PDSclassifier* R package in the form of the *calculateSMI* function.

## Statistics and data visualization

Data interrogations, analyses, visualizations and interpretation were mostly processed using R (v4.2.1) in RStudio and Python (v.3.9). Statistical analyses were performed in R using the *stats* (v.4.2.1) or *ggpubr* (v.0.4.0) package for plots and included the two-sided Wilcoxon rank-sum test, Kruskal–Wallis rank-sum test, Fisher's exact test and Pearson's correlation coefficient test. For copy number by arm analysis, Pearson's chi-squared test post-hoc analysis was performed using the *chisq.posthoc.test* R package (v.0.1.2), and Benjamini–Hochberg adjusted $P$ values were determined using the *p.adjust* function in the *stats* R package. Other R packages that were used for data analysis and visualization include *ggtern* (v.3.3.5), *ComplexHeatmap* (v.2.10.0), *circlize* (v.0.4.15), *umap* (v.0.2.8.0), *ggplot2* (v.3.3.6), *patchwork* (v.1.1.1), *riverplot* (v.0.10), *ggforce* (v.0.3.3) and *RColorBrewer* (v.1.1-2).

## Reporting summary

Further information on research design is available in the Nature Portfolio Reporting Summary linked to this article.

## Data availability

The publicly available bulk gene expression dataset used in this study for both the human and mouse models is referred to in the Methods, with corresponding GEO accession numbers or through TCGAbiolinks for the TCGA RNA-seq dataset. Data for scRNA-seq were accessed directly from previously published studies[6] as detailed in the Methods. The SPINAL and FOCUS data in this publication were generated by the S:CORT consortium and are freely available for use by academic researchers and not-for-profit organizations for academic, teaching and educational purposes. Gene expression profiles for the S:CORT-led data have been made available at GEO (SPINAL, GSE248381; FOCUS, GSE156915). The SPINAL and FOCUS data are also available for commercial use, on commercial terms, through Cancer Research Horizons (https://www.cancerresearchhorizons.com).

## Code availability

The *PDSclassifier* R package (v.1.0.0) is available on the Molecular Pathology Lab GitHub (https://github.com/MolecularPathologyLab/PDSclassifier). The 'SubtypeExploreR' ShinyApp interactive web application is live (https://subtypeexplorer.qub.ac.uk) and the code has been made available on GitHub (https://github.com/MolecularPathologyLab/SubtypeExploreR). All relevant and original codes and scripts related to the article are publicly available on the website (https://dunne-lab.com) and on GitHub (https://github.com/MolecularPathologyLab).

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

## Acknowledgements

This work was supported by a Cancer Research UK (CRUK) early detection grant (P.D.D.; A29834), a CRUK International accelerator

program (ACRCelerate) (P.D.D., O.J.S., S.J.L., M.L. and T.S.M.; A26825), a UK Medical Research Council (MRC) and CRUK co-funded Stratified Medicine Consortium program grant (S:CORT) (P.D.D., M.L. and T.S.M.; MR/M016587/1), an MRC National Mouse Genetics Network program (P.D.D. and S.J.L.; MC_PC_21042), CRUK Beatson institute funding (O.J.S.; A21139, A17196 and A31287). CRUK program grant (S.J.L.; DRCNPG-Jun22\100002), Lee Placito Medical Research Fund (E.J.M.; University of Oxford), Health Data Research UK Grant (M.L.), AIRC–Associazione Italiana per la Ricerca sul Cancro, Investigator Grants 20697 (A.B.) and 22802 (L.T.), Promedica Foundation F-87701-41-01 (V.H.K.), a My First AIRC Grant (C.I.; ID 19047); AIRC 5×1000 grant 21091 (L.T., A.B.); European Research Council Consolidator Grant 724748 BEAT (A.B.); H2020 grant agreement no. 754923 COLOSSUS (L.T.); H2020 INFRAIA grant agreement no. 731105 EDIReX (A.B.); Fondazione Piemontese per la Ricerca sul Cancro-ONLUS, 5×1000 Ministero della Salute 2016 (L.T.); BOF-Fundamental Clinical Research mandate from KU Leuven and by the Belgian Foundation Against Cancer (S.T.; FAF-C/2018/1301). General support for the Dunne research group is provided by the QUB Foundation.

## Author contributions

S.B.M. and P.D.D. conceived the initial data-driven investigation; E.M.K was involved with the conceptualization and R.M.B. was involved with the investigation. S.B.M. discovered and developed the classifier and methodology as well as conducting data analysis and data visualization. R.M.B., S.M.C., N.C.F., P.K.T., R.A., S.L.M. and E.R. analyzed the data and performed data visualization. M.W.L. and V.H.K. performed digital image-based classification; M.W.L., V.H.K. and N.C.F. analyzed histological data. M.B.L. conducted pathological analysis. M.L.M., R.A.R., T.R.M.L., A.K.N. and A.C. performed mouse experiments and generated mouse model data; K.L.G. curated the mouse data. E.J.M. and H.L.B.D. performed the multiplex analysis. E.G. and M.V. performed xenograft data generation, data curation and data analysis. C.B., E.H. and E.D. analyzed the data; K.L.R. was involved with data generation and sequencing. A. Blake curated the data and acquired the software; S.D.R. was involved with data generation. S.D.R., A. Blake, L.T., M.B.L., S.T., T.S.M., M.L., S.J.L., V.H.K., O.J.S. and P.D.D. were involved with resource acquisition. A. Bertotti, L.T., M.B.L., E.M.K., S.T., T.S.M., M.L., S.J.L., V.H.K., O.J.S. and P.D.D. supervised the project. S.B.M and P.D.D. wrote the original draft of the manuscript and all authors reviewed and edited the final version.

## Competing interests

The authors declare no competing interests.

## Additional information

**Extended data** is available for this paper at https://doi.org/10.1038/s41588-024-01654-5.

**Correspondence and requests for materials** should be addressed to Philip D. Dunne.

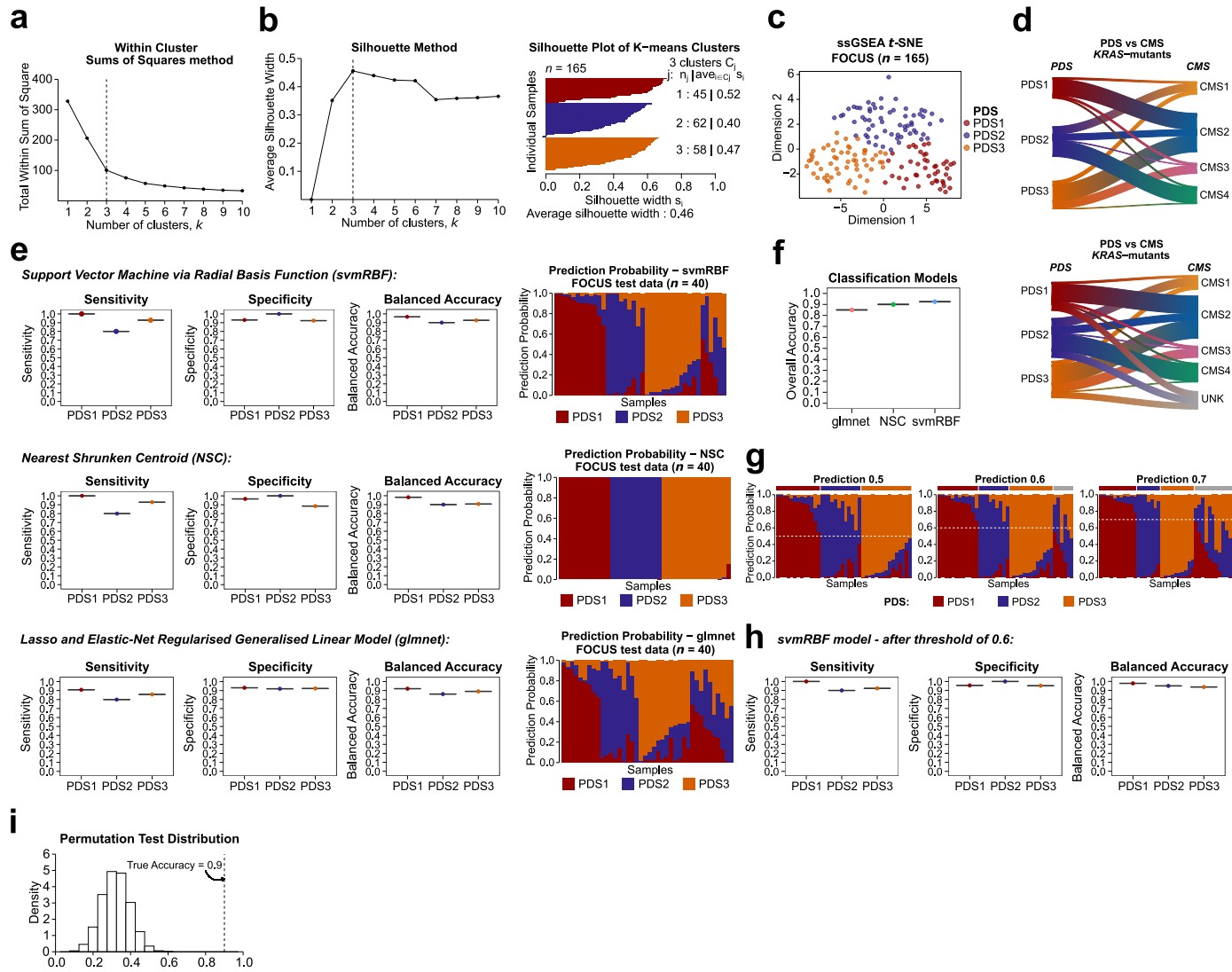

**Extended Data Fig. 1 | Classification model comparisons and development of PDS classification. a**, Elbow or within cluster sums of squares method on $k = 1{:}10$ indicated $k = 3$ (*shown with vertical black dashed line*) as optimal number of clusters. **b**, Silhouette width method on $k = 1{:}10$ further confirms $k = 3$ as the optimal number of clusters. **c**, *t*-SNE plot on ssGSEA gene set score matrix of the discovery set ($n = 165$) with PDS annotated. **d**, Sankey plots show PDS (discovery calls) and CMS per sample across the *KRAS* mutants (*KRAS*muts) discovery set with (*top*) or without (*bottom*) CMS 'unknown'. **e**, Sensitivity, specificity, balanced accuracy, and the prediction probability per sample on the test set ($n = 40$) for three different classification models: support vector machine via radial basis function (svmRBF; *top*), nearest shrunken centroid (NSC; *centre*), and lasso and elastic-net regularised generalised linear model (glmnet; *bottom*). **f**, Comparison of the overall accuracy between the three classification models. **g**, Bar chart represents PDS prediction probabilities using svmRBF model on the test data with three different threshold levels at 0.5 (*left*), 0.6 (*centre*) and 0.7 (*right*). The dashed white line denotes the threshold where samples below the threshold are labelled as 'Mixed' samples, annotated at the top. **h**, Sensitivity, specificity, and balanced accuracy following application of the 0.6 threshold to the PDS prediction. **i**, Histogram highlights permutation test for the PDS classification on the test data.

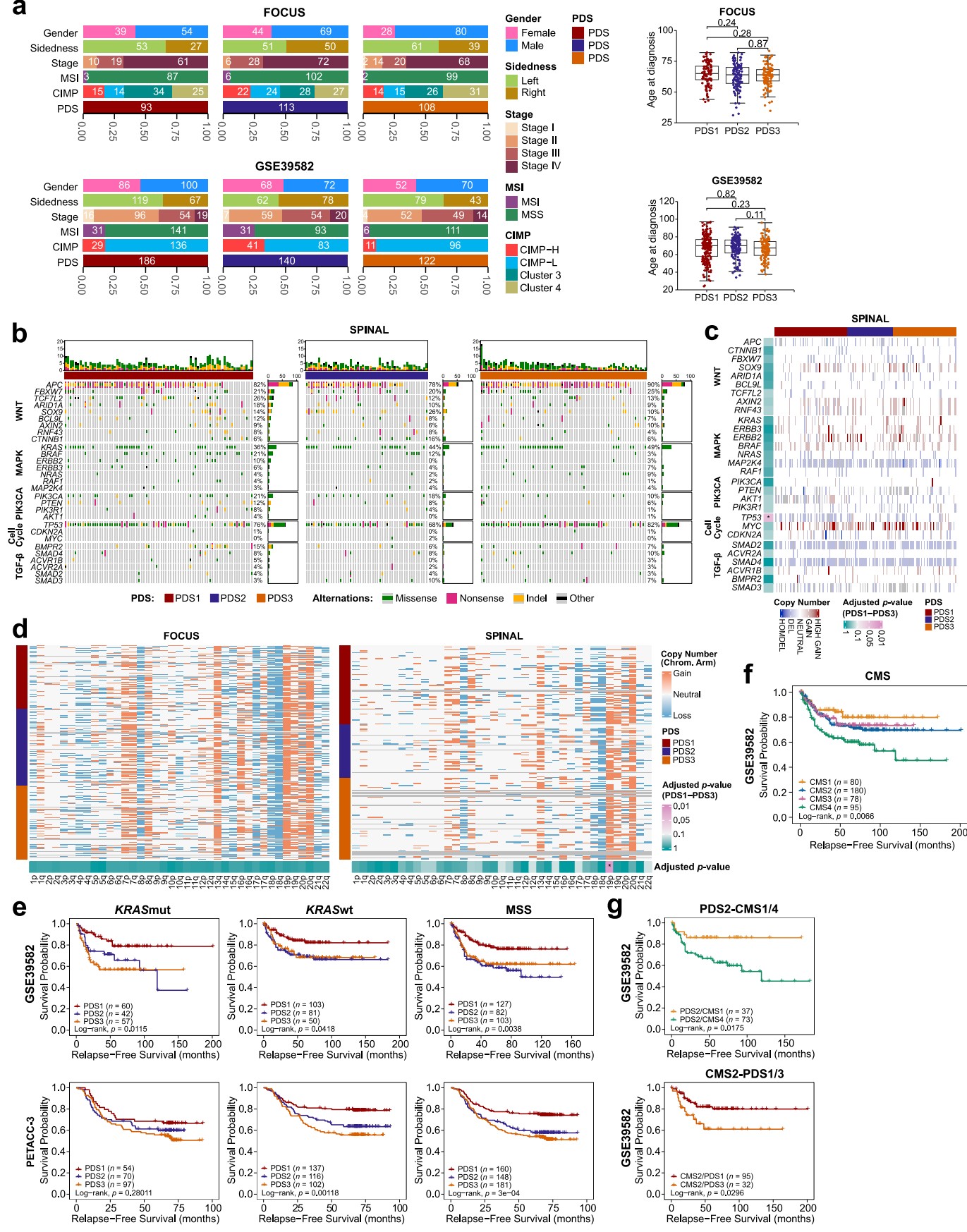

**Extended Data Fig. 2 | See next page for caption.**

**Extended Data Fig. 2 | Clinicopathological and genomic associations of PDS.**
**a**, Proportions of samples examined for clinical/molecular features including sidedness, stage, MSI, and CIMP (*left*), and boxplot displaying age at diagnosis (*right*) across PDS in FOCUS (boxplot: PDS1, *n* = 93; PDS2, *n* = 113; PDS3, *n* = 108) and GSE39582 (boxplot: PDS1, *n* = 186; PDS2, *n* = 139; PDS3, *n* = 122) cohorts. *P*-values (boxplot): Wilcoxon rank-sum test. **b**, Oncoprint with key driver mutation genes from WNT, MAPK, PIK3CA, cell cycle and TGF-β pathways in the SPINAL cohort. **c**, Copy number gain/loss per gene from the SPINAL cohort visualised as heatmap. Two-sided Fisher's exact test between PDS1 and PDS3 followed by post hoc analysis, Benjamini-Hochberg adjusted *P*-value annotated at the left-side bar of the heatmap; asterisk denotes Benjamini-Hochberg

adjusted *P*-value < 0.05 (*TP53*, *P*-value = 0.0081). **d**, Copy number variation by chromosome arms in the FOCUS (*left*) and SPINAL (*right*), ordered based on PDS as rows. Statistics as in **c** (19p, *P*-value = 0.00204). **e**, Kaplan-Meier plots of the relapse-free survival (RFS) of colon cancer patients per CMS in the GSE39582 cohort. **f**, RFS Kaplan-Meier plots of colon cancer patients classified by PDS in the GSE39582 (*top*) and PETACC-3 (*bottom*) cohorts in *KRAS* mutants (*KRAS*mut), *KRAS* wild-types (*KRAS*wt) and microsatellite stable (MSS) only. **g**, RFS Kaplan-Meier plots on the PDS2/CMS1 vs PDS2/CMS4 and CMS2/PDS1 vs CMS2/PDS3 subsets in the GSE39582 cohort. **h**, Heatmap represents PDS 'Hallmarks' ssGSEA scores with PDS classifications across three TCGA cancer type cohorts: COREAD (*left*), LUAD (*centre*) and PAAD (*right*).

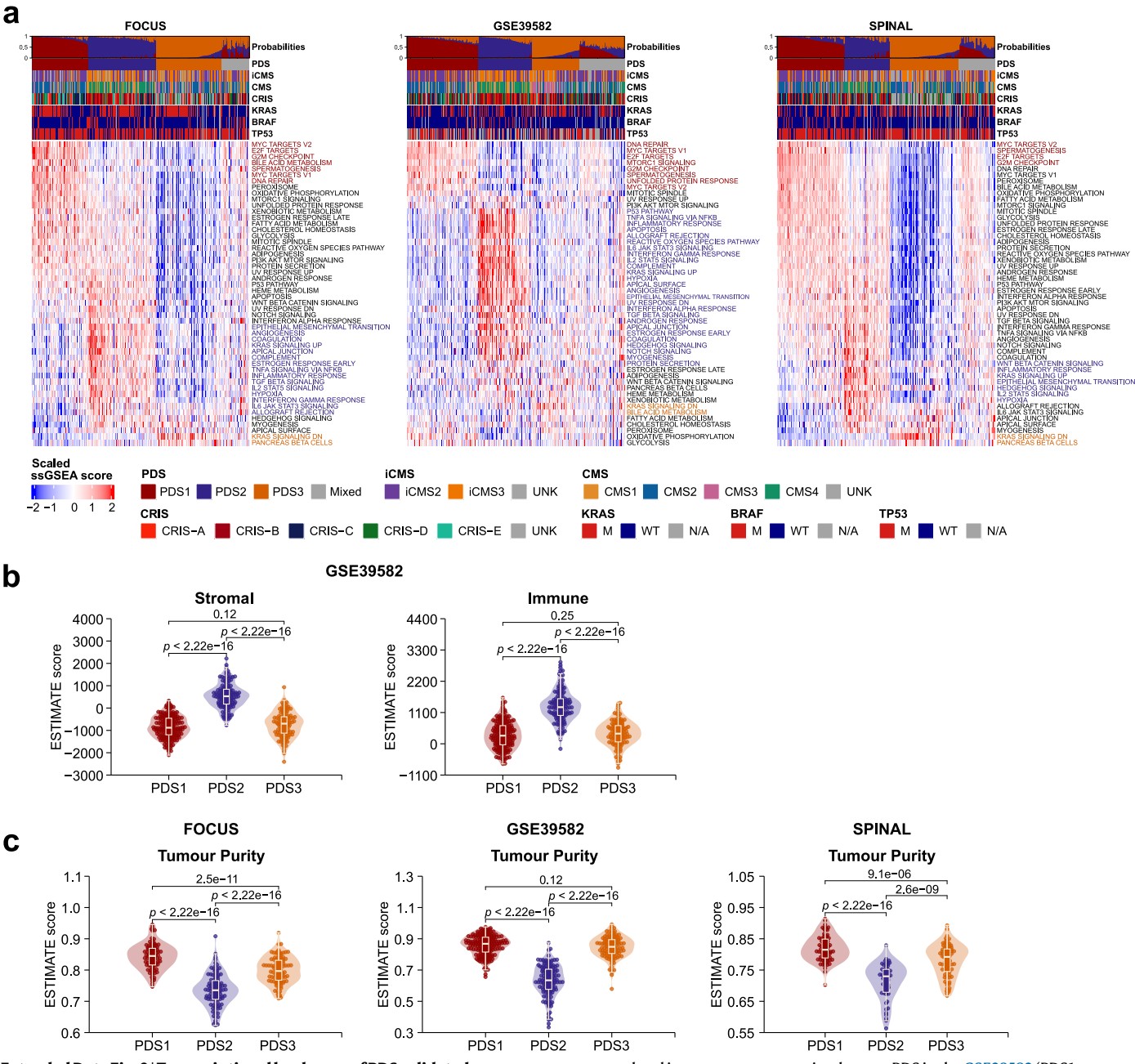

**Extended Data Fig. 3 | Transcriptional landscape of PDS validated across independent CRC cohorts. a**, Heatmaps represent the 'Hallmark' gene sets ssGSEA scores across the FOCUS (*left*), GSE39582 (*centre*) and SPINAL (*right*) cohorts, where the samples are ordered based on PDS with annotated PDS prediction probabilities, PDS, iCMS, CMS, CRIS, and mutational status of *KRAS*, *BRAF* and *TP53*. Upregulated PDS-specific Hallmarks for each independent cohorts are shown in coloured text on the side. **b**, Violin plots display ESTIMATE stromal and immune scores examined across PDS in the GSE39582 (PDS1, $n = 186$; PDS2, $n = 140$; PDS3, $n = 122$) cohort. Boxes within violin plots depict the interquartile range, median, minimum, and maximum value (excluding outliers as dots). *P*-values: two-sided Wilcoxon rank-sum test. **c**, as in **b**, examining ESTIMATE tumour purity measure across PDS in the FOCUS (PDS1, $n = 93$; PDS2, $n = 113$; PDS3, $n = 108$), GSE39582 (numbers as in **b**), and SPINAL (PDS1, $n = 80$; PDS2, $n = 54$; PDS3, $n = 82$) cohorts.

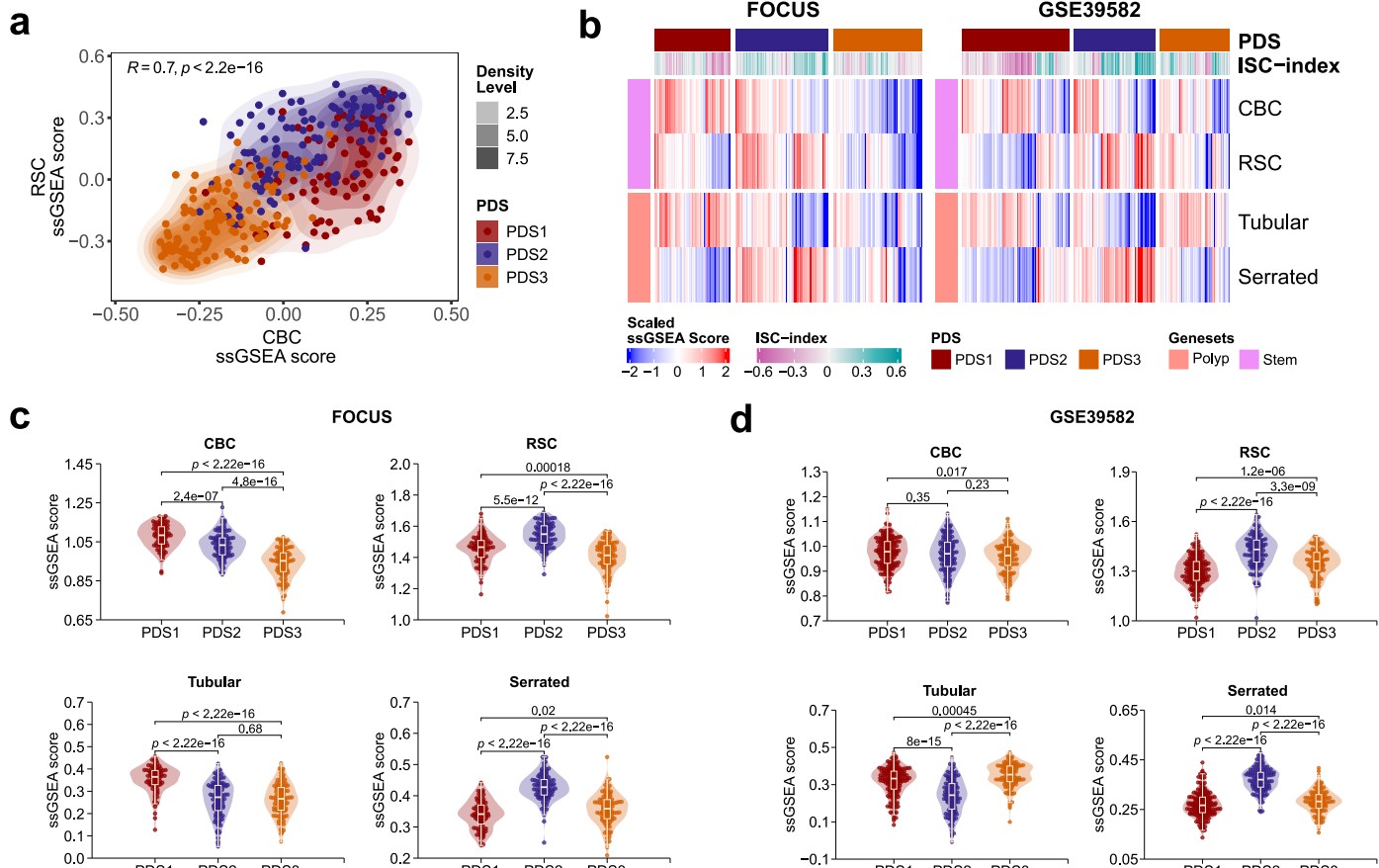

**Extended Data Fig. 4 | Stem and polyp gene signatures interrogation in PDS.**
**a**, Scatter gradient-density plot represents LGR5[+] crypt base columnar (CBC) and
LGR5[-]/ANXA1[+] regenerative stem cells (RSC) ssGSEA scores with annotated PDS
sample in the FOCUS cohort. *P*-value: two-sided Pearson correlation co-efficient.
**b**, Heatmaps depict ssGSEA scores per sample for the stem and polyp signatures
in the FOCUS and GSE39582 cohorts with annotated PDS calls and the Intestinal

Stem Cell Index (ISC-index). **c**, Violin plots compare ssGSEA scores across PDS
for CBC, RSC stem signatures and tubular, serrated precursor polyp signatures
in the FOCUS (PDS1, *n* = 93; PDS2, *n* = 113; PDS3, *n* = 108), and **d**, GSE39582 (PDS1,
*n* = 186; PDS2, *n* = 140; PDS3, *n* = 122) cohort. Boxes within violin plots depict the
interquartile range, median, minimum, and maximum value (excluding outliers
as dots). *P*-values: two-sided Wilcoxon rank-sum test.

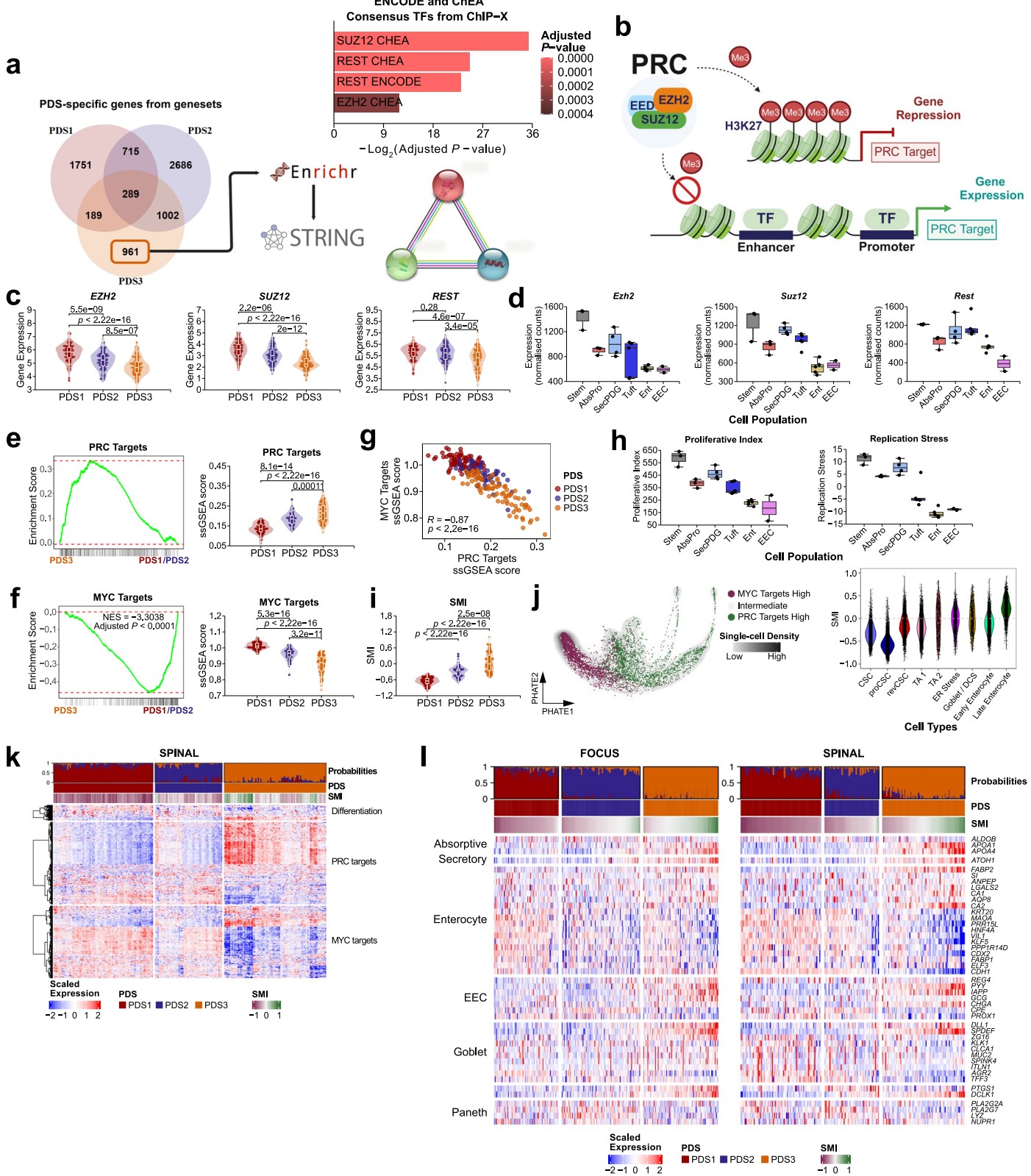

**Extended Data Fig. 5 | See next page for caption.**

**Extended Data Fig. 5 | PDS3 association to PRC Targets led to resemblance to differentiation-like traits. a**, Piechart denotes PDS-specific genes from PDS gene sets, where the PDS3 genes were analysed using Enrichr against the ChEA 2016 database resulting in significant association with SUZ12, REST and EZH2 (*top*), forming protein-protein interaction STRING network (*bottom*). Enrichr: Benjamini-Hochberg adjusted *P*-value. **b**, The genes/proteins show link to polycomb repressive protein complex (PRC) with the core protein involving SUZ12 and EZH2. A simplified schematic describes the role of PRC in marking with trimethylation (Me3) at lysine 27 in histone 3 (H3K27) that leads to repression of the PRC target genes while the PRC target genes are expressed upon the absence of this marker from PRC. **c**, Gene expression measure of *EZH2*, *SUZ12*, and *REST* across PDS in the FOCUS (PDS1, *n* = 93; PDS2, *n* = 113; PDS3, *n* = 108), and **d**, across mouse epithelium cell populations in the GSE143915 cohort. Boxes depict the interquartile range, median, minimum, and maximum value (excluding outliers as dots). *P*-values (in **c**): two-sided Wilcoxon rank-sum test. **e**, GSEA enrichment and violin plot displaying ssGSEA score across PDS for PRC targets, and **f**, MYC targets in the SPINAL (PDS1, *n* = 80; PDS2, *n* = 54; PDS3, *n* = 82) cohort. *P*-values

(violin plots): two-sided Wilcoxon rank-sum test. GSEA: Benjamini-Hochberg adjusted *P*-value, NES = Normalised Enrichment Score. **g**, Correlation between MYC targets and PRC targets with annotated PDS calls in the SPINAL cohort. **h**, Boxplot highlights measure of proliferation index and replication stress across mouse epithelial cell populations (*n*= as in **d**). **i**, Violin plot represents SMI across PDS in the SPINAL cohort (*n*= and statistics as in **f**). **j**, PHATE visualisation of murine organoid-derived scRNA-seq with MYC targets High and PRC targets High cell annotations (*left*), and violin plot with SMI across annotated epithelial cell types (*right*). **k**, Heatmap displays the gene expression of differentiation markers, PRC targets and MYC targets signatures across PDS in the SPINAL cohort, annotated with the PDS prediction probabilities, PDS calls, and SMI. **l**, Heatmap represents gene expressions for the differentiation makers. AbsPro = absorptive progenitor, SecPDG = secretory progenitor/deep crypt secretory cells/goblet, Ent = enterocytes, EEC = enteroendocrine cell, CSC = colonic stem cell, proCSC = hyper-proliferative CSC, revCSC = revival CSC, TA = transit amplifying cell, DSC = deep crypt secretory cell.

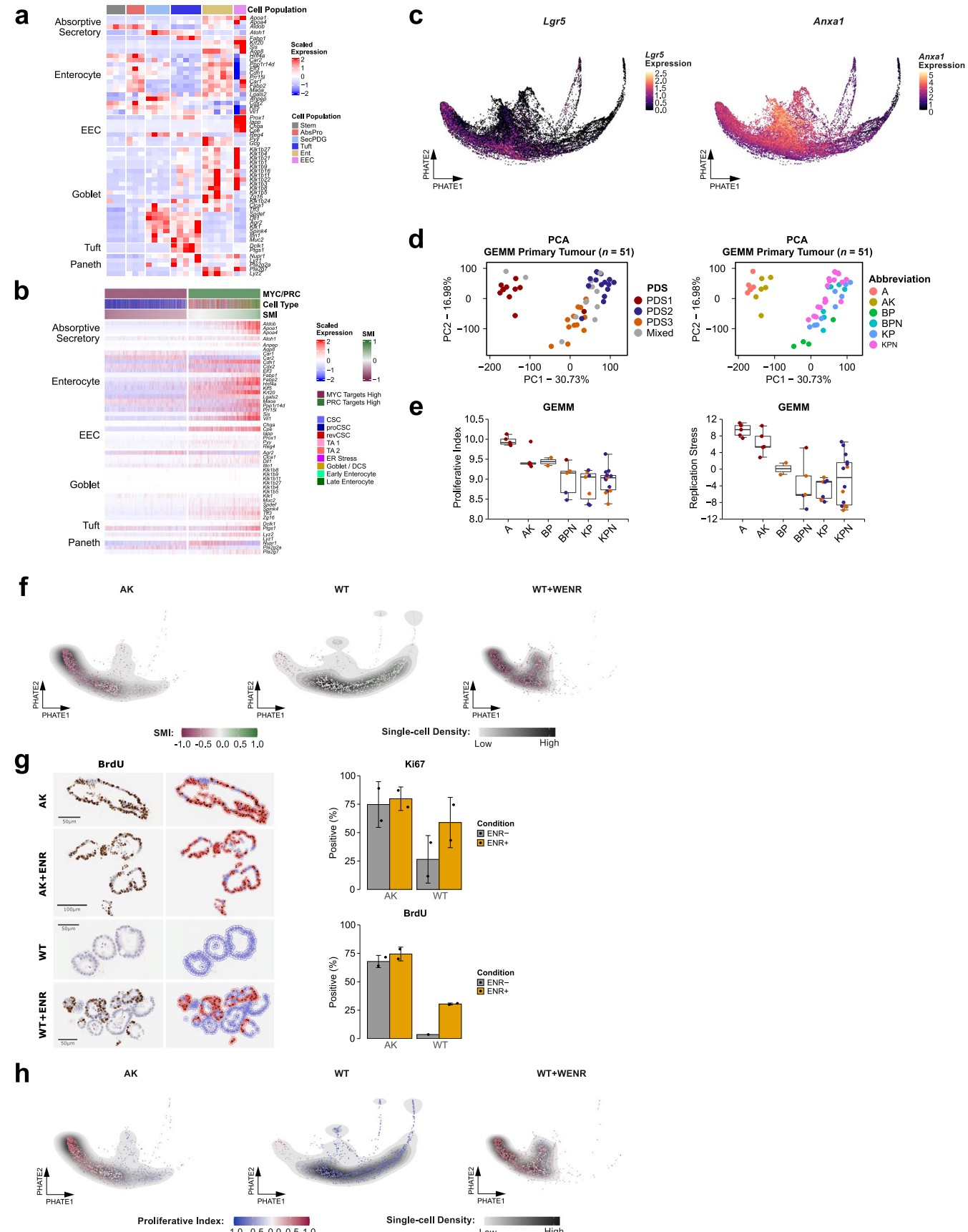

**Extended Data Fig. 6 | See next page for caption.**

**Extended Data Fig. 6 | Caveat of differentiated-like PDS3 in preclinical setting. a**, Heatmap representation of gene expressions for the differentiation makers in the GSE143915 cohort, where the top annotation indicates mouse epithelial cell populations. **b**, Heatmap depicts gene expression for the differentiation markers in the murine organoid-derived scRNA-seq data, where the top annotations display MYC targets High and PRC targets High, epithelial cell types, and SMI. **c**, PHATE visualisations of *Lgr5* (*left*) and *Anxa1* (*right*) expression in the mouse organoid-derived scRNA-seq dataset. **d**, PCA plots on genetically engineered mouse models-derived primary tumour dataset annotated with genotype (*left*) and PDS (*right*). **e**, Boxplots compares proliferative index (*left*)

and replication stress (*right*) across the genotypic models with PDS annotations. **f**, PHATE visualisation on AK (*left*), WT (*centre*) and WT + WENR (*right*) models of murine-derived scRNA-seq data, annotated with SMI and single cell density. **g**, *Right*: Representative BRDU⁺ images of murine organoids cells, cultured with or without EGF, Noggin and R-spondin-1 growth media supplements (AK, AK + ENR, WT + ENR, *n* = 2 replicates each per group; WT, *n* = 1). Scale bars: AK, 50µm; AK + ENR, 100µm; WT, 50µm; WT + ENR, 50µm. *Left*: Barchart displays the positivity (%) of the BRDU⁺ and Ki67⁺ staining, enumerated via QuPath. Data are mean±s.d. **h**, PHATE visualisation annotated with Proliferation Index and single cell density. WENR or ENR: W = WNT3A, E = EGF, N = Noggin, R = R-spondin-1.

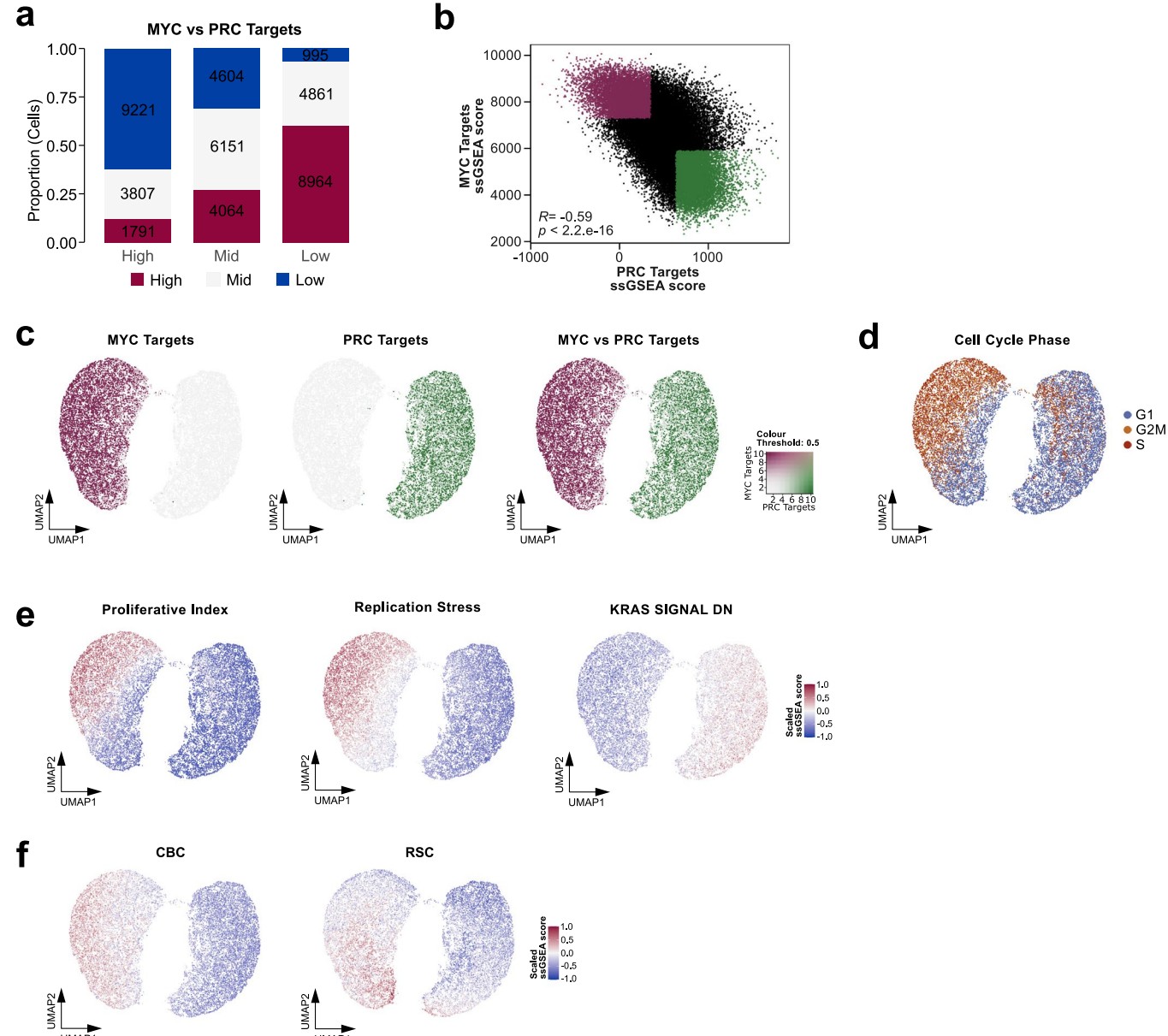

**Extended Data Fig. 7 | MYC targets and PRC targets examined at the single-cell level. a**, Proportion of cells with tertile-based stratification using MYC targets and PRC targets single-cell ssGSEA scores into High, Mid and Low. **b**, Scatterplot shows the correlation between MYC targets and PRC targets. The MYC targets-High and PRC targets-High (top tertiles) highlighted in red and green respectively. *P*-value: two-sided Pearson correlation co-efficient. **c**, UMAP with MYC target-High and PRC target-High cells only. **d**, UMAP with cell cycle phase annotated per cells. **e**, UMAP displays proliferative index, replication stress, KRAS SIGNAL DN and neuroactive ligand receptor interaction. **f**, UMAP visualisation with CBC and RSC ssGSEA scores.

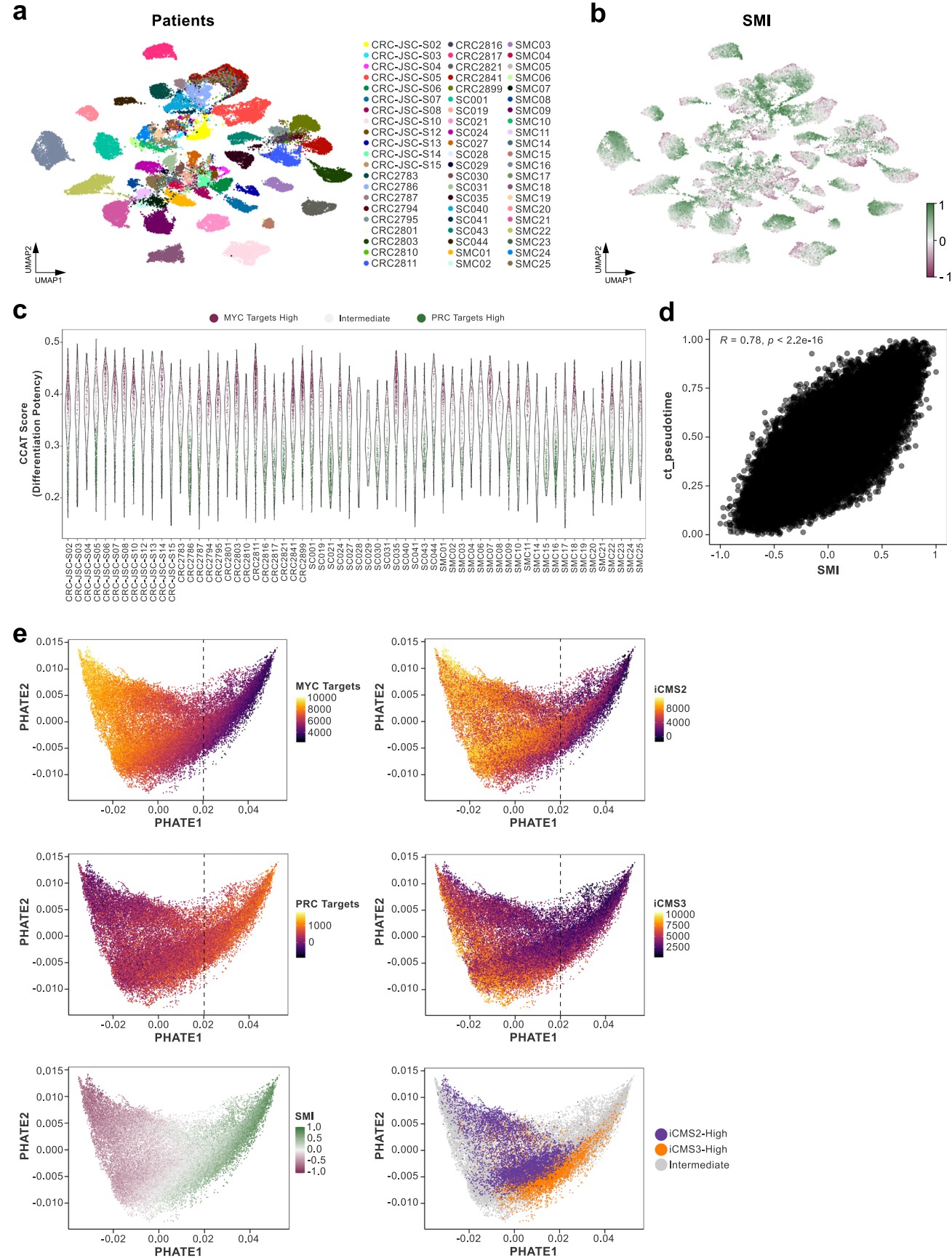

**Extended Data Fig. 8 | See next page for caption.**

**Extended Data Fig. 8 | MYC/PRC highlights nuanced biology previously unidentified and mutually exclusive from iCMS biology. a**, UMAP visualisation of epithelial scRNA-seq dataset from n = 63 CRC patients. **b**, UMAP visualisation with SMI annotations. **c**, Boxplot displaying all the patients assessing differentiation potency per cell with MYC targets-High/PRC targets-High. **d**, Scatterplot shows positive correlation between SMI and pseudotime score.

*P*-value: two-sided Pearson correlation co-efficient. **e**, PHATE visualisation on MYC targets, PRC targets ssGSEA scores and SMI in contrast to iCMS2, iCMS3 ssGSEA scores and iCMS2-High/iCMS3-High (based on tertile split of ssGSEA scores). The vertical dashed black line roughly indicates the MYC/PRC high/low cells defining distinctive differences compared to that of iCMS2/iCMS3.

# Reporting Summary

## Statistics

For all statistical analyses, confirm that the following items are present in the figure legend, table legend, main text, or Methods section.

| n/a | Confirmed | |
|---|---|---|
| ☐ | ☒ | The exact sample size (*n*) for each experimental group/condition, given as a discrete number and unit of measurement |
| ☐ | ☒ | A statement on whether measurements were taken from distinct samples or whether the same sample was measured repeatedly |
| ☐ | ☒ | The statistical test(s) used AND whether they are one- or two-sided *Only common tests should be described solely by name; describe more complex techniques in the Methods section.* |
| ☐ | ☒ | A description of all covariates tested |
| ☐ | ☒ | A description of any assumptions or corrections, such as tests of normality and adjustment for multiple comparisons |
| ☐ | ☒ | A full description of the statistical parameters including central tendency (e.g. means) or other basic estimates (e.g. regression coefficient) AND variation (e.g. standard deviation) or associated estimates of uncertainty (e.g. confidence intervals) |
| ☐ | ☒ | For null hypothesis testing, the test statistic (e.g. *F*, *t*, *r*) with confidence intervals, effect sizes, degrees of freedom and *P* value noted *Give P values as exact values whenever suitable.* |
| ☒ | ☐ | For Bayesian analysis, information on the choice of priors and Markov chain Monte Carlo settings |
| ☒ | ☐ | For hierarchical and complex designs, identification of the appropriate level for tests and full reporting of outcomes |
| ☐ | ☒ | Estimates of effect sizes (e.g. Cohen's *d*, Pearson's *r*), indicating how they were calculated |

*Our web collection on statistics for biologists contains articles on many of the points above.*

## Software and code

Policy information about availability of computer code

Data collection
A combination of both public and proprietary gene expression datasets (both microarray and RNA-Seq) were used for the study. The publicly available gene expression datasets were downloaded and are available at Gene Expression Omnibus (GEO) with the accession number reported in the study as: GSE156915 (FOCUS), GSE39582, and GSE31279 (laser-captured microdissected dataset). RNA-Seq data includes TCGA Colon and Rectal Adenocarcinoma (COREAD) downloaded as HT-Seq counts from Genomic Data Commons (GDC) via TCGAbiolinks R package (v2.16.1); GSE143915 via GEO (mouse intestinal crypt dataset). The genetically engineered mouse model (GEMM) primary tumour dataset can be accessed via the accession code GSE218776 (RNA-Seq). In the case of analyses with the data from PETACC-3 trial (NCT00026273), it was analysed by Dr Petros Tsantoulis and overseen by Prof Sabine Tejpar.

Other proprietary expression dataset (S:CORT consortium) includes: SPINAL (CRC cohort; microarray), and polyp dataset (RNA-Seq) will be disclosed prior to the publication. These datasets will be released in collaboration with a Cancer Research UK data access committee, who are committed to supporting the FAIR principles and to ensure use of these cohorts for academic researchers.

Processed count expression matrices from previously published single cell RNA sequencing data of CRC patient samples (Joanito et al., 2022, Nature Genetics) were requested and downloaded through Synapse (syn26844071). Single cell dataset derived from murine organoid models from Qin and Cardoso Rodriguez et al., 2023, BioRxiv) was downloaded from Zenodo (10.5281/zenodo.7586958). The datasets included in the study will be further made available prior to the publication.

Data analysis
We used R (v4.1.2) and RStudio (v2022.09.2.382), QuPath (v0.2.3) and SideFX Houdini 19.5 has been utilised in the study. All the software versions (including R-related packages) have been provided in the "Method" section.

Statistical analysis conducted in this study has been performed in R using stats (v4.2.1) or ggpubr (v0.4.0) R package for plots, including two-

sided Wilcoxon rank-sum test, Kruskal-Wallis rank-sum test, Fisher's exact test, and Pearson's correlation coefficient test. For copy number by arm analysis, Pearson' Chi-squared test post-hoc analysis was performed using chisq.posthoc.test R package (v0.1.2) and adjusted P-value with Benjamini-Hochberg using p.adjust function in stats R package. Other R packages that have been utilised for data analysis and data visualisation include, ggtern (v3.3.5), ComplexHeatmap (v2.10.0), circlize (v0.4.15), umap (v0.2.8.0), ggplot2 (v3.3.6), patchwork (v1.1.1), riverplot (v0.10), ggforce (v0.3.3), RColorBrewer (v1.1-2).

The PDSclassifier R package (v0.1.0) is available on the Molecular Pathology Lab GitHub (https://github.com/MolecularPathologyLab/PDSclassifier). All new data and code will be made available prior to publication, and all the scripts related to the article will be made available on our website (www.dunne-lab.com).

For manuscripts utilizing custom algorithms or software that are central to the research but not yet described in published literature, software must be made available to editors and reviewers. We strongly encourage code deposition in a community repository (e.g. GitHub). See the Nature Portfolio guidelines for submitting code & software for further information.

# Data

Policy information about availability of data

All manuscripts must include a data availability statement. This statement should provide the following information, where applicable:
- Accession codes, unique identifiers, or web links for publicly available datasets
- A description of any restrictions on data availability
- For clinical datasets or third party data, please ensure that the statement adheres to our policy

In addition to the statement below, we have also included within the Methods section detailing access points for all cohorts used. This is alongside links/references to all methods/scripts used for computational analyses throughout.

Data and Code Availability.
The PDSclassifier R package (v0.1.0) is available on the Molecular Pathology Lab GitHub (https://github.com/MolecularPathologyLab/PDSclassifier). All new data and code will be made available prior to publication, and all the scripts related to the article will be made available on our website (www.dunne-lab.com).

Also, the datasets included in the study will be also further made available prior to the publication.

The FOCUS (GSE156915) and SPINAL datasets were generated within the S:CORT programme, where microarray gene expression profiles, mutation, clinical, immunohistochemistry (IHC), tissue blocks and tumour microarrays (TMAs) were available. FOCUS: MRC-funded randomised trial cohort consisting of 360 formalin-fixed paraffin-embedded (FFPE) primary tumour samples for metastatic CRC. SPINAL: 258 FFPE samples from CRC patients, mixed stages.

The data in this publication generated by the S:CORT Consortium is available for use by not-for-profit organisations for academic, teaching and educational purposes. Gene expression profiles for the S:CORT-led SPINAL has been made available at GEO with GSEXXX. The data is available for commercial use, on commercial terms, via Cancer Research Horizons https://www.cancerresearchhorizons.com/.

Other publicly available datasets were accessed from Gene Expression Omnibus (GEO) with accession number: GSE39582, GSE31279, GSE143915, GSE218776, and from ArrayExpress E-MTAB-6363. The validation of clinical association was carried out PETACC-3 cohort. The Cancer Genome Atlas (TCGA) dataset for Colon and Rectal Adenocarcinoma (COREAD), was accessed and extracted from the Genomic Data Commons (GDC) via TCGAbiolinks.

Two epithelial single-cell RNA sequencing datasets were also utilised for the study – a CRC tissue derived scRNA-seq merged datasets from five different cohorts, and a scRNA-seq dataset derived from murine organoids mono-/co-cultured with fibroblast and/or macrophages. For the scRNA-seq human CRC dataset, the processed count expression matrix for n=49,155 epithelial cells and the corresponding epithelial metadata were downloaded through the Synapse under the accession code syn26844071. The murine organoid scRNA-seq dataset consists of n=29,452 epithelial cells from wild-type mouse colonic organoids and at least 5 different genotypic CRC organoids, including shApc (A), KrasG12D/+ (K), shApc and KrasG12D/+ (AK), KrasG12D/+ and Trp53R172H/- (KP) and shApc, KrasG12D/+ and Trp53R172H/- (AKP), and all the corresponding metadata were also downloaded from Qin and Cardoso Rodriguez et al.

# Research involving human participants, their data, or biological material

Policy information about studies with human participants or human data. See also policy information about sex, gender (identity/presentation), and sexual orientation and race, ethnicity and racism.

| | |
|---|---|
| Reporting on sex and gender | NA |
| Reporting on race, ethnicity, or other socially relevant groupings | NA |
| Population characteristics | NA |
| Recruitment | NA |
| Ethics oversight | NA |

Note that full information on the approval of the study protocol must also be provided in the manuscript.

# Field-specific reporting

Please select the one below that is the best fit for your research. If you are not sure, read the appropriate sections before making your selection.

☒ Life sciences          ☐ Behavioural & social sciences          ☐ Ecological, evolutionary & environmental sciences

For a reference copy of the document with all sections, see nature.com/documents/nr-reporting-summary-flat.pdf

# Life sciences study design

All studies must disclose on these points even when the disclosure is negative.

| | |
|---|---|
| Sample size | All samples sizes are indicated throughout the manuscript. No sample size calculation was performed. |
| Data exclusions | As mentioned in the "Method" section, for class discovery, only KRAS-mutant primary CRC tumours were selected (n=165) with KRAS wildtypes, BRAF-mutants, HRAS-mutants and NRAS-mutants excluded. The rationale being the focus of the study initially was KRAS-mutant stratification, thus exclusion of respective samples. |
| Replication | Following class discovery, a number of cohorts were used to validate/replicate these findings. Details of the cohorts used are in the data/code sections of this report and detailed in methods of the manuscript. The validation was done in at least three different cohorts: FOCUS, GSE39582, and SPINAL with biological findings remaining consistent across all three cohorts. |
| Randomization | NA |
| Blinding | NA |

# Behavioural & social sciences study design

All studies must disclose on these points even when the disclosure is negative.

| | |
|---|---|
| Study description | NA |
| Research sample | NA |
| Sampling strategy | NA |
| Data collection | NA |
| Timing | NA |
| Data exclusions | NA |
| Non-participation | NA |
| Randomization | NA |

# Ecological, evolutionary & environmental sciences study design

All studies must disclose on these points even when the disclosure is negative.

| | |
|---|---|
| Study description | NA |
| Research sample | NA |
| Sampling strategy | NA |
| Data collection | NA |
| Timing and spatial scale | NA |
| Data exclusions | NA |
| Reproducibility | NA |
| Randomization | NA |

| | Blinding | NA |

Did the study involve field work?  ☐ Yes  ☐ No

## Field work, collection and transport

| Field conditions | NA |
| Location | NA |
| Access & import/export | NA |
| Disturbance | NA |

# Reporting for specific materials, systems and methods

We require information from authors about some types of materials, experimental systems and methods used in many studies. Here, indicate whether each material, system or method listed is relevant to your study. If you are not sure if a list item applies to your research, read the appropriate section before selecting a response.

## Materials & experimental systems

| n/a | Involved in the study |
|---|---|
| ☐ | ☒ Antibodies |
| ☒ | ☐ Eukaryotic cell lines |
| ☒ | ☐ Palaeontology and archaeology |
| ☐ | ☒ Animals and other organisms |
| ☐ | ☒ Clinical data |
| ☒ | ☐ Dual use research of concern |
| ☒ | ☐ Plants |

## Methods

| n/a | Involved in the study |
|---|---|
| ☒ | ☐ ChIP-seq |
| ☒ | ☐ Flow cytometry |
| ☒ | ☐ MRI-based neuroimaging |

## Antibodies

| Antibodies used | Ki67, Chromogranin A, Synaptophysin, Hs-ANXA1 (465411), and Hs-LGR5-C2 (311021-C2).<br><br>BrdU (BD Biosciences 347580, TRS High, 1:250), Ki67 (Cell Signalling 12202, ER2 20min, 1:1000), Chromogranin-A (AbCam ab108388, TRS High, 1:600) and Synaptophysin (Cell Signalling 36406, TRS High, 1:150) |
| Validation | Previously developed and validated by Ester Gil Vasquez et al., https://pubmed.ncbi.nlm.nih.gov/35931031/ |

## Eukaryotic cell lines

Policy information about cell lines and Sex and Gender in Research

| Cell line source(s) | NA |
| Authentication | NA |
| Mycoplasma contamination | NA |
| Commonly misidentified lines (See ICLAC register) | NA |

## Palaeontology and Archaeology

| Specimen provenance | NA |
| Specimen deposition | NA |

| Dating methods | NA |

☐ Tick this box to confirm that the raw and calibrated dates are available in the paper or in Supplementary Information.

| Ethics oversight | NA |

Note that full information on the approval of the study protocol must also be provided in the manuscript.

# Animals and other research organisms

Policy information about studies involving animals; ARRIVE guidelines recommended for reporting animal research, and Sex and Gender in Research

| Laboratory animals | All experiments were performed on mice with C57BL/6 background aged between 6-12 weeks. |
| Wild animals | No wild animals were used in the study. |
| Reporting on sex | Mice of both sexes were included. It has been detailed in the "Method" section. |
| Field-collected samples | Field-collected samples were not used in the study. |
| Ethics oversight | All animal experiments were performed according to a UK Home Office licence (Project License 70/8646) and were reviewed by the animal welfare and ethical board of the University of Glasgow. |

Note that full information on the approval of the study protocol must also be provided in the manuscript.

# Clinical data

Policy information about clinical studies
All manuscripts should comply with the ICMJE guidelines for publication of clinical research and a completed CONSORT checklist must be included with all submissions.

| Clinical trial registration | We used a subset of retrospective and anonymous molecular data from the FOCUS and PETACC3 clinical trials. These data were not prospectively used within the trial.<br><br>The FOCUS trial study was previously registered as an International Standard Randomised Controlled Trial, number ISRCTN 79877428. The PETACC3 trial study was previously registered under ClinicalTrials.gov Identifier NCT00026273. |
| Study protocol | FOCUS: https://www.isrctn.com/ISRCTN79877428<br>PETACC3: https://clinicaltrials.gov/ct2/show/NCT00026273 |
| Data collection | NA |
| Outcomes | NA |

# Dual use research of concern

Policy information about dual use research of concern

## Hazards

Could the accidental, deliberate or reckless misuse of agents or technologies generated in the work, or the application of information presented in the manuscript, pose a threat to:

| No | Yes | |
|----|-----|---|
| ☒ | ☐ | Public health |
| ☒ | ☐ | National security |
| ☒ | ☐ | Crops and/or livestock |
| ☒ | ☐ | Ecosystems |
| ☒ | ☐ | Any other significant area |

## Experiments of concern

Does the work involve any of these experiments of concern:

| No | Yes | |
|----|-----|---|
| ☒ | ☐ | Demonstrate how to render a vaccine ineffective |
| ☒ | ☐ | Confer resistance to therapeutically useful antibiotics or antiviral agents |
| ☒ | ☐ | Enhance the virulence of a pathogen or render a nonpathogen virulent |
| ☒ | ☐ | Increase transmissibility of a pathogen |
| ☒ | ☐ | Alter the host range of a pathogen |
| ☒ | ☐ | Enable evasion of diagnostic/detection modalities |
| ☒ | ☐ | Enable the weaponization of a biological agent or toxin |
| ☒ | ☐ | Any other potentially harmful combination of experiments and agents |

# Plants

| | |
|---|---|
| Seed stocks | NA |
| Novel plant genotypes | NA |
| Authentication | NA |

# ChIP-seq

## Data deposition

☐ Confirm that both raw and final processed data have been deposited in a public database such as GEO.

☐ Confirm that you have deposited or provided access to graph files (e.g. BED files) for the called peaks.

| | |
|---|---|
| Data access links<br>*May remain private before publication.* | NA |
| Files in database submission | NA |
| Genome browser session<br>(e.g. UCSC) | NA |

## Methodology

| | |
|---|---|
| Replicates | NA |
| Sequencing depth | NA |
| Antibodies | NA |
| Peak calling parameters | NA |
| Data quality | NA |
| Software | NA |

# Flow Cytometry

## Plots

Confirm that:

☐ The axis labels state the marker and fluorochrome used (e.g. CD4-FITC).

☐ The axis scales are clearly visible. Include numbers along axes only for bottom left plot of group (a 'group' is an analysis of identical markers).

☐ All plots are contour plots with outliers or pseudocolor plots.

☐ A numerical value for number of cells or percentage (with statistics) is provided.

## Methodology

Sample preparation | NA

Instrument | NA

Software | NA

Cell population abundance | NA

Gating strategy | NA

☐ Tick this box to confirm that a figure exemplifying the gating strategy is provided in the Supplementary Information.

# Magnetic resonance imaging

## Experimental design

Design type | NA

Design specifications | NA

Behavioral performance measures | NA

## Acquisition

Imaging type(s) | NA

Field strength | NA

Sequence & imaging parameters | NA

Area of acquisition | NA

Diffusion MRI      ☐ Used      ☐ Not used

## Preprocessing

Preprocessing software | NA

Normalization | NA

Normalization template | NA

Noise and artifact removal | NA

Volume censoring | NA

## Statistical modeling & inference

Model type and settings | NA

Effect(s) tested | NA

Specify type of analysis:      ☐ Whole brain      ☐ ROI-based      ☐ Both

Statistic type for inference | NA

(See Eklund et al. 2016)

Correction | NA

## Models & analysis

| n/a | Involved in the study |
|-----|----------------------|
| ☒ | ☐ Functional and/or effective connectivity |
| ☒ | ☐ Graph analysis |
| ☒ | ☐ Multivariate modeling or predictive analysis |

Functional and/or effective connectivity — NA

Graph analysis — NA

Multivariate modeling and predictive analysis — NA

