## [Peer Review File · Nature Genetics]

Peer Review Information

Manuscript Title: Pathway level subtyping identifies a slow-cycling biological phenotype associated with poor clinical outcomes in colorectal cancer.

Corresponding author name(s): Dr Philip (D) Dunne

Reviewer Comments & Decisions:

Decision Letter, initial version:

9th Feb 2023

Dear Dr Dunne,

I'm so sorry for having taken so long to return this decision to you. Thank you so much for bearing with me.

Your Article entitled "Pathway level subtyping identifies a slow-cycling and transcriptionally lethargic biological phenotype associated with poor clinical outcomes in colon cancer independent of genetics." has now been seen by 3 referees, whose comments are attached. In the light of their advice we have decided that we cannot offer to publish your manuscript in Nature Genetics.

The reviewers agree that your work is well-executed and impressive, they remain unconvinced that your classification system for colorectal cancer represents a serious challenge to existing strategies. While there is some interest in the PDS3 group that you have extracted, the reviewers are of the opinion that this subset requires more validation and characterisation. As such, the manuscript has unfortunately failed to garner the level of support that we would expect from reviewers at this stage.

You might want to consider our sister journal *Nature Communications* as a potential venue for the publication of these results. *Nature Communications* publishes high quality and influential research and across the full spectrum of the natural sciences. More information on the journal, the potential benefits of transfer and a link to transfer your paper, can be found at the bottom of this email. Please note that the editorial team at Nature Communications will consider your manuscript independently of our suggestion to transfer. In the interests of time, I did not consult with my colleagues at Nature Communications to see if they would be interested in sending a revised manuscript back to reviewers but I'd be happy to have that conversation with them if it would be helpful to you.

I am sorry that we cannot be more positive on this occasion but hope that you will find our referees'

comments helpful when preparing your paper for submission elsewhere.

With all best wishes,

Safia Danovi
Editor
Nature Genetics

Referee expertise:

Referee #1: transcriptomics

Referee #2: colorectal cancer incl. evolution and pathology

Referee #3: colorectal cancer

Reviewers' Comments:

Reviewer #1:

Remarks to the Author:

In this paper, Malla et al. present an approach that uses pathway, gene sets and gene ontology, rather than gene-level data, for the discovery of molecular subtypes in Colorectal Cancer. Using this approach, the authors define three unique pathway-derived subtypes (PDS). The authors found these biological and clinical features remain consistent across tumour samples independent of KRAS mutational status.

Overall, I think the study is addressing an important question: whether using individual-level pathway composite expression scores to define tumor subtypes is a better than using individual gene expression levels. The study is rather comprehensive and paper is well written. But there are some important issues that the authors need to address.

- In section "Pathway-level class discovery in KRASmut tumours define unique biological subtypes". Line 145. The authors mentioned "pathway-level single sample scores". How is this defined? It is not trivial to summarize the expression profile of a multi-gene pathway/gene set. In the method section, lines 593-594, the authors mentioned that they used GSVA R package, but the method they selected is ssgsea, not gsva, which is the default of the gsva function. But in line 657, the authors claimed that they used GSVA scores. why they use different score types in the same study, without any explanation?
- Calculating pathway-level score is the most important part of the proposed method. I felt that it is critical that a careful survey of different methods to calculate individual sample-level pathway scores. In addition to GSVA and SSGSEA, the authors should also check several alternative methods to define individual-level pathway/gene set scores include ssGSEA (Barbie et al. 2009), PLAGI (Tomfohr et al. 2005), SLEA (Gunem and Lopez-Bigas 2012), Pathfier (Driero et al. 2013) iPath (Su et al. 2021). These methods are developed using very different ideas. At least, these existing methods should be

acknowledged. Their unique features should be discussed. I think this is the biggest issue for the paper.

- The authors seem to suggest that using pathway-level composite scores is a better choice than using individual gene expression levels to define tumor subtypes. Because obviously, the proposed method is much more complicated to run than using just individual gene-level data to define tumor subtypes. If this is the case, I felt that some comparison is warranted to see what tumor subtypes will emerge is using individual gene expression data. Is that possible? At least, some arguments need to be provided to justify why using this much more complicated approach to define tumor subtypes.

- In section "Image-based H&E classifier for rapid PDS tumour classification." "Standard errors were provided for all AUROC values, why standard error was not provided for F1 scores?"

- For the classification task, the authors should conduct a permutation test as control, to demonstrate that the classification accuracy is not due to overfitting.

- Labels of classification tasks.

Lines 284. PDS1-vs-PDS2-3

Lines 285. PDS2-vs-PDS1-3

Lines 289. PDS3-vs-PDS1-2

But in Figure 4C. the labels are PDS1-vs-ALL, PDS2-vs-ALL, PDS3-vs-ALL. These labels are obviously inconsistent. Please clarify and correct.

- Figure 2C. Since the heatmap to the left is the exact same thing as the left part under KRASmut (red bar) in the heatmap to the right, I suggest only show the FOCUS samples that are not KRASmut in the heatmap to the right.

- Figure 4C. the label for vertical axis is "true PDS calls". I felt that this is misleading. The PDS is defined based on pathway scores summarized from gene expression data. It is defined by the authors. I would rather call this "expression-based PDS calls" instead of "true ...".

- Lines 289-290. "This result suggests that PDS3 has an intermediate morphology". This seems to suggest that PDS3 subtype tumor show characteristics in between PDS1 and PDS3. However, from the violin plots shown in Figure 3, most show a monotone increasing or decreasing trend for the 3 subtypes. Does that somehow contradict the observation on morphology?

- Line 593, 595. What is "ontology score"? I have never heard of it. A reference is needed at least. I think the authors means "pathway score" instead, or better yet, sample-level pathway composite score. Please clarify.

- Lines 657-658. "... followed by the total sum of the GSVA scores per sample across gene sets, as previously described⁴⁶." Why calculate the sum of GSVA scores across gene sets? I don't understand.

- In discussion, it would be interesting to discuss potentials of using individual-level pathway scores instead of single gene expression as biomarkers to predict clinical prognostics. This seems natural after the authors demonstrate the success of using pathway scores for unsupervised clustering.

Reviewer #2:

Remarks to the Author:

The manuscript by Malla et al describes a new method that classifies human CRCs into three approximately equal sized groups (PDS1, 2, 3) based on pathway classifications of gene expression data. The effort is compared with the prior CMS classes, where roughly CMS1/4 correspond to PDS2, and CMS2 and CMS3 tumors are split across PDS 1 and 3. They validate the algorithm across more than one dataset, and show that mouse CRC models can be classified as PDS1 or 2, but PDS3 is poorly represented. Strengths are that the work appears to be performed well, and that the PDS3 group appears to identify a relatively unique group of CRCs with poorer outcomes and neural-like traits, and a less mitotic ("lethargic") phenotype.

Uncertain is the value of this classification system. Similar to CMS, it uses expression data to classify CRCs. However, intratumoral heterogeneity of expression patterns is often present in CRCs, and some multiregional studies (Nature volume 611, pages744–753 (2022)) have found different tumor regions have different CMS profiles in most CRCs, albeit some studies (Nature Genetics, vol54, 963, 2022) have found little overlap between tumors. This potential intratumoral heterogeneity problem could be address more.

Of more concern is the potential impact of the PDS classifier. Clinical utility is one measure of impact, and the manuscript describes survival differences with GSE3958 and PETACC3 (Fig 2K). But these cohorts were published in 2013 and 2009 and may not represent more modern treatments, especially immunotherapy for MSI-H CRCs. Microsatellite instability is a major CRC biomarker and is broadly classified as PDS2 (CMS1), but is not well addressed.

Overall this is an impressive amount of work that demonstrates a new CRC classification method. Potential addressable weaknesses are problems with biological intratumoral expression heterogeneity and uncertain clinical impact or utility in a more modern therapeutic setting.

Reviewer #3:

Remarks to the Author:

Malla et al. propose a novel bulk RNAseq based classification system for colorectal cancer, utilizing previously established signaling pathway associated genesets to inform classification of colorectal cancer, and then extrapolate this to various cancer types. This method purportedly improves on current classification approaches (CMS, CRIS) which use bulk genomic and transcriptomic data to inform their classification system. The authors suggest that this PDS classification system is a better predictor of tumor development and treatment response than current methods, as it can classify tumors into biologically discrete groups regardless of mutational status. Overall, the approach of using shared signaling programs to develop classifiers is interesting, but in this manuscript no new biology-driven classifiers are proposed. The authors use generic hallmark and other genesets and tell us how these pathways are differentially expressed in distinct tumors. This appears to be a very blunt approach and as such it is not surprising that using generic genesets yields a rather blunt classification into 3 subgroups, which seem to largely reflect extent of tumor/stromal representation in the tumor (PDS1), cell cycle (PDS2) or the absence of either (PDS3). Given the generic nature of the genesets used, it is also not surprising that the classifiers developed in CRC apply broadly to multiple cancer

types. The study would be of greater interest if more granular data, such as those obtained from clinically annotated single cell gene sets, could be used to derive de novo factors/gene sets, and then tested across bulk cohorts.

The use of transcriptome based molecular subtyping in CRC has failed to be adopted in clinical practice because existing classifiers are too blunt, and also because some studies using them, have had contradictory results in terms of their utility in association with prognosis – this is particularly true for CMS4. A major confounder in these approaches has been the confounding of signals from tumor vs stroma, and the imprecise nature of biopsy sampling of tumor/stroma in clinical samples. Given the blunt nature of the classifiers developed here, I do not think this study represents a significant, clinically actionable advance over existing classifiers. Further, the study primarily performs transcriptomic analyses with a small amount of IF validation. Some of the more interesting hypotheses proposed, e.g. the epigenetic regulation of the neuroendocrine state, require and lack experimental validation. For these reasons, I do not see this study as a strong candidate for publication in Nature Genetics.

However, perhaps the most intriguing part of the study is the data showing lack of representation of specific PDS programs in PDX and GEMM models. This observation is also confounded by potential differences in tumor/stroma composition between these cohorts. Ideally, a rigorous comparison would involve first profiling patient tissue, deriving PDX from the tissue and then profiling the PDX to allow direct comparison of tumor and PDX from the same patient, ideally at single cell transcriptomic and protein IF levels. If such rigorous validation could be performed, this may provide robust evidence for which cell states are and are not retained in ex vivo or cognate GEMM models, in turn informing the development of future improved models and influencing the choice of appropriate preclinical models for drug testing or mechanistic analysis.

Additional Points:

- o Figure 2K. Authors show KM plots for the non-randomized GSE39582 cohort and PETACC-3 cohort based on PDS classification system compared to CMS subtypes substratified by PDS subtypes. The authors should also show KM plots for CMS1, CMS2, CMS3, CMS4 (classification system they are comparing to in this instance).
- o The authors claim that “ Although the tumours employed throughout our study are pathologically-confirmed adenocarcinomas and not neuroendocrine tumours, we do see elevation of separate EEC and neuron cell transcriptional signatures in PDS3 tumours in our bulk tumour data (Figure 6L)” is an interesting observation. They go on to show a representative image of a PDS3 sample of the SPINAL cohort stained with SYN (Figure 6N), despite describing staining of “CRGA” in the text. The authors should correct this and also show representative images of EEC and neuroendocrine markers to make their point. Overall, the authors show biology within PDS3/CMS2 tumors, displaying neural traits, which had not been previously defined despite attempts to subtype this large tumor subtype as defined by the CMS system.
- o The authors explain that the GEMMS and PDX tumors did not align well with the PDS3 subtype, associated with worse prognosis across the 3 PDS subtypes but the authors make a compelling argument as to the limitations of PDX models which select for fast growing cells and may not capture the heterogeneity of PDS3 patient tumors which are slow growing. However, a significant portion of CRC tumors as defined by the PDS classification system in the FOCUS cohort are of the PDS3 subtype (30%), suggesting a limitation in the claims of the potential of this classification system to improve upon our understanding of treatment outcomes.
- o The authors claim that the PDS3 subtype contains a “senescence-like physiology” but do not adequately characterize this in the manuscript.
- o The authors assessed the clinical landscape of the FOCUS cohort to explore significant differences in PDS classification of tumors such as gender or sidedness, but do not comment on other clinical factors

such diagnosis age, race/ethnicity, metastatic disease, MMR status, treatment before surgery. (Figure 2, Supplemental 2).

o Typo in legend of Figure 2K ("CMS2/PDS1, CMS2/PDS1 vs all CMS4 samples only (right).") Should be changed to "CMS2/PDS1, CMS2/PDS3 vs all CMS4"

o "Finally, using whole face tissue sections from our SPINAL cohort, we observed that protein expression for the differentiated EEC marker, such as chromogranin A (CRGA)..." This is a typo and should be changed to "CHGA"

Decision Letter, Appeal:

24th Aug 2023

Dear Dr Dunne,

Thank you for asking us to reconsider our decision on your manuscript "Pathway level subtyping identifies a slow-cycling biological phenotype associated with poor clinical outcomes in colon cancer". I have now discussed the points of your letter with my colleagues, and we are willing to send your revised manuscript back out to your original reviewers.

As discussed previously, we have a high bar for appeals and while we appreciate your additional analyses, please note that we will require wholesale support from your reviewers to proceed to the next step. If said support is not forthcoming, we will - of course - be flag to assist you with finding an alternative home for the manuscript in the Nature family.

When preparing a revision, please ensure that it fully complies with our editorial requirements for format and style; details can be found in the Guide to Authors on our website (<http://www.nature.com/ng/>).

Please be sure that your manuscript is accompanied by a separate letter detailing the changes you have made and your response to the points raised. At this stage we will need you to upload:

1) a copy of the manuscript in MS Word .docx format.

2) The Editorial Policy Checklist:

<https://www.nature.com/documents/nr-editorial-policy-checklist.pdf>

3) The Reporting Summary:

(Here you can read about the role of the Reporting Summary in reproducible science:

<https://www.nature.com/news/announcement-towards-greater-reproducibility-for-life-sciences-research-in-nature-1.22062>)

Please use the link below to be taken directly to the site and view and revise your manuscript:

[redacted]

With kind wishes,

Safia Danovi

Author Rebuttal to Initial comments

**Reviewer #1:**

*Remarks to the Author:*

*In this paper, Malla et al. present an approach that uses pathway, gene sets and gene*
*ontology, rather than gene-level data, for the discovery of molecular subtypes in Colorectal*
*Cancer. Using this approach, the authors define three unique pathway-derived subtypes*
*(PDS). The authors found these biological and clinical features remain consistent across*
*tumour samples independent of KRAS mutational status.*

*Overall, I think the study is addressing an important question: whether using individual-level*
*pathway composite expression scores to define tumor subtypes is a better than using*
*individual gene expression levels. The study is rather comprehensive and paper is well written*

*We thank the Reviewer for their positive comments on the importance of the topics we address*
*in this work, and the comprehensive nature of our study.*

*There are some important issues that the authors need to address. In section “Pathway-level*
*class discovery in KRASmut tumours define unique biological subtypes”. Line 145. The*
*authors mentioned “pathway-level single sample scores”. How is this defined? It is not trivial*
*to summarize the expression profile of a multi-gene pathway/gene set. In the method section,*
*lines 593-594, the authors mentioned that they used GSVA R package, but the method they*
*selected is ssgsea, not gsva, which is the default of the gsva function. But in line 657, the*
*authors claimed that they used GSVA scores. why they use different score types in the same*
*study, without any explanation?*

*The Reviewer makes an excellent point on making sure we clearly define what we mean when*
*we state “pathway-level single sample scores”, therefore we have taken a number of actions*
*to address this point.*

*1) We have changed the text throughout from “pathway-level single sample scores” to “single*
*sample gene set enrichment analysis (ssGSEA) scores”. We hope that this change enables*
*the reader to see how these analyses remain consistent, and the text is now more direct on*
*what was used as the method to generate the scores.*

*2) In addition to these textual changes, we also add the following details for the Reviewer:*

For the entirety of the class discovery and classifier development, we make use of ssGSEA
scores as our “standard” method to interrogate the pathways/gene sets, and therefore the
characterisations that we have performed within this study have used the ssGSEA method
from the GSVA R package.

3) As the Reviewer correctly identifies, in Line 657, our standard ssGSEA method is not
applied when we enumerate the ‘**replication stress**’ signature, as this approach utilises a
method developed specifically for scoring replication stress from Dreyer *et al.*, 2021,
Gastroenterology, which uses the GSVA method. As such, the replication stress method
previously described by Dreyer et al, calculates a composite score for replication stress in a
way that is different to the ssGSEA scores we have otherwise used throughout.

4) To directly address the Reviewer’s comment on use of the two different methods, and to
ensure consistency in our data presented throughout, we utilised both ssGSEA and GSVA
methods to calculate replication stress, which shows that, for this particular signature
regardless of the methods used the replication stress is quite consistent as evidenced by a
strong positive correlation between the two (**Figure R1**).

**Figure R1:** Replications stress scores has been examined in three different datasets used in
 the study (FOCUS, GSE39582, SPINAL) across PDS via GSVA and ssGSEA method in GSVA
 R package, presented in boxplots. Replication stress scores via GSVA and ssGSEA both are
 highly correlated ($R > 0.9$). **Shown for review only.**

*Calculating pathway-level score is the most important part of the proposed method. I felt that*
 *it is critical that a careful survey of different methods to calculate individual sample-level*
 *pathway scores. In addition to GSVA and SSGSEA, the authors should also check several*
 *alternative methods to define individual-level pathway/gene set scores include ssGSEA*
 *(Barbie et al. 2009), PLAGL (Tomfohr et al. 2005), SLEA (Gunem and Lopez-Bigas 2012),*
 *Pathfier (Droier et al. 2013) iPath (Su et al. 2021). These methods are developed using very*

different ideas. At least, these existing methods should be acknowledged. Their unique
 features should be discussed. I think this is the biggest issue for the paper.

The Reviewer highlights the extensive selection of methods that exist for performing pathway-
 level scoring of transcriptional data, as while the approaches we have taken in this study reflect
 those methods most used within our co-authorship and wider networks, it is worth checking
 the outputs derived from each.

To address this, we selected the Hallmarks signatures that were significantly associated with
 PDS groups when using ssGSEA, and here we performed a re-assessment of them using the
 GSVA, PLAGE, SINGSCORE and iPATH methods (Figure R2). As the Reviewer indicates,
 these methods are developed in very different ways, and while we find overarching similarities
 between outputs using ssGSEA, GSVA and SINGSCORE, alongside similar yet weaker
 pattern strengths with iPATH, these analyses highlight the clearly distinct outputs derived when
 using PLAGE.

Numerous studies have assessed the performance of GSEA-like methodologies in terms of
 their statistical outputs (Tarca *et al.*, PLOS ONE, 2013). However, such studies have rarely
 focused on how different biological interpretations can be derived from assessing the same
 signatures, in the same samples, using these seemingly complementary methods. Inspired by
 the Reviewers comments here, and in order to give the point the detail and focus required to
 fully explore it, we have initiated a follow-up study to test how these variations in the statistical
 and biological outputs can change the clinical associations from tumour cohorts and also the
 end-user's interpretation.

**FIGURE R2:** Heatmap visualisations of PDS significant Hallmarks gene set scores utilising
 GSVa, PLAGE, SINGSCORE, and iPATH methods on three different cohorts: FOCUS (*left-*
 *column*), GSE39582 (*middle-column*) and SPINAL (*right-column*), annotated with PDS
 prediction probabilities and PDS calls. **Shown for review only.**

*The authors seem to suggest that using pathway-level composite scores is a better choice*
 *than using individual gene expression levels to define tumor subtypes. Because obviously, the*
 *proposed method is much more complicated to run than using just individual gene-level data*
 *to define tumor subtypes. If this is the case, I felt that some comparison is warranted to see*
 *what tumor subtypes will emerge is using individual gene expression data. Is that possible?*
 *At least, some arguments need to be provided to justify why using this much more complicated*
 *approach to define tumor subtypes.*

Furthermore, the Reviewer asks for a **comparison of our pathway-level classifier to a**
 **gene-level one**, and what subtypes might have emerged if we used gene-level classification.

The gene-level approach is the most widely used method, as exemplified by the original
 Nature Medicine CMS study by Guinney et al., and the recent Joanito et al., iCMS study.
 Therefore, to address the Reviewers points above, we now compare the outputs from our PDS
 approach to these gene-level classification approaches throughout the revised manuscript. In
 addition to these direct comparisons in the data presented, we have also now included
 additional text in the Introduction and Discussion in regard to comparisons and
 complementarity between these approaches.

The point about which method is best (gene-level v pathway-level) is also extensively covered
 in response to point 1 from Reviewer 3 below.

*In section “Image-based H&E classifier for rapid PDS tumour classification.” Standard errors*
*were provided for all AUROC values, why standard error was not provided for F1 scores?*

The absence of standard error for the F1-score is due to the fact that the reported AUROC
scores and F1-scores are summarizing different aspects of the results of the image-based
PDS experiments.

- • For the AUROC scores we reported the mean and standard deviation of single AUROCs
for each of the five dataset folds (as visualised in previous submission Figure 4B, now
Figure 3C): this metric informs about the stability of performance of the trained classifiers
over different dataset folds.
- • We reported the F1-scores that summarize parts of the single confusion matrix presented
in Figure 3D (previously in Figure 4C). This confusion matrix aggregates the classification
results of the five test sets by five different trained classifiers: this metric informs about the
classification performance of impPDS overall when considering the whole dataset
independently of the five folds.

*For the classification task, the authors should conduct a permutation test as control, to*
*demonstrate that the classification accuracy is not due to overfitting.*

Being wary of this, we had initially performed cross-validation (LOOCV) on train set + test set
was set aside to prevent overfitting during classifier development. However, we thank the
Reviewer for the opportunity to provide more detail and now provide a simple permutation test
as a control as requested.

These analyses demonstrate that the distributions of results from the permutation test
(n=10,000) accuracy calculations for the PDS classification on the test data were well below
the true accuracy of 0.9, highlighting the robustness of the classifier. These data give a
stronger assurance that the classification is not due to overfitting from the training data (**Figure**
**R3**).

**Figure R3:** Histogram shows Permutation Test Distribution (n=10,000) of the accuracies on
 the test data with true accuracy to be around 0.9. **Shown for review only.**

*Labels of classification tasks. Lines 284. PDS1-vs-PDS2-3, Lines 285. PDS2-vs-PDS1-3,*
 *Lines 289. PDS3-vs-PDS1-2. But in Figure 4C. the labels are PDS1-vs-ALL, PDS2-vs-ALL,*
 *PDS3-vs-ALL. These labels are obviously inconsistent. Please clarify and correct.*

*We thank the Reviewer for this point, which related to the different output labels used initially*
 *for calls between the gene-expression PDS v the image based PDS. To avoid confusion, we*
 *have removed these labels and made these results more consistent with the text.*

*Figure 2C. Since the heatmap to the left is the exact same thing as the left part under*
 *KRASmut (red bar) in the heatmap to the right, I suggest only show the FOCUS samples that*
 *are not KRASmut in the heatmap to the right.*

*We appreciate the Reviewer's suggestion here, and for the opportunity to clarify why we chose*
 *to display both sets of heatmaps in the way that we have. The heatmaps in Figure 2B of the*
 *revised manuscript (originally Figure 2C that the Reviewer was referring to) contained all*
 *samples on the left, followed by an ordering of the samples into KRASmut and KRASwt on*
 *the right. It is precisely the identical nature of their appearances that we wish to highlight in*
 *this figure, which we hope give emphasis on how KRAS mutational status does not sway the*
 *biological patterns that emerged from the PDS specific gene set feature selection for the*
 *classification (Figure R4).*

*These may ultimately contain the same data overall, however we feel that the presentation of*
 *the data in these different ways gives the reader a clear indication of how the biological*
 *patterns are independent of mutational status of KRAS (in Figure 2B) and almost all other*
 *mutational data we assessed (Figure 2E).*

**Figure R4:** Heatmap visualisations of the PDS-specific gene sets (features for the classifier)
 ssGSEA scores, annotated with PDS calls, CMS and KRAS mutational status. Heatmap (*left*)
 highlights the PDS-specific patterns on the FOCUS cohort, whereas the same biological
 patterns can be visualised on the heatmap (*right*) arranged based on KRAS mutation status.

*Figure 4C. the label for vertical axis is “true PDS calls”. I felt that this is misleading. The PDS*
 *is defined based on pathway scores summarized from gene expression data. It is defined by*
 *the authors. I would rather call this “expression-based PDS calls” instead of “true ...”.*

*We agree and have changed this label to avoid confusion.*

*Lines 289-290. “This result suggests that PDS3 has an intermediate morphology”. This seems*
 *to suggest that PDS3 subtype tumor show characteristics in between PDS1 and PDS3.*
 *However, from the violin plots shown in Figure 3, most show a monotone increasing or*
 *decreasing trend for the 3 subtypes. Does that somehow contradict the observation on*
 *morphology?*

*In line 289-290 of the initial submission, we were referring to the results from the image-based*
 *PDS classifier, which highlighted that both PDS1 and PDS2 had their own unique histological*
 *features, the neural network approach used failed to find distinct features for PDS3, indicating*
 *that these tumours displayed some shared features with both PDS1 and PDS2.*

*The detail related to this point was questioned further by Reviewer 2 and Reviewer 3, and as*
 *such we have performed a more detailed series of investigations on how the PDS*
 *transcriptional classification relates to morphological features. These analyses are detailed*
 *more extensively in the revised manuscript and in response to the other Reviewer’s points*
 *below, we summarise them here. We performed a new series of investigations using:*

- 1. The epithelial v stromal origin of genes within the gene sets that underpin each PDS class.
2. The histological features that are visually evident in sample-matched H&Es that are
classified by PDS using the gene expression classifier.
3. Enumeration of histological features using previously defined gene expression signatures
for enumerating tumour microenvironmental populations.
4. Testing the consequences of increasing transcriptional threshold of the PDS classifier, by
measuring transcriptional heterogeneity of sample classification and clinical correlates at
each threshold.

These are all now presented in a fully revised Figure 3 in the main manuscript.

*Line 593, 595. What is “ontology score”? I have never heard of it. A reference is needed at*
*least. I think the authors means “pathway score” instead, or better yet, sample-level pathway*
*composite score. Please clarify.*

The Reviewer has made good point and we agree that it is important to keep the terminologies
consistent, and so as we have mentioned above in our first response to this Reviewer, we
have changed this to ssGSEA score throughout.

*Lines 657-658. “.. followed by the total sum of the GSVA scores per sample across gene sets,*
*as previously described⁴⁶.” Why calculate the sum of GSVA scores across gene sets? I don’t*
*understand.*

The Reviewer’s comment has been addressed above and relates to the nature of the
“replication stress” assessment.

*In discussion, it would be interesting to discuss potentials of using individual-level pathway*
*scores instead of single gene expression as biomarkers to predict clinical prognostics. This*
*seems natural after the authors demonstrate the success of using pathway scores for*
*unsupervised clustering.*

We have now included an expanded section on the clinical and translational value of our
approach. This point has also been bolstered by responses to a comment from Reviewer 2 in
regard to MSI v MSS, and also extensively to Reviewer 3 where we have expanded our
investigations into the biologies underpinning PDS classification and how this compares to
established classifiers in this space.

**Reviewer #2:**256 *Remarks to the Author:*

*The manuscript by Malla et al describes a new method that classifies human CRCs into three*
*approximately equal sized groups (PDS1, 2, 3) based on pathway classifications of gene*
*expression data. The effort is compared with the prior CMS classes, where roughly CMS1/4*
*correspond to PDS2, and CMS2 and CMS3 tumors are split across PDS 1 and 3. They validate*
*the algorithm across more than one dataset, and show that mouse CRC models can be*
*classified as PDS1 or 2, but PDS3 is poorly represented. Strengths are that the work appears*
*to be performed well, and that the PDS3 group appears to identify a relatively unique group of*
*CRCs with poorer outcomes and neural-like traits, and a less mitotic (“lethargic”) phenotype.*

*We thank the Reviewer for their comments on the well performed nature of our submission*
*and the unique nature of its findings.*

*Uncertain is the value of this classification system. Similar to CMS, it uses expression data to*
*classify CRCs. However, intratumoral heterogeneity of expression patterns is often present in*
*CRCs, and some multiregional studies (Nature volume 611, pages744–753 (2022)) have*
*found different tumor regions have different CMS profiles in most CRCs, albeit some studies*
*(Nature Genetics, vol54, 963, 2022) have found little overlap between tumors. This potential*
*intratumoral heterogeneity problem could be address more.*

*The Reviewer’s point has given us the opportunity to delve further into the signalling versus*
*morphological intratumoural heterogeneity (ITH); a topic that has been a particular focus of*
*our research over a number of years. The development of the PDS approach presented here*
*has been undertaken in the context of this previous work. Previously, we have assessed the*
*consequences of ITH within gene-level classifiers like CMS (Dunne et al., CCR 2016) and*
*ways to avoid it in classifier development (Dunne et al., Nat Comms 2017). More recently, we*
*measured the potential for ITH across widely used transcriptional signatures (Fisher et al.,*
*CCR 2022), including the collections used in the development of the PDS classifier package.*
*We have developed a web application that allows users to measure the potential for ITH to*
*confound genes or transcriptional signature, accessible via <https://confoundr.qub.ac.uk/>.*

*To address the Reviewers point, we tested the presence and extent of heterogeneity observed*
*in the outputs from both the gene expression-based PDS and the image-based PDS (imPDS)*
*classification approaches.*

*Using single sample probability scores as an indicator of ITH, we can clearly observe how the*
*PDS classifier, when set at a threshold of 0.6 as default, consistently stratifies tumours into*
*PDS classes with homogeneous transcriptional features (Figure R5 - Top). However, in*

contrast, when matched H&E images from the same tumours are assessed by our impDS
 deep learning model, the histologically heterogeneous nature of these tumours is clearly
 shown in the impDS probability scores (**Figure R5 - Bottom**). As such, our PDS classification
 can identify tumour that are transcriptionally homogenous yet histologically heterogeneous.

This visual output for transcriptional probability scores has been included within the revised
 PDS classifier package, <https://github.com/sidmall/PDSclassifier>, to allow users to visualise
 the heterogeneity underpinning PDS classification in their data.

**Figure R5:** Density plots explore the PDS prediction probability scores of the samples for
 PDS1, PDS2 and PDS3 in the FOCUS (*top*) and SPINAL (*bottom*) cohorts. The barcharts
 shows PDS prediction probability scores per samples (*top*) and imPDS prediction probability
 scores per slides (*bottom*).

The probability scores for imPDS suggest the absence of distinct morphological features that
 can distinguish the PDS classes. Therefore, using the image data, we next assessed in more
 detail the features that are most associated with PDS classes within the imPDS model. These
 analyses revealed that while PDS1 tumours were associated with epithelial features and PDS2
 tumours associated with stromal features, these same features were apparent across all PDS3
 tumours leading to poor classification from a histological point of view for this specific class
 (Figure R6).

**Figure R6:** Development of image-based PDS (imPDS) classification emphasised the distinct
histomorphological features that encompasses epithelial-rich PDS1 and stroma-rich PDS2
with high predictability from H&E. But imPDS fails to define features for PDS3, with PDS3
samples (four different PDS3 samples shown above) displaying both epithelial and stroma-
rich histological features. We have included some additional images here for review only, to
complement the images in the main figure.

Furthermore, to address the Reviewers general point on the “value of the classification
system”, we believe that as different end-users may have different end goals for tumour
classification, have developed a series of new data that showcases the adjustable threshold
parameter we have embedded within the PDS classifier package.

The current default threshold for PDS classification has been set as 0.6, meaning that any
tumour with a probability for a specific class that sits above this threshold will be given a
classification label for that class; for example, a sample with a probability score of >0.6 for
PDS1 will be classed as PDS1 regardless of what the other <0.4 probability scores are
attributed to. Having a lower threshold will result in fewer unclassified/mixed samples being
returned; something which some users may see as value, in that they can classify all their
samples. Our PDS default threshold is already more stringent than the 0.5 thresholds applied
for CMS and CRIS classification (although we acknowledge these are built using different
classification models).

While we have used such a threshold for the translational study presented here, we also
acknowledge that samples with higher probabilities for an individual class will be more
enriched for the biologies underpinning each PDS class; for example, samples with higher
probability for any class appear to display higher expression of the biologies/signatures that
characterise that class (**Figure R7**). These probability scores are included in Figure 2G and
associated supplementary figures.

**Figure R7:** Heatmap visualisations represents the significant PDS-specific Hallmarks gene
 set ssGSEA score across PDS samples in the FOCUS and GSE39582 cohorts, annotated
 with PDS prediction probabilities and PDS calls.

In some cases, however, we also believe that a user may value only selecting cases with the
 highest probability, and therefore highest expression of the biologies underpinning each PDS
 class. The cost of this additional biological homogeneity and signalling strength will be a
 concurrent decrease in positive calls in each class overall and an increase in cases labelled
 as 'mixed'.

To address the Reviewers general point, we developed a series of new data for this revised
 manuscript that showcases two such end users; those that value fewer mixed cases (0.6
 threshold; a potential "translational threshold") or those that value only classifying the most
 homogeneous samples (0.8 threshold; a potential "biological/clinical threshold"). These
 analyses reveal that while the RFS outcomes of PDS classification remain the same, the
 higher threshold produces more mixed samples, but in doing so it classifies a far more
 homogeneous set of samples with the highest biological signalling associated with each class
 (**Figure R8**).

**Figure R8:** Ternary plots at default 0.6 (left) and 0.8 (right) PDS prediction probability
 thresholds in the FOCUS cohort. Kaplan-Meier relapse-free survival (RFS) analysis in
 GSE39582 cohort across PDS calls derived from 0.8 PDS prediction probability threshold.
 Forest plot displays univariate cox proportional analysis outcome comparing PDS1 with PDS2
 and PDS3 with hazard ratio (HR) and p-value. The PDS prediction probability barchart and
 density plots with 0.6 (middle-row) and 0.8 threshold (bottom-row).

To complement these PDS and imPDS heterogeneity assessments, we also performed a more
 in-depth interrogation of the potential cellular source of the n=626 gene sets underpinning the
 PDS classifier, using a cohort of laser capture microdissected CRC tissue stratified into
 epithelial and stromal compartments (GSE31279).

In line with the analyses presented above, we observe a clear signal that indicates the cellular
 compartmental origin of the gene sets underpinning PDS1 as epithelial, PDS2 as stromal and
 no enrichment for either compartment for the gene sets underpinning PDS3 (**Figure R9**).

Taken together, these assessments related to the transcriptional and histological
 heterogeneity of PDS, alongside the ability to refine the threshold based on what is most
 important for a user's specific question, enables the PDS classifier to provide value to the field
 overall in a way that can be refined to either increase the overall proportion of classified
 samples, or to focus on the identification of the most homogeneous transcriptional and
 biological signalling.

**Figure R9:** Heatmap visualisation (*left*) represents the PDS-specific gene sets ssGSEA
 scores examined across laser capture micro-dissected epithelium and stroma samples.
 Boxplots (*right*) highlights epithelium or stroma association for the specific gene set of PDS1
 (*top*), PDS2 (*middle*) and PDS3 (*bottom*).

*Of more concern is the potential impact of the PDS classifier. Clinical utility is one measure of*
 *impact, and the manuscript describes survival differences with GSE39582 and PETACC3 (Fig*
 *2K). But these cohorts were published in 2013 and 2009 and may not represent more modern*
 *treatments, especially immunotherapy for MSI-H CRCs. Microsatellite instability is a major*
 *CRC biomarker and is broadly classified as PDS2 (CMS1), but is not well addressed.*

The Reviewer is entirely correct in that over time the treatment options available to patients
 have changed, and will continue to change, since the generation of the cohorts we have used
 in this study. While we cannot accurately pre-empt these advances and the new treatment
 schedules they will bring, to address this point we propose two solutions:

The design underpinning our PDS study, and the public release of the transcriptional PDS
 classifier, means that it can be deployed on data from current/future trials as they emerge. Our
 classifier has been developed on tumour tissue, regardless of subsequent treatments or
 outcomes measured, and we have demonstrated how the biological traits underpinning each

PDS class can be made more robust through alterations of the threshold. Therefore, we
 believe that the clinical value PDS classification (or CMS, iCMS, etc..) according to any new
 treatment schedule can easily be assessed when these data emerge.

Secondly, to directly address the Reviewers point about MSI clinical management, we have
 performed a re-analysis of the clinical associations of PDS classification in our cohorts using
 the MSS cases only (**Figure R10**). These analyses reveal that similar clinical value is observed
 in the absence of MSI cases, which as the Reviewer rightly says will continue to be stratified
 towards immunotherapeutic treatment options.

GSE39582 (MSS ONLY):

PETACC-3 (MSS ONLY):

**Figure R10: Kaplan-Meier relapse-free survival (RFS) analysis on microsatellite stable**
 **(MSS)**

only patients across PDS (*left*) and only in CMS2-PDS1/PDS3 subset vs all CMS4
 (*right*) in

GSE39582 (*top*) and PETACC-3 (*bottom*) cohorts.

Overall, this is an impressive amount of work that demonstrates a new CRC classification

*method. Potential addressable weaknesses are problems with biological intratumoral*
*expression heterogeneity and uncertain clinical impact or utility in a more modern therapeutic*
*setting.*

*We thank the Reviewer once again for their feedback, and for the specific points for us to*
*address in this revised manuscript.*

**Reviewer #3:**

*Remarks to the Author:*

*Malla et al. propose a novel bulk RNAseq based classification system for colorectal cancer,*
*utilizing previously established signaling pathway associated genesets to inform classification*
*of colorectal cancer, and then extrapolate this to various cancer types. This method*
*purportedly improves on current classification approaches (CMS, CRIS) which use bulk*
*genomic and transcriptomic data to inform their classification system. The authors suggest*
*that this PDS classification system is a better predictor of tumor development and treatment*
*response than current methods, as it can classify tumors into biologically discrete groups*
*regardless of mutational status. Overall, the approach of using shared signaling programs to*
*develop classifiers is interesting, but in this manuscript no new biology-driven classifiers are*
*proposed. The authors use generic hallmark and other genesets and tell us how these*
*pathways are differentially expressed in distinct tumors. This appears to be a very blunt*
*approach and as such it is not surprising that using generic genesets yields a rather blunt*
*classification into 3 subgroups, which seem to largely reflect extent of tumor/stromal*
*representation in the tumor (PDS1), cell cycle (PDS2) or the absence of either (PDS3). Given*
*the generic nature of the genesets used, it is also not surprising that the classifiers developed*
*in CRC apply broadly to multiple cancer types. The study would be of greater interest if more*
*granular data, such as those obtained from clinically annotated single cell genesets, could be*
*used to derive de novo factors/genesets, and then tested across bulk cohorts.*

*The use of transcriptome based molecular subtyping in CRC has failed to be adopted in clinical*
*practice because existing classifiers are too blunt, and also because some studies using them,*
*have had contradictory results in terms of their utility in association with prognosis – this is*
*particularly true for CMS4. A major confounder in these approaches has been the confounding*
*of signals from tumor vs stroma, and the imprecise nature of biopsy sampling of tumor/stroma*
*in clinical samples. Given the blunt nature of the classifiers developed here, I do not think this*
*study represents a significant, clinically actionable advance over existing classifiers. Further,*
*the study primarily performs transcriptomic analyses with a small amount of IF validation.*
*Some of the more interesting hypotheses proposed, e.g. the epigenetic regulation of the*
*neuroendocrine state, require and lack experimental validation. For these reasons, I do not*
*see this study as a strong candidate for publication in Nature Genetics.*

However, perhaps the most intriguing part of the study is the data showing lack of
representation of specific PDS programs in PDX and GEMM models. This observation is also
confounded by potential differences in tumor/stroma composition between these cohorts.
Ideally, a rigorous comparison would involve first profiling patient tissue, deriving PDX from
the tissue and then profiling the PDX to allow direct comparison of tumor and PDX from the
same patient, ideally at single cell transcriptomic and protein IF levels. If such rigorous
validation could be performed, this may provide robust evidence for which cell states are and
are not retained in ex vivo or cognate GEMM models, in turn informing the development of
future improved models and influencing the choice of appropriate preclinical models for drug
testing or mechanistic analysis.

We thank the Reviewer for the comments and take onboard both the positive and critical
feedback. As stated at the start of this rebuttal letter, we firmly believe that the review process
has resulted in the development and presentation of a far stronger story in the revised
manuscript. We believe that the comments from this Reviewer have directed our analyses to
address critical points that highlight the novelty and value provided to the field by this work.

We now present a series of data that aims to address the Reviewer's general points, primarily
related to the novelty/value that can be derived from the approaches we have used to develop
the PDS classification. These new assessments presented in the revised manuscript also aim
to address each of the specific points raised above by using the granular single cell data types
indicated by the Reviewer, which we feel now highlight the subtle biologies that we have
uncovered, and also more testing of pre-clinical systems to identify the reasons underpinning
the lack of suitable models currently.

Importantly, by performing a series of comparisons with the recently published single cell iCMS
discovery study, we demonstrate the unique nature of the biology underpinning PDS3
classification, which highlights how our pathway-level approach in bulk was able to identify
important developmental and cell fate-related biology that has been repeatedly overlooked
using gene-level discovery in either single cell data or bulk.

We first wish to address two related points raised by the Reviewer in the first paragraph,
which were also similar to a point raised by Reviewer 1, namely:

- 1. This method purportedly improves on current classification approaches (CMS, CRIS)
which use bulk genomic and transcriptomic data to inform their classification system.
- 2. The authors suggest that this PDS classification system is a better predictor of tumor
development and treatment response than current methods, as it can classify tumors
into biologically discrete groups regardless of mutational status.

We initially set out to develop a new classifier based on pathway-level data, in the hope that it
would reveal biological/clinical information that is not currently associated with existing bulk
and single cell classification systems like CMS and iCMS respectively. We can see how our
study could be seen as competing with existing classifiers, however it was never our intention
to be as blunt as to present our new classifier as a stand-alone system; a point that had initially
inspired the multi-subtype summary figure in the initial submission that clearly shows how
these classifiers all provide different, yet equally important, information. Adding more
complexity to tumour characterisation may seem counterintuitive, but these multiple biological
labels are all revealing addition tiers of information.

In this revised manuscript we have made this much more explicit in the text, particularly the
Introduction and Discussion, and have presented our data in the new figures in a way that we
believe clearly highlights the value of each system, alongside a revised summary diagram.

Relevant section in the Introduction:

*“Although this was developed in bulk CRC, which includes signalling from both tumour and*
*non-tumour origin, PDS reveals a previously unseen continuum of epithelial cell states within*
*CMS2, associated with cell cycle, transcriptional activity, and stress response. Furthermore,*
*these subtle intrinsic biological traits are clearly distinct from the iCMS classification system*
*when applied to the same single cell cohorts, reinforcing the importance of using multiple*
*distinct classifiers to maximise the biological value derived from such data. Overall, these data*
*support the use of PDS in conjunction with existing subtyping approaches to ensure that*
*tumour studies are informed by multiple tiers of cancer-relevant information that cannot be*
*fully revealed by individual subtyping methods.”*

Relevant sections in the Discussion:

*“...the biological pathway activation approach that underpins PDS classification provides a*
*novel basis for forward and reverse translation studies, in combination with these existing*
*subtyping approaches.”*

*“In summary, our study presents an approach to tumour classification that relies on patterns*
*observable within broad biological signalling rather than individual gene clustering. Although*
*it would be easy to view independent subtyping approaches as competing, like CMS, iCMS,*
*CRIS and PDS, however it will only be through classification of samples using these*

*complementary approaches in parallel that we can reveal biological granularity and new*
 *mechanistic understanding and would otherwise be missed by using them in isolation.*
 *Maximising the amount of phenotypic information we derived from tumour data, using*
 *classification tools that providing synergistic insights into different yet equally important*
 *transcriptional signalling, will provide the most comprehensive landscape for stratification of*
 *tumours into discrete biological groups that can support more robust mechanistic and*
 *translational studies.”*

The new analyses will now be presented and described in more detail below.

In the initial submission, we put particular focus on PDS1 and PDS3, which align to
 elevated

expression of MYC targets and canonical stem-like features in PDS1, as opposed to
 elevation

of PRC targets and reduced canonical/regenerative stem-like features in PDS3 (**Figure
 R11**).

**Figure R11:** Scatterplot displays correlation between MYC targets and PRC targets ssGSEA
 scores, with PDS annotations in the FOCUS cohort (*left*). This association between MYC
 targets and PRC targets was further shown in a mouse colon epithelium dataset (GSE143915)
 where cell type annotations highlight the stem association with Myc Targets and more
 differentiated cell types such as enteroendocrine (EEC) cells with PRC Targets (*right*).

In this revised manuscript, we performed a much deeper investigation of PDS biological traits
 within a series of new single-cell datasets and experimental pre-clinical models. These
 assessments aim to address the specific comments from the Reviewer in terms of using more
 granular datasets to develop de novo classifiers/genesets that can be applied across bulk.
 While we maintain our initial PDS discovery using the bulk cohorts, we push further with these

approaches by using the more granular data sets to apply these new classifications to
 additional human/mouse bulk and lineage-specific single cell cohorts. We believe that the use
 of these new data types provides additional independent validation of the PDS biology and its
 relationship with MYC targets and PRC targets. Furthermore, these new assessments provide
 multiple layers of additional mechanistic insight into the biology encapsulated PDS tumour
 biology and its associations with a subtle biological signalling cascade along a stem-
 differentiation trajectory.

Utilising single-cell RNA-seq (scRNA-seq) data from murine derived CRC organoid models
 (Qin and Cardoso Rodriguez et al., BioRxiv 2023) we demonstrate how this PDS-related MYC-
 PRC signalling axis aligns with a gradient of intrinsic biology along cellular developmental
 dynamics and stem-differentiation trajectory (**Figure R12**). MYC targets are elevated during
 the early cellular development, prominent in stem cells, whereas the cell development
 progresses, there are gradual increment in PRC targets expression and highly associated with
 more differentiated cell types.

**Figure R12:** Single-cell RNA-seq dataset derived from murine CRC organoid models with cell
 type annotations were utilised to perform RNA velocity and pseudotime analysis; shown in
 PHATE visualisation. Top tertiles of MYC targets and PRC targets ssGSEA score were defined
 as “High” groups corresponding to MYC target-High/PDS1-like and PRC targets-High/PDS3-
 like. PHATE visualisations represent pseudotime analysis and RNA velocity with cells
 annotated as MYC targets High or PRC targets High.

Utilising these classification scores, we can assess the lineages that are most aligned
 with
 PDS1/MYC-targets compared to PDS3/PRC-targets, giving additional evidence
 supporting
 the biologically differentiated state of PDS3 tumours (**Figure R13**).

**Figure R13:** Pseudotime analysis score represented in boxplots comparing MYC Targets High
 vs PRC Targets High (left) and different annotated cell types (right). The red and green line
 displays the median of pseudotime scores in MYC Targets High and PRC Targets High,
 respectively.

Moving from this organoid cohort into the granular single cell dataset used for the development
 of the iCMS classification system, we confirm the presence of this biological trajectory in single
 cell data derived from CRC patients (Figure R14). Pseudotime and RNA velocity analysis
 provides evidence of cellular trajectory movements from MYC targets High cells towards PRC
 targets High cells at the patient level, again suggesting this innate cellular biology is identifiable
 across the diverse range of data types investigated in our study.

**Figure R14:** UMAP visualisation of pseudotime analysis and RNA velocity on the
scRNA-seq

CRC patient data (Joanito et al., NatGen, 2022), where each cluster represents
patients.

UMAP visualisation on RNA velocity of the two patients from the cohort: CRC-JSC-S05
(left)

and SMC16 (right) with cell annotation of MYC targets High and PRC targets High.

These same traits are also clearly evident when the cohort is clustered using PDS-
specific

gene sets and overlaid with the same MYC- and PRC-target scores, as we again
show a

610 strong correlation between MYC- and PRC-target scores in individual epithelial cells,
in a

611 pattern that aligns strongly with pseudotime and developmental trajectories (**Figure
R15**).

**Figure R15:** Scatterplot of MYC targets and PRC targets ssGSEA scores in a single cell
data

annotated with pseudotime score (left). UMAP visualisation of the CRC single cell data

with

617 cell annotation with MYC targets High and PRC targets High where arrows indicate
RNA

velocity (right).

Using this cohort, we now also provide a new series of analyses to test if these
biologies still

align to the stem-like features we present in bulk transcriptional and *in situ* hybridisation

(ISH) assessments in Figure 4. These new analyses again align with the findings that

PDS3/PRC-target signalling is associated with an absence of canonical or regenerative

stem-like features (Figure R16).

**Figure R16:** UMAP visualisations of the MYC targets, PRC targets ssGSEA scores
and the

628 MYC targets-PRC targets blend (top). UMAP visualisation with ssGSEA scores using
crypt-

629 base columnar (CBC) and regenerative stem cells (RSC) gene signatures (bottom).

***The value of a new discovery approach in more granular***
***single cell data***

*The Reviewer states:*

*“The study would be of greater interest if more granular data, such as those obtained from*
*clinically annotated single cell genesets, could be used to derive de novo factors/genesets,*
*and then tested across bulk cohorts.”*

We certainly agree with the Reviewer that this would be a valuable approach, and this point
inspired many of our assessments above. Furthermore, in the recent study by, Joanito *et al.*,
the authors deployed the methodology, format, and approach (described by the Reviewer
above) during the development/testing of the recent iCMS classification system published in
*Nature Genetics*. As such, this established classification presents us with the opportunity to
compare the biologies identified using our pathway-level bulk approach, with those identified
using the gene-level single cell methodology described by the Reviewer.

We believe the data we present in the revised version reveals nuanced PDS-related tumour
biology that is indistinguishable using CMS or iCMS. Our PDS approach stratifies tumours
based on subtle alterations in the balance of stem/differentiation cellular states not apparent
in tissue and have been overlooked using this iCMS single cell approaches, or the previous
gene-levels classifiers developed in bulk.

***Comparison of pathway-level class discovery in bulk***
***data (PDS) with gene-level class***

***discovery in single cell (iCMS)***

Using the iCMS classification labels provided from the Joanito *et al.*, study in the single-cell
data we perform a comparison to the findings from our PDS study. First of all, as with many
studies that perform discovery using single cell data, the complete cohort was collapsed into
“**pseudobulk**” ahead of iCMS classification system development. This means that the outputs
are based on a summarised score of all epithelial cells according to each individual patient,
rather than biologies within each individual single-cell; an appropriate approach and we do not
question its validity.

This results in a patient-level classification for iCMS, where each individual cell within a patient
sample will be assigned the same iCMS label; meaning that unlike our PDS data that clearly
indicates varying degrees of stem-differentiation balance across every tumour, iCMS gives no
scope for the presence of such intra-tumoral heterogeneity and overlooks these subtle shifts
in cellular states (**Figure R17**).

Therefore, unlike the cell developmental trajectory that emerged from our MYC/PRC targets

signatures, iCMS stratification characterises biology that is clearly distinct from the cellular
 trajectories and biological traits associated with PDS.

**Figure R17:** UMAP visualisation and RNA velocity of CRC patients single cell data
 annotated

with iCMS classification labels from Joanito et al. Violin plot examines pseudotime
 scores for

single cells across iCMS groups.

Given the absence of a single sample classification tool in the original iCMS system, and in
 order to make our PDS-related single cell assessments comparable, we overlay individual
 iCMS2 and iCMS3 ssGSEA scores, rather than blunt classification calls, on each individual
 single cell.

Now, instead of classifying every cell as iCMS2 or iCMS3, this new analysis assigns iCMS2
 and iCMS3 labels based on the strength of that signalling to each individual cell (denoted by
 increasing colour of each cell) according to the colour threshold legend on the figure below.
 This results in a clustering of strong iCMS2 cells in the left of the UMAP, strong iCMS3 cells
 towards the top centre/left, yet leaves a large void across the right side of the UMAP; the
 precise region that contains single cells with the strongest PRC target expression and PDS3
 traits (**Figure R18**).

**Figure R18:** UMAP visualisations represent single cells annotated with iCMS2 and iCMS3
 ssGSEA scores, and the blend of the two. The annotated region inside the dashed red line
 indicates the portion of single cells that do not strongly associates with either of the iCMS gene
 signatures.

Furthermore, we can also show the PDS-related heterogeneity that resides within the
 uniform iCMS labels, as application of single cell scores in either iCMS2 or iCMS3
 samples independently indicates the presence of varying levels of distinct PDS biology (**Figure R19**).

**Figure R19:** Violin plots display the measure of differentiation potency using
 Correlation of Connectome and Transcriptome (CCAT) method, where higher the score the
 greater the potential of the cell to differentiate; shown in each CRC patient in iCMS2 (*left*) or iCMS3
 (*right*).

While assessment of these UMAPs in a 2D way can sometimes overlook associations that are
 only apparent along different dimensions/axis, we next complemented these new data in the
 revised manuscript with a series of assessment and data visualisations for this rebuttal letter
 only.

Therefore, we next examined if the MYC targets and PRC targets overlaps with iCMS2 and
 iCMS3 in terms of the level of ssGSEA scores in a 3D UMAP visualisation below. We again
 observe that MYC targets and iCMS2 are highly correlated to each other, whereas iCMS3
 partly overlaps with MYC targets. Overall, these assessments again support our main findings,
 a clear majority of cells cannot be explained by either iCMS2 or iCMS3, and corresponds to
 large portion of PRC targets High cells representative of PDS3 (**Figure R20**).

**Figure R20:** 3D UMAP visualisations where different gene set specific ssGSEA score
 (MYC targets, PRC targets, iCMS2 and/or iCMS3) are blended to observe overlaps
 between the
 714 biological signalling and association across single cells in the data.

[Link to 3D UMAP plots (download first to view): 1, 2, 3, 4, 5] Shown for review only.

*Absence of de novo biology-driven classifiers proposed in our study*

As part of our study, we have developed the PDS classifier, and characterised the biologies
 underpinning each individual class. Furthermore, to ensure that our PDS classifier can be
 utilised by any researcher, we developed and released the companion “PDSclassifier” R-
 based accurately to ensure that end-users could classify tumours into the same robust clusters
 identified during class discovery. While developed initial for human data, we also generated a
 companion mouse equivalent package, to enable users to classify PDS in mouse GEMM data.
 To complement the initial submission, we made this classification package, and the tumour
 cohorts underpinning its development, fully available open access from our research groups
 GitHub page: <https://github.com/MolecularPathologyLab/PDSclassifier>.

Further to this, to address the Reviewers point about the absence of biology-driven classifiers
 proposed in the study, for this revised manuscript we have also generated a single sample
 classification tool that can be used across bulk and/or single cells cohorts in both human and
 mouse based on the MYC- and PRC-target biologies aligned to PDS1-3. Utilising our findings
 with these gene signatures, presented in the revised manuscript and through the responses
 to Reviewer 3’s requests, we develop a new methodology to enumerate the level of stem-
 differentiation state in individual single cells or across the overall landscape of a bulk tumour,
 in the form of the ‘Stem Maturation Index’ (SMI). This classifier is now embedded within an
 updated PDSclassifier to accompany this revised article.

Our analyses using SMI, included in the revised Figure 5 and subsequent figures, clearly
 demonstrates its alignment with PDS classification and the biologies underpinning PDS across
 single cell, FACS bulk lineages, or overall tumour bulk (**Figure R21**).

**Figure R21:** Schematic highlights generation of “Stem Maturation Index” and the
 implementation of SMI in the PDSclassifier to be used in both bulk tissue and single-cell data.
 UMAP visualisation of SMI in the murine organoid scRNA-seq data. Application of SMI in
 mouse colon epithelial data across cell types (*left*) and CRC bulk data across PDS (*right*).

**Testing our SMI classification against the established
 iCMS classifier**

As we propose that this new SMI classifier tool can give a clear measurement of an individual
 cells comparative position along the stem-differentiation axis in bulk and single cell data, we
 test this in the same single cell cohort used to develop the iCMS classifier. These assessments
 again clearly show that SMI most strongly associates with the cells that appear to be
 overlooked by the iCMS classification approach (**Figure R22**).

**Figure R22:** UMAP visualisation with SMI annotation on the CRC patient scRNA-seq data.
 SMI shows strong positive correlation with pseudotime score. UMAP visualisation represents
 SMI (*left*) and iCMS (*right*) where portion of positive SMI biology corresponding to PRC
 targets/PDS3 is not presented by iCMS. PHATE visualisations accentuate this point further
 where PRC target/PDS3 biology is distinctive from iCMS.

Taken together, these data in our revised manuscript highlight how our PDS, MYC-PRC and
 SMI *de novo* classification tools can be used to identify and enumerate the presence of a “third
 plane” of cell differentiation state, which clearly sits outside of the stem phenotypic landscapes
 explained by the previously developed stem cell index or the iCMS approaches. Thus, while
 gene-level approaches to classification can give you signalling cascades that are associated
 with stem-rich biologies, our PDS approach gives a wider macro-view of the overall state of a
 bulk tumour, or individual single cell along a developmental cell fate differentiation axis (**Figure**
 **R23**).

**Figure R23:** Tumour Epithelial Landscape describe cellular stem-differentiation phenotypic
 plane measured by “Stem Maturation Index” (SMI), which is highly enriched for PRC targets
 and strongly associated with PDS3 tumours.

Importantly, the development of this biology-driven classifier now gives the field an additional
 tool, alongside the new PDS classifier package, to derive more biological information from any
 human/mouse bulk or single cell cohort. Our revised summary figure (**Figure R24**) again looks
 to highlight the strength of utilising these classification tools in a non-competing way, where
 each approach used in combination can add value that isn’t available by use any of them in a
 stand-alone way.

**Figure R24:** In search for biology-driven molecular CRC tumours using bulk tissue data, led
 to development of pathway-derived subtypes with distinctive molecular, phenotypic, and
 clinical association. Further exploration of the PDS-driven biology in single cell, uncovered
 stem-differentiation cellular states and developmental trajectories, in the form of Stem

**Maturation Index (SMI).**

***Absence of appropriate pre-clinical models***

We next sought to address to Reviewers points related to the absence of appropriate pre-
clinical model systems for PDS3 in particular. The Reviewer has indicated the value of a PDX-
led approach for assessing depleted populations following engraftment, and while such a
study would certainly be informative and valuable, it follows an entirely distinct format to the
one we have presented here thus far and is therefore outside the scope of the current
manuscript. We do however use the underlying theme from the Reviewers point, and apply
that to the GEMM-derived organoid models that we have assessed previously from bulk
sampling in the initial submission, complemented with the single cell data we present for the
first time in Figure 5 (**Figure R25**) of the revised manuscript, complemented with a series of
ex vivo assessments of cellular populations that are selected for during pre-clinical culturing.

**Figure R25:** PHATE visualisations of scRNA-seq murine organoid models: AK
 (left), WT
 (middle) and WT+WENR (right), annotated with cell types. WENR = WNT3A, EGR,
 Noggin,
 R-Spondin-1. These models are then represented in Waddington plots, annotated
 with cell
 type (top) and MYC-PRC targets High groups (bottom).

These assessments are complemented with SMI measurements, which again give a
 clear
 picture of the PDS-related biologies within each lineage population according to
 genotype
 and culture conditions (**Figure R26**).

**Figure R26:** PHATE visualisations of scRNA-seq murine organoid models: AK
 (left), WT
 (middle) and WT+WENR (right), annotated with SMI.

Additionally, we provide evidence on association of high proliferation rate in MYC targets/stem
 cells in comparison to PRC targets/differentiated cell types in single cell murine organoid data
 (Figure R27). Level of proliferation can also be observed for each individual models we
 highlighted above where it suggests addition of these growth factors can push proliferation
 higher in addition to stem-like phenotype in WT.

**Figure R27:** PHATE visualisation of scRNA-seq murine organoid data annotated with
 proliferation index ssGSEA score (*top-left*). Violin plots examining proliferative index across
 cell types (*top-right*). PHATE visualisations of scRNA-seq murine organoid models: AK (left),
 WT (middle) and WT+WENR (right), annotated with proliferative index and single cell density.

Finally, we perform ISH for key genes that are associated with these biologies, using ex vivo
 cultures of WT and AK organoids, grown in the presence and absence of these growth factors
 (EGF, Noggin, and R-Spondin-1).

**Figure R28:** Representative Ki67⁺ and BrdU ISH images of mouse SI organoids on
tamoxifen
induced VilCreER Apcf1/fl KrasG12D/+ (AK) and uninduced wild-type (WT) with/without
EGF,
Noggin and R-spondin-1 (ENR). These has been scored for positivity and visualised in
barplot
below each ISH image panel.

Taking Ki67 and BrdU as examples (**Figure R28**), we show here the low level of Ki67⁺/BrdU
stains in the WT whereas addition of ENR substantially increases Ki67⁺/BrdU staining thus
elevation of proliferation rate, which resembles closely to the AK murine organoids without any
addition of growth factors. This experiments very much supports our findings shown in the
single cell murine organoid data above. Overall, our findings emphasise on the lack of models
for PDS3 specific biology due to slow-cycling nature that is difficult to recapitulate in the
existing preclinical models.

***Epigenetic regulation of the PDS3*** ***(entero/neuroendocrine) differentiated state.***

Given the absence of an appropriate PDS3 model, and in line with the Reviewers point related
to investigating the regulation of the epigenetic state to drive PDS3 biology, we performed a
series of *in vivo* studies in an attempt to “push” PDS1 AK models closer to this PDS3 state.
These experiments are presented here in the rebuttal letter only, given their outcomes.

In the first *in vivo* experiment, we tested the selective EZH2 inhibitor, Tazemetostat (EZH2i;
250mg/kg orally twice daily for 7-9 days, depending on signs) following induction of AK models
(**Figure R29**). We show the outcome of the experiments in the data below where we compared
using the Hallmarks and MYC/PRC targets gene sets. No distinguishing feature can be noted
between the treated and untreated, including KRAS signal DN prominent in PDS3 in human
data. Moreover, there is no difference between EZH2i treated vs untreated for Myc targets and
PRC targets.

865

**Figure R29:** Intracolonic induction of AK GEMMs were treated with or without EZH2 inhibitor
 (Tazemetostat, 2x daily) and primary tumour mass were extracted for RNA sequencing.
 Biological associations were examined with ssGSEA scores using the Hallmarks gene sets
 and visualised in a heatmap (shown above). Comparison between EZH2 inhibitor (EZH2i)
 treated vs EZH2 vehicle (EZH2v) control for Myc targets and PRC targets showed no
 significant difference. **Shown for review only.**

In the second in vivo experiment, we tested EZH2 inhibitor (Trazemetostat) in addition
 with/without MEK inhibitor (Trametinib) on intraperitoneally induced AK GEMMs where we
 once again examined Hallmarks and MYC/PRC targets gene sets (**Figure R30**). Notably, if
 we compare the EZH2i and the control groups, we do not see any evidence of biology closely
 resembling that of PDS3. Similarly, there was no difference between the groups for Myc targets
 and PRC targets.

**Figure R30:** AK GEMMs were delivered two treatments, once before and after intraperitoneal
 (IP) induction with tamoxifen. The treatments were combinations of EZH2 inhibitor
 (Tazemetostat) and/or MEK inhibitor (Trametinib). The tissues were harvested at the end of
 the treatments and were RNA sequenced. Biological associations were examined with
 ssGSEA scores using the Hallmarks gene sets and visualised in a heatmap (shown above).
 Comparison between different treatment groups for Myc targets and PRC targets showed no
 significant difference. **Shown for review only.**

*Potential for confounding of results due to stromal features*

The Reviewer raises a point similar to that of Reviewer 2, in relation to how confounded these
 new subtypes may be to the changes in stromal content, however we appreciate that this
 reviewer was specifically relating this to PDX/GEMM v human comparisons.

We have previously performed analyses on how stromal intratumoural heterogeneity (ITH)
 can influence gene-level classifiers like CMS (Dunne et al., CCR 2016) and ways to avoid it
 in classifier development (Dunne et al., Nat Comms 2017). More recently, we have measured
 the potential for ITH across widely used transcriptional signatures (Fisher et al., CCR 2022).
 In this 2022 CCR publication we developed a web application that allows users to measure
 the potential for ITH to confound any gene, geneset or transcriptional signature, accessible
 via <https://confoundr.qub.ac.uk/>. We have tested the performance of our new PDS classifier
 using these approaches, with the finding detailed comprehensively in the response to

Reviewer 2 above.

*Addressing the use of generic gene sets for downstream characterisation*

Using our experience from the development of the data application above, and in order to
 address the Reviewers general concerns about the use of generic and blunt gene sets for the
 development and characterisation of PDS in this current study, we have developed a
 companion data application to support this revised manuscript. This new data application is
 described in the revised manuscript:

“While data availability has increased, interrogation of molecular cohorts is critically
 constrained due to the programming skills required to perform even basic analyses, meaning
 molecular cohorts rarely realise their full potential. Therefore, to complement the non-
 exhaustive characterisations according to PDS presented here, and to support the FAIR
 principles, we developed the ‘SubtypeExploreR’ platform (Figure R31). This enables any
 user to interrogate any transcriptional gene and/or signature, including existing signatures
 from numerous databases or an unlimited combination of *de novo* unpublished classifiers,
 according to PDS and other CRC subtypes in our bulk cohorts (Figure
 4G;

<https://subtypeexplorer.qub.ac.uk/>). This resource will ensure that these data are not just
 accessible, but also become (re)usable by a much wider audience, ensuring that users can
 test their own biology of interest in these datasets according to the subtyping classification
 system of their choice.”

<https://SubtypeExploreR.qub.ac.uk/>

**Figure R31:** Two CRC cohorts: FOCUS and GSE39582, has been utilised in a
newly developed ShinyApp platform called “SubtypeExploreR”, which provides users to
investigate their gene or gene sets of interest across three different CRC molecular
subtyping approaches, including PDS, CMS and iCMS.

*Additional Points:*

*Figure 2K. Authors show KM plots for the non-randomized GSE39582 cohort and PETACC-3*
*cohort based on PDS classification system compared to CMS subtypes substratified by PDS*
*subtypes. The authors should also show KM plots for CMS1, CMS2, CMS3, CMS4*
*(classification system they are comparing to in this instance).*

We thank the Reviewer for these specific points and have now detailed these
results in Supplementary Figure 2 (**Figure R32**) of the revised manuscript.

**GSE39582**

All

CMS-UNK

Excluded

**Figure R32:** Kaplan-Meier relapse-free survival (RFS) analysis in GSE39582 cohort for
 CMS with (left) or without (right) unclassified (UNK) samples.

*The authors claim that the PDS3 subtype contains a “senescence-like physiology” but do not*
 *adequately characterize this in the manuscript.*

*Again, we thank the Reviewer for this. In the initial manuscript we had not included these data*
 *and we can see that the terminology we used of “senescence-like” was confusing. While the*
 *slow-cycling nature of PDS3 is somewhat aligned to a potential senescence-like physiology, it*
 *is not associated with assessment of senescence signature scores using various established*
 *gene sets in cohorts used in this study (Figure R33).*

*As such, we have removed this statement from the revised article to avoid confusion.*

**Figure R33:** Heatmap visualisation on ssGSEA score for the senescence-related
 gene signatures in the FOCUS cohort, annotated with PDS prediction probabilities and PDS
 calls.

Shown for review only.

*The authors assessed the clinical landscape of the FOCUS cohort to explore significant*
 *differences in PDS classification of tumors such as gender or sidedness, but do not comment*
 *on other clinical factors such diagnosis age, race/ethnicity, metastatic disease, MMR status,*
 *treatment before surgery. (Figure 2, Supplemental 2).*

As the Reviewer has noted, previously we presented assessment of PDS association with
 gender and tumour sidedness. Here, we show the other clinical factors assessed in terms of
 PDS such as tumour stage, MSI, CIMP and diagnosis age (**Figure R34**). Regarding the
 race/ethnicity, we do not have the information on these cohorts, and also this are adjuvant
 cohorts.

**Figure R34:** Clinicopathological assessment on FOCUS and GSE39582 cohorts. Proportional
 stack barcharts display gender, tumour sidedness, stage, MSI and CIMP status across PDS
 in the FOCUS (top) and GSE39582 (bottom) cohorts. There was also no difference in age at
 the time of diagnosis across PDS in both cohorts shown in boxplots.

*Typo in legend of Figure 2K (“CMS2/PDS1, CMS2/PDS1 vs all CMS4 samples only (right).”)*
 *Should be changed to “CMS2/PDS1, CMS2/PDS3 vs all CMS4”*

*The typo has been corrected.*

*“Finally, using whole face tissue sections from our SPINAL cohort, we observed that protein*

*expression for the differentiated EEC marker, such as chromogranin A (CRGA)...” This is a*
*typo and should be changed to “CHGA”*

The typo has been corrected.

Decision Letter, first revision:

26th Sep 2023

Dear Dr Dunne,

Thank you for submitting your revised manuscript "Pathway level subtyping identifies a slow-cycling biological phenotype associated poor clinical outcomes in colorectal cancer." (NG-A61420R1). It has now been seen by the original referees and their comments are below. The reviewers find that the paper has improved in revision, and therefore we'll be happy in principle to publish it in Nature Genetics, pending minor revisions to satisfy the referees' final requests and to comply with our editorial and formatting guidelines.

Sincerely,

Safia Danovi
Editor
Nature Genetics

Reviewer #1 (Remarks to the Author):

The authors conducted a comprehensive revision. All of my previous comments have been addressed. I do not have further questions.

Reviewer #2 (Remarks to the Author):

The authors have addressed my major concerns. The paper is a remarkable analysis. Concerns remain on the ultimate utility and impact of classification schemes that attempt, in general, to place individually very diverse tumors into a limited number of categories.

Reviewer #3 (Remarks to the Author):

Malla et al., present their revised manuscript on developing a novel pathway based classifier to subtype primary colorectal cancer. Overall, the revised manuscript is substantially improved, incorporating a large amount of new work. Specifically, the development of the "biology based" classifier has been clarified and strengthened, additional analyses have been performed comparing PDS with existing classifiers and delineating how tumor heterogeneity influences the classification system, including by mapping the classifier onto available single cell RNAseq datasets. These efforts have substantially strengthened the computational aspects of the manuscript. In particular, the work done by the team to generate publicly accessible GUI tools to assess gene expression based on PDS and other classifiers is to be commended, and will undoubtedly increase the impact of the work by making it more accessible to a wide range of biological users.

The authors have also made serious attempts to deepen the biological insights to be gained regarding PDS3. The work showing MYC/PRC expression anticorrelation relative to PDS3 is interesting, but not experimentally tested. This work has clearly been challenging, as shown by their inability to derive PDX models representing PDS3, including upon treatment with epigenetic modulators. While the Kaplan Meier curves showing a strong relationship between PDS3 classification and poor outcomes are striking, the study lacks compelling evidence that such slow-cycling cells really can be "awoken" to drive tumor growth under specific circumstances. Ideally, this would require isolation of PDS3 cells from fresh clinical samples, and then organoid formation or similar assays to define their tumor regenerative capacity. The absence of such functional evidence limits the impact of the study. However, given the interesting, and now well-validated new "biology-based" approach for classifying tumors, the lack of clear novel functional insight or clinical relevance need not, in my view, preclude publication in Nature Genetics if the study is sufficiently strong on the basis of the computational advance.

Final Decision Letter:

3rd Jan 2024

Dear Dr Dunne,

I am delighted to say that your manuscript "Pathway level subtyping identifies a slow-cycling biological phenotype associated with poor clinical outcomes in colorectal cancer." has been accepted for publication in an upcoming issue of Nature Genetics.

Over the next few weeks, your paper will be copyedited to ensure that it conforms to Nature Genetics style. Once your paper is typeset, you will receive an email with a link to choose the appropriate publishing options for your paper and our Author Services team will be in touch regarding any

additional information that may be required.

Your paper will be published online after we receive your corrections and will appear in print in the next available issue. You can find out your date of online publication by contacting the Nature Press Office (press@nature.com) after sending your e-proof corrections.

Please note that *Nature Genetics* is a Transformative Journal (TJ). Authors may publish their research with us through the traditional subscription access route or make their paper immediately open access through payment of an article-processing charge (APC). Authors will not be required to make a final decision about access to their article until it has been accepted. [Find out more about Transformative Journals](https://www.springernature.com/gp/open-research/transformative-journals)

Authors may need to take specific actions to achieve [compliance](https://www.springernature.com/gp/open-research/funding/policy-compliance-faqs) with funder and institutional open access mandates. If your research is supported by a funder that requires immediate open access (e.g. according to [Plan S principles](https://www.springernature.com/gp/open-research/plan-s-compliance))

then you should select the gold OA route, and we will direct you to the compliant route where possible. For authors selecting the subscription publication route, the journal's standard licensing terms will need to be accepted, including <https://www.nature.com/nature-portfolio/editorial-policies/self-archiving-and-license-to-publish>. Those licensing terms will supersede any other terms that the author or any third party may assert apply to any version of the manuscript.

If you have not already done so, we invite you to upload the step-by-step protocols used in this manuscript to the Protocols Exchange, part of our on-line web resource, natureprotocols.com. If you complete the upload by the time you receive your manuscript proofs, we can insert links in your article that lead directly to the protocol details. Your protocol will be made freely available upon publication of your paper. By participating in natureprotocols.com, you are enabling researchers to more readily reproduce or adapt the methodology you use. [Natureprotocols.com](http://natureprotocols.com) is fully searchable, providing your protocols and paper with increased utility and visibility. Please submit your protocol to <https://protocolexchange.researchsquare.com/>. After entering your [nature.com](http://www.nature.com) username and password you will need to enter your manuscript number (NG-A61420R2). Further information can be found at <https://www.nature.com/nature-portfolio/editorial-policies/reporting-standards#protocols>

Sincerely,

Safia Danovi
Editor
Nature Genetics